# On the Generalization of Temporal Graph Learning with Theoretical Insights

## Abstract

Temporal graph learning (TGL) is a widely-used technique in various real-world applications, but its theoretical foundations largely remain uncharted. We fill in this gap by studying the generalization ability of different TGL algorithms (e.g., GNN-based, RNN-based, and memory-based methods) under the finite-wide over-parameterized regime. We establish the connection between the generalization error of TGL algorithms and ① "*the number of layers/steps*" in the GNN-/RNN-based TGL methods and ② "*the feature-label alignment (FLA) score*", where FLA can be used as a proxy for the expressive power and explains the performance of memory-based methods. Guided by our theoretical analysis, we propose *Simplified-Temporal-Graph-Network* (SToNe), which enjoys a small generalization error, the better overall performance, and a lower model complexity. Extensive experiments on real-world datasets demonstrate the effectiveness of SToNe. This paper provides critical insights into TGL from a theoretical perspective and paves the way for designing practical TGL algorithms in future studies.

## 1 Introduction

Temporal graph learning (TGL) has emerged as an important machine learning problem and is widely used in a number of real-world applications, such as traffic prediction Yuan & Li (2021); Zhang et al. (2021), knowledge graphs Cai et al. (2022); Leblay & Chekol (2018), and recommender systems Kumar et al. (2019); Rossi et al. (2020); Xu et al. (2020a). A typical downstream task of temporal graph learning is link prediction, which focuses on predicting future interactions among nodes. For example in an online video recommender system, the user-video clicks can be modeled as a temporal graph whose nodes represent users and videos, and links are associated with timestamps indicating when users click videos. Link prediction between nodes can be used to predict if and when a user is interested in a video. Therefore, designing graph learning models that can capture node evolutionary patterns and accurately predict future links is important.

TGL is generally more challenging than static graph learning, thereby requiring more sophisticated algorithms to model the temporal evolutionary patterns. In recent years, many TGL algorithms Kumar et al. (2019); Xu et al. (2020a); Rossi et al. (2020); Sankar et al. (2020); Wang et al. (2021c) have been proposed that leverage memory blocks, self-attention, time-encoding function, recurrent neural networks (RNNs), temporal walks, and message passing to better capture the meaningful structural or temporal patterns. For instance, JODIE Kumar et al. (2019) maintains a memory block for each node and utilizes an RNN to update the memory blocks upon the occurance of each interaction; TGAT Xu et al. (2020a) utilizes self-attention message passing to aggregate neighbor information on the temporal graph; TGN Rossi et al. (2020) combines memory blocks with message passing to allow each node in the temporal graph to have a receptive field that is not limited by the number of message-passing layers; DySAT Sankar et al. (2020) uses self-attention to capture structural information and uses RNN to capture temporal dependencies; CAW Wang et al. (2021c) captures temporal dependencies between nodes by performing multiple temporal walks from the root node and representing neighbors' identities via the probability of a node appearing in the temporal walks. GraphMixer Cong et al. (2023) combines MLP-mixer Tolstikhin et al. (2021) with 1-hop most recent neighbor aggregation for temporal graph learning.

Although empirically powerful, the theoretical foundations of the aforementioned TGL algorithms remain largely under-explored. A recent study Cong et al. (2023) have demonstrated that simple

Figure 1: Relationship between the generalization error (in Theorem 1) and empirical average precision score. Generalization error (GE) and average precision (AP) have an inverse correlation, i.e., the larger the GE, the lower the AP. Each marker is one experiment run. The same method's GE changes at each run because it depends on feature-label alignment, which changes with different weight initialization. More details on the computation of GE are deferred to Appendix I.4.

models could still exhibit superior performance compared to more complex models Kumar et al. (2019); Xu et al. (2020a); Rossi et al. (2020); Sankar et al. (2020); Wang et al. (2021c), but rigorous understanding of this phenomena remains elusive. Therefore, it is imperative to unravel the learning capabilities of these methods from the theoretical perspective, which will be in turn beneficial to design a more practically efficacious TGL algorithm in the future studies. Recently, Souza et al. (2022); Gao & Ribeiro (2022) study the expressive power of temporal graphs by extending the 1-WL (Weisfeiler-Lehman) test to the temporal graph. Souza et al. (2022) shows that using an injective aggregation function in TGL models is necessary to be most expressive, and that increasing the number of layers/steps in GNN-/RNN-based TGL methods or using memory blocks can increase the WL-based expressive power. However, WL-based expressive power analysis may not ① explain why one TGL method performs better than another when neither of them uses injective aggregation functions (e.g., mean-average aggregation), ② explain why existing TGL methods prefer to use shallow GNN structure (e.g., Xu et al. (2020a); Fan et al. (2021); Rossi et al. (2020) uses 1- or 2-layer in official implementation) or a small number of RNN steps (e.g., Wang et al. (2021c); Sankar et al. (2020) use less than 4 steps in official implementation) despite the theory suggesting deeper/larger structures, and ③ explain how well TGL algorithms generalize to the unseen data.

As an alternative, we directly study the generalization ability of different TGL algorithms. We establish the generalization error bound of GNN-based Xu et al. (2020a); Fan et al. (2021), RNN-based Wang et al. (2021c), and memory-based Kumar et al. (2019) TGL algorithms under the finite-wide over-parameterized regime in Theorem 1 via deep learning theory. In particular, we show that the generalization error bound decreases with respect to the number of training data, but increases with respect to the number of layers/steps in the GNN-/RNN-based methods and the feature-label alignment (FLA), where FLA is defined as the projection of labels on the inverse of empirical neural tangent kernel $\mathbf{y}(\mathbf{JJ}^\top)^{-1}\mathbf{y}$, $\mathbf{J} = [\text{vec}(\nabla_\theta f_i(\boldsymbol{\theta}_0))]_{i=1}^N$ as in Definition 1. We show that the FLA could potentially ① serve as a proxy for the expressive power measurement, ② explain the impact of input data selection on model performance, and ③ explain why the memory-based methods do not outperform GNN-/RNN-based methods, even though their generalization error is less affected as the number of layers/steps increases. Our generalization bound demonstrates the importance of "selecting proper input data" and "using simpler model architecture", which have been previously empirically observed in Cong et al. (2023), but lack theoretical understanding.

Guided by our theoretical analysis, we propose *Simplified-Temporal-Graph-Network* (SToNe), which achieves compatible performance on most real-world datasets, but with less weight parameters and lower computational cost. As shown in Figure 1, our proposed SToNe enjoys not only small generalization error theoretically but also a compatible overall average precision score. Extensive ablation studies are conducted on real-world datasets to demonstrate the effectiveness of SToNe.

**Contributions.** We summarize the main contributions as follows:

- (*Theory*) In Section 3, we analyze the generalization error of memory-based, GNN-based, and RNN-based methods in Theorem 1. Then, in Section 3.3, we reveal the relationship between generalization error and the number of layers/steps in the GNN-/RNN-based method and feature-label alignment (FLA), where FLA could also be potentially used as a proxy for the expressive power and explain the performance of the memory-based method.
- (*Algorithm*) In Section 4, guided by our theoretical analysis, we propose SToNe which not only enjoys a small theoretical generalization error but also bears a better overall empirical performance, a smaller model complexity, and a simpler architecture compared to baseline methods.

- (*Experiments*) In Section 5 and Appendix B, we conduct extensive experiments and ablation studies on real-world datasets to demonstrate the effectiveness of our proposal SToNe .

## 2 RELATED WORKS AND PRELIMINARIES

In this section, we briefly summarize related works and preliminaries. The more detailed discussions on expressive power, generalization, and existing TGL algorithms are deferred to Appendix H.

**Generalization and expressive power.** Statistical learning theories have been used to study the generalization of GNNs, including uniform stability Verma & Zhang (2019); Zhou & Wang (2021); Cong et al. (2021), Rademacher complexity Garg et al. (2020); Oono & Suzuki (2020); Du et al. (2019), PAC-Bayesian Liao et al. (2020), PAC-learning Xu et al. (2020c), and uniform convergence Maskey et al. (2022). Besides, Souza et al. (2022); Gao & Ribeiro (2022) study the expressive power of temporal graph networks via graph isomorphism test. However, the aforementioned analyses are data-independent and only dependent on the number of layers or hidden dimensions, which cannot fully explain the performance difference between different algorithms. As an alternative, we study the generalization error of different TGL methods under the finite-wide over-parameterized regime Xu et al. (2021a); Arora et al. (2019; 2022), which not only is model-dependent but also could capture the data dependency via feature-label alignment (FLA) in Definition 1. Most importantly, FLA could be empirically computed, reveals the impact of input data selection on model performance, and could be used as a proxy for the expressive power of different algorithms.

**Temporal graph learning.** Figure 2 is an illustration of a temporal graph, where each node has node feature $\mathbf{x}_i$, each node pair could have multiple temporal edges with different timestamps $t$ and edge features $\mathbf{e}_{ij}(t)$. We classify several chosen representative temporal graph learning methods into *memory-based* (e.g., JODIE), *GNN-based* (e.g., TGAT, TGSRec), *memory&GNN-based* (e.g., TGN, APAN, and PINT), *RNN-based* (e.g., CAW), and *GNN&RNN-based* (e.g., DySAT) methods.

$(v_1, v_2)$ at time $t_1, t_5$
$(v_3, v_5)$ at time $t_2$
$(v_2, v_4)$ at time $t_3, t_4$
$(v_4, v_5)$ at time $t_6$

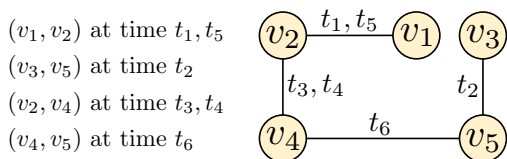

Figure 2: An illustration of temporal graph data with nodes $v_1, \ldots, v_5$ and timestamps $t_1, \ldots, t_6$ that indicate when two nodes interact.

For example, JODIE Kumar et al. (2019) maintains a memory block for each node and updates the memory block by an RNN upon the happening of each interaction; TGAT Xu et al. (2020a) first constructs the temporal computation graph, then recursively computes the hidden representation of each node using GNN; TGN Rossi et al. (2020) first uses memory blocks to capture all temporal interactions via JODIE then applies TGAT on the latest representation of the memory blocks of each node to capture the spatial information; CAW Wang et al. (2021c) proposes to first construct a set of sequential temporal events via random walks, then use RNN to aggregate the information from temporal events; DySAT Sankar et al. (2020) uses GNN to extract the spatial features and applies RNN on the output of GNN to capture the temporal dependencies.

## 3 GENERALIZATION OF TEMPORAL GRAPH METHODS

Firstly, we introduce the problem setting of theoretical analysis in Section 3.1. Then, we formally define the feature-label alignment (FLA) and derive the generalization bound in Section 3.2. Finally, we provide discussion on the generalization bound and its connection to FLA in Section 3.3.

### 3.1 PROBLEM SETTING FOR THEORETICAL ANALYSIS

For theoretical analysis, let us suppose the TGL models receives a sequence of data points $(X_1, y_1), ..., (X_N, y_N)$ that is the realization of some non-stationary process, where $X_i$ and $y_i$ stand for a single input data (e.g., a root node and its induced subgraph or random-walk path) and its label at the $i$-th iteration. During training, stochastic gradient descent (SGD) first uses $(X_1, y_1)$ and the initial model $f(\boldsymbol{\theta}_0)$ to generate the $\boldsymbol{\theta}_1$. Next, SGD uses the second example $(X_2, y_2)$ and the previously obtained model $f(\boldsymbol{\theta}_1)$ to generate $\boldsymbol{\theta}_2$, and so on. The training process is outlined in Algorithm 1. We aim to develop a unified theoretical framework to examine the generalization

---

**Algorithm 1** Using SGD for temporal graph learning

---

**Require:** Initialize $\boldsymbol{\theta}_0$ from Gaussian distribution $\mathcal{N}(0, 1/m)$ with $m$ as hidden dimension.
    **for** $i = 1, 2, \ldots, N$ **do**
        Sample an input data $(X_i, y_i)$ condition on all the previous sampled data $\mathcal{D}_1^{i-1}$
        Predict the label of $X_i$ as $f_i(\boldsymbol{\theta}_{i-1})$
        Update weight parameters by $\boldsymbol{\theta}_i = \boldsymbol{\theta}_{i-1} - \eta \nabla_\theta \, \text{loss}(f_i(\boldsymbol{\theta}_{i-1}), y_i)$.
    **end for**
**Ensure:** Return weight parameters by selecting $\widetilde{\boldsymbol{\theta}}$ randomly from $\{\boldsymbol{\theta}_0, \ldots, \boldsymbol{\theta}_{N-1}\}$.

---

capability of the three most fundamental types of TGL methods, including *GNN-based method*, *RNN-based method*, and *memory-based method* on a node-level binary classification task. Our goal is to upper bound the expected 0-1 error $\mathbb{E}[\text{loss}_N^{0-1}(\widetilde{\boldsymbol{\theta}})|\mathcal{D}_1^{N-1}]$ conditioned on a sequence of sampled data points $\mathcal{D}_1^{N-1}$, where $\text{loss}_i^{0-1}(\boldsymbol{\theta}) = \mathbf{1}\{y_i f_i(\boldsymbol{\theta}) < 0\}$ is the 0-1 loss computed by model $f(\boldsymbol{\theta})$ on data $(X_i, y_i)$ at $i$-th iteration, and $\mathcal{D}_1^{i-1} = \{(X_1, y_1), \ldots, (X_{i-1}, y_{i-1})\}$ is the set of all data points sampled before the $i$-th iteration. Our analysis of the fundamental TGL methods can pave the way to understand more advanced algorithms, such as *GNN&RNN-based* and *memory&GNN-based* methods.

**GNN-based method.** We compute the representation of node $v_i$ at time $t$ by applying GNN on the temporal graph that only considers the temporal edges with timestamp before time $t$. The GNN has $\boldsymbol{\theta} = \{\mathbf{W}^{(\ell)}\}_{\ell=1}^L$ as the parameters to optimize and $\alpha \in \{0, 1\}$ as a binary variable that controls whether a residual connection is used. The final prediction on node $v_i$ is computed as $f_i(\boldsymbol{\theta}) = \mathbf{W}^{(L)}\mathbf{h}_i^{(L-1)}$, where the hidden representation $\mathbf{h}_i^{(L-1)}$ is computed by

$$\mathbf{h}_i^{(\ell)} = \sigma\big(\mathbf{W}^{(\ell)} \sum\nolimits_{j \in \mathcal{N}(i)} P_{ij}\mathbf{h}_j^{(\ell-1)}\big) + \alpha\mathbf{h}_i^{(\ell-1)}, \quad \mathbf{h}_i^{(1)} = \sigma\big(\mathbf{W}^{(1)} \sum\nolimits_{j \in \mathcal{N}(i)} P_{ij}\mathbf{x}_j\big).$$

Here $\sigma(\cdot)$ is the activation function, $P_{ij}$ is the aggregation weight used for propagating information from node $v_j$ to node $v_i$, and $\mathcal{N}(i)$ is the set of all neighbors of node $v_i$ in the temporal graph. For simplicity, we assume the neighbor aggregation weights are fixed (e.g., $P_{ij} = 1/|\mathcal{N}(i)|$ for row normalized propagation) and consider the time-encoding vector as part of the node feature vector $\mathbf{x}_i$. For parameter dimension, we have $\mathbf{W}^{(1)} \in \mathbb{R}^{m \times d}$, $\mathbf{W}^{(\ell)} \in \mathbb{R}^{m \times m}$ for $2 \le \ell \le L-1$, and $\mathbf{W}^{(L)} \in \mathbb{R}^{1 \times m}$, where $m$ is the hidden dimension and $d$ is the input feature dimension. TGAT Xu et al. (2020a) can be thought of as GNN-based method but uses self-attention neighbor aggregation.

**RNN-based method.** We compute the representation of node $v_i$ at time $t$ by applying a multi-step RNN onto a sequence of temporal events $\{\mathbf{v}_1, \ldots, \mathbf{v}_{L-1}\}$ that are constructed at the target node, where each temporal event feature $\mathbf{v}_\ell$ is pre-computed on the temporal graph. We consider the time-encoding vector as part of event feature $\mathbf{v}_\ell$. The RNN has trainable parameters $\boldsymbol{\theta} = \{\mathbf{W}^{(1)}, \mathbf{W}^{(2)}, \mathbf{W}^{(3)}\}$ and $\alpha \in \{0, 1\}$ is a binary variable that controls whether a residual connection is used. Then, the final prediction on node $v_i$ is computed as $f_i(\boldsymbol{\theta}) = \mathbf{W}^{(3)}\mathbf{h}_{L-1}$, where the hidden representation $\mathbf{h}_{L-1}$ is recursively compute by

$$\mathbf{h}_\ell = \sigma\Big(\kappa\big(\mathbf{W}^{(1)}\mathbf{h}_{\ell-1} + \mathbf{W}^{(2)}\mathbf{x}_\ell\big)\Big) + \alpha\mathbf{h}_{\ell-1} \in \mathbb{R}^m,$$

$\sigma(\cdot)$ is the activation function, $\mathbf{h}_0 = \mathbf{0}_m$ is an all-zero vector, and $\mathbf{x}_\ell = \mathbf{W}^{(0)}\mathbf{v}_\ell$. We normalize each $\mathbf{h}_\ell$ by $\kappa = 1/\sqrt{2}$ so that $\|\mathbf{h}_\ell\|_2^2$ does not grow exponentially with $L$. We have $\mathbf{W}^{(1)}, \mathbf{W}^{(2)} \in \mathbb{R}^{m \times m}$, $\mathbf{W}^{(3)} \in \mathbb{R}^{1 \times m}$ as trainable parameters, but $\mathbf{W}^{(0)} \in \mathbb{R}^{m \times d}$ is non-trainable. CAW Wang et al. (2021c) is a special case of RNN-based method for edge classification tasks, where temporal events are sampled by temporal walks from both the source node and the destination node of an edge.

**Memory-based method.** We compute the representation of node $v_i$ at time $t$ by applying weight parameters on the memory block $\mathbf{s}_i(t)$. Let us define $\boldsymbol{\theta} = \{\mathbf{W}^{(1)}, \ldots, \mathbf{W}^{(4)}\}$ as the parameters to optimize. Then, the final prediction of node $v_i$ is computed by $f_i(\boldsymbol{\theta}) = \mathbf{W}^{(4)}\mathbf{s}_i(t)$ and $\mathbf{s}_i(t) \in \mathbb{R}^m$ is updated whenever node $v_i$ interacts with other nodes by

$$\mathbf{s}_i(t) = \sigma\Big(\kappa\big(\mathbf{W}^{(1)}\mathbf{s}_i^+(h_i^t) + \mathbf{W}^{(2)}\mathbf{s}_j^+(h_j^t) + \mathbf{W}^{(3)}\mathbf{e}_{ij}(t)\big)\Big),$$

where $\sigma(\cdot)$ is the activation function, $h_i^t$ is the latest timestamp that node $v_i$ interacts with other nodes before time $t$, $\mathbf{s}_i(0) = \mathbf{W}^{(0)}\mathbf{x}_i$, and $\mathbf{s}_i^+(t) = \text{StopGrad}(\mathbf{s}_i(t))$ is the memory block of node

$v_i$ at time $t$. We consider the time-encoding vector as part of edge feature $\mathbf{e}_{ij}(t)$. We normalize $\mathbf{s}_i(t)$ by $\kappa = 1/\sqrt{3}$ so that $\|\mathbf{s}_i(t)\|_2^2$ does not grow exponentially with time $t$. We have trainable parameters $\mathbf{W}^{(1)}, \mathbf{W}^{(2)} \in \mathbb{R}^{m \times m}, \mathbf{W}^{(3)} \in \mathbb{R}^{m \times d}, \mathbf{W}^{(4)} \in \mathbb{R}^m$ and fixed parameters $\mathbf{W}^{(0)} \in \mathbb{R}^{m \times d}$. JODIE Kumar et al. (2019) can be thought of as memory-based method but using the attention mechanism for final prediction.

## 3.2 Assumptions and main theoretical results

For the purpose of rigorous analysis, we make the following standard assumptions on the feature norms Cao & Gu (2019); Du et al. (2018), which could be satisfied via feature re-scaling.

**Assumption 1.** *All features has $\ell_2$-norm bounded by $1$, i.e., we assume $\|\mathbf{x}_i\|_2, \|\mathbf{e}_{ij}(t)\|_2, \|\mathbf{v}_\ell\| \leq 1$.*

In addition, we assume that the activation functions are Lipschitz continuous in TGL methods. The following assumption holds for common activation functions such as ReLU and LeakyReLU (which are often used in GNNs), Sigmoid and Tanh (which are often used in RNNs).

**Assumption 2.** *The activation function has Lipschitz constant $\rho \geq 1$.*

Furthermore, we make the following assumptions on the propagation matrix of GNN models, which are previously used in Liao et al. (2020); Cong et al. (2021). In practice, we know that $\tau = 1$ holds for row normalized and $\tau = \sqrt{\max_{i \in \mathcal{V}} d_i}$ holds for symmetrically normalized propagation matrix.

**Assumption 3.** *The row-wise sum is bounded by $\tau = \max_{i \in \mathcal{V}}(\sum_{j \in \mathcal{N}(i)} P_{ij})$ where $\tau \geq 1$.*

Finally, we adopt the following assumption regarding the non-stationary data generation process, which is standard assumption in time series prediction and online learning analysis Kuznetsov & Mohri (2015; 2016). As the data generation process transitions to a stationary state with an identical distribution at each step, this deviation $\Delta$ diminishes to zero.

**Assumption 4.** *We assume the discrepancy measure that quantifies the deviation between the intermediate steps (i.e., $i = 1, ..., N - 1$) and the final step (i.e., $i = N$) data distribution as*

$$\Delta = \sup_{f(\boldsymbol{\theta})} \Big| \frac{1}{N} \sum_{i=1}^N \mathbb{E}\left[loss_i^{0-1}(\boldsymbol{\theta}_{i-1}) | \mathcal{D}_1^{i-1}\right] - \frac{1}{N} \sum_{i=1}^N \mathbb{E}\left[loss_N^{0-1}(\boldsymbol{\theta}_{i-1}) | \mathcal{D}_1^{N-1}\right] \Big|,$$

*where the supremum is on any model, $loss_i^{0-1}(\boldsymbol{\theta}) = \mathbf{1}\{y_i f_i(\boldsymbol{\theta}) < 0\}$ is 0-1 loss, and $\mathcal{D}_1^{i-1} = \{(X_1, y_1), \ldots, (X_{i-1}, y_{i-1})\}$ is the sequence of data points before the $i$-th iteration.*

We introduce the feature-label alignment (FLA) score, which measures how well the representations produced by different algorithms align with the ground-truth labels.

**Definition 1.** *FLA is defined as $\mathbf{y}^\top (\mathbf{J}\mathbf{J}^\top)^{-1} \mathbf{y}$, where $\mathbf{J} = [vec(\nabla_\theta f_i(\boldsymbol{\theta}_0))]_{i=1}^N$ is the gradient of different temporal graph algorithms computed on each individual training example and $\boldsymbol{\theta}_0$ is the untrained weight parameters initialized from a Gaussian distribution.*

FLA has appeared in the convergence and generalization analysis of over-parameterized neural networks Arora et al. (2019). In practice, FLA quantifies the amount of perturbation we need on $\boldsymbol{\theta}$ along the direction of $\mathbf{J} = [vec(\nabla_\theta f_i(\boldsymbol{\theta}_0))]_{i=1}^N$ to minimize the logistic loss, which could be used to capture the expressiveness of different TGL algorithms, i.e., the smaller the perturbation, the better the expressiveness. Detailed discussion can be found in Appendix I.1. Computing FLA requires $\mathcal{O}(N^2|\boldsymbol{\theta}|)$ time complexity to compute $\mathbf{J}\mathbf{J}^\top$ and $\mathcal{O}(N^3)$ time complexity to compute matrix inverse, where $N$ is the number of training data and $|\boldsymbol{\theta}|$ is the number of weight parameters. In over-parameterized regime, we assume $|\boldsymbol{\theta}| > N$, and we can compute $\mathbf{J}\mathbf{J}^\top$ on a sampled subset of training data rather than the full training data to make sure this assumption holds.

The following theorem establishes the generalization capability of different TGL algorithms.

**Theorem 1.** *Given any $\delta \in (0, 1/e]$, FLA-related constant $R = \mathcal{O}(\sqrt{\mathbf{y}^\top (\mathbf{J}\mathbf{J}^\top)^{-1} \mathbf{y}})$, and number of training iterations $N$ (one training example per iteration), there exists $m^\star = \mathcal{O}(N^2/L^2) \log(1/\delta)$ such that, if hidden dimension $m \geq m^\star$ and using learning rate $\eta = \frac{R}{mC\sqrt{2N}}$, with probability at least $1 - \delta$ over the randomness of $\boldsymbol{\theta}_0$ initialization, we can upper bound the expected 0-1 error by*

$$\mathbb{E}[loss_N^{0-1}(\widetilde{\boldsymbol{\theta}}) \mid \mathcal{D}_1^{N-1}] \leq \mathcal{O}\Big(\frac{DRC}{\sqrt{N}}\Big) + \mathcal{O}\Big(\sqrt{\frac{\ln(1/\delta)}{N}}\Big) + \Delta,$$

where $\widetilde{\boldsymbol{\theta}}$ is uniformly sampled from $\{\boldsymbol{\theta}_0, \ldots, \boldsymbol{\theta}_{N-1}\}$, $R = \mathcal{O}(\sqrt{\mathbf{y}^\top (\mathbf{JJ}^\top)^{-1} \mathbf{y}})$, and the constant $C$ and $D$ of ① the L-layer GNN are $C = ((1 + 3\rho)\tau)^{L-1}$ and $D = L$, ② the L-step RNN are $C = (1 + 3\rho/\sqrt{2})^{L-1}$ and $D = L$, and ③ the memory-based method are $C = \rho$ and $D = 4$.

In Theorem 1, we show that the generalization error decreases as the number of training iterations $N$ increases, with a single data point being used for training at each iteration. On the other hand, the generalization error increases with respect to the number of layers/steps $L$ in GNN-/RNN-based methods, the feature-label alignment constant $R$, the maximum Lipschitz constant of the activation function $\rho$, and the graph convolution-related constant $\tau$. In the following section, we delve into more details on the effect of the number of layers/steps $L$ and the feature-label alignment constant $R$. The proofs are deferred to Appendices C, D, and E, respectively.

### 3.3 DISCUSSION ON THE GENERALIZATION BOUND: INSIGHTS AND LIMITATIONS

**Dependency on depth and steps.** The generalization error of GNN- and RNN-based methods tends to increase as the number of layers/steps $L$ increases. This partially explains why the hyper-parameter selection on $L$ is usually small for those methods. For example, *GNN-based method* TGAT uses 2-layer GNN (i.e., $L = 3$), *RNN-based method* CAW selects 3-steps RNN (i.e., $L = 4$), and *GNN&RNN-based method* DySAT uses 2-layer GNN and 3-steps RNN to achieve outstanding performance. On the other hand, *memory-based method* JODIE alleviates the dependency issue by using memory blocks and can leverage all historical interactions for prediction, which enjoys a generalization error independent of the number of steps. However, since gradients cannot flow through the memory blocks due to "stop gradient", its expressive power may be lower than that of other methods, which will be further elaborated when discussing the impact of feature-label alignment. *Memory&GNN-based methods* TGN and APAN alleviate the lack of expressive power issue by applying a single-layer GNN on top of the memory blocks.

**Dependency on feature-label alignment (FLA).** Although the dependency on the number of layers/steps of GNN/RNN can partially explain the performance disparity between these methods, it is still not clear if using "stop gradient" in the memory-based method and the selection of input data (e.g., using recent or uniformly sampled neighbors in a temporal graph) can affect the model performance. In the following, we take a closer look at the FLA score, which is inversely correlated to the generalization ability of the TGL models, i.e., the smaller the FLA, the better the generalization ability. According to Figure 3, we observe that ① JODIE (memory-based) has a relatively larger FLA score than most of the other TGL methods. This is potentially due to "stop gradient" operation that prevents gradients from flowing through the memory blocks and could potentially hurt the expressive power. ② APAN and TGN (memory&GNN-based) alleviate the expressive power degradation issue by applying a single layer GNN on top of the memory blocks, resulting in a smaller FLA than the pure memory-based method. ③ TGAT (GNN-based) has a relatively smaller FLA score than other methods, which is expected since GNN has achieved outstanding performance on static graphs. ④ DySAT (GNN&RNN-based) is originally designed for snapshot graphs, so its FLA score might be highly dependent on the selection of time-span size when converting a temporal graph to snapshot graphs. A non-optimal choice of time-span might cause information loss, which partially explains why DySAT's FLA is large. Additionally, the selection of the input data also affects the FLA. We will discuss further in the experimental section with empirical validation and detailed analysis.

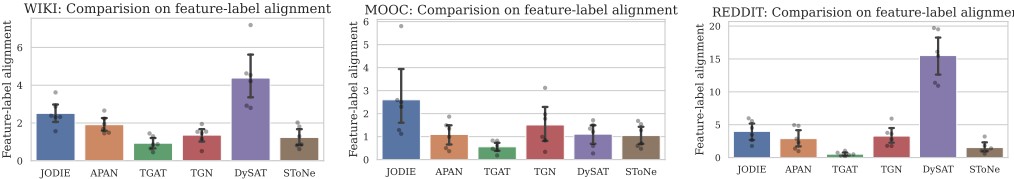

Figure 3: Comparison of FLA (y-axis) of different methods (x-axis) on real-world datasets.

## 4 ALGORITHM INSPIRED BY THEORETICAL INSIGHTS

Guided by our theoretical analysis, we introduce *Simplified-Temporal-Graph-Network* (SToNe) that not only enjoys a small generalization error but also empirically works well. The design of SToNe is guided by the following key insights that are presented in Section 3.3:

- *Shallow and non-recursive network.* This is because the generalization error increases with respect to the number of layers/steps in GNN-/RNN-based methods, which motivates us to consider a shallow and non-recursive neural architecture to alleviate such dependency.
- *Selecting proper input data instead of using memory blocks.* Although memory blocks could alleviate the dependency of generalization error on the number of layers/steps, it will also affect the FLA and hurt the generalization ability of the models. As an alternative, we propose to capture the important historical interactions by empirically selecting the proper input data.

To this end, we introduce the data preparation and the neural architecture in Section 4.1, then highlight the key features of SToNe that can differentiate itself from existing methods in Section 4.2.

## 4.1 SIMPLIFIED TEMPORAL GRAPH NETWORK: INPUT DATA AND NEURAL ARCHITECTURE

**Input data preparation.** To compute the representation of node $v_i$ at time $t$, we first identify the $K$ most recent nodes that have interacted with $v_i$ prior to time $t$ and denote them as temporal neighbors $\mathcal{N}_K^t(v_i)$. Then, we sort all nodes inside $\mathcal{N}_K^t(v_i)$ by the descending temporal order. If a node $v_j$ interacts with node $v_i$ multiple times, each interaction is treated as a separate temporal neighbor. For example in Figure 2, for any time $t > t_6$ and large enough constant $K$, we have $\mathcal{N}_K^t(v_4) = \{(v_5, t_6), (v_2, t_4), (v_2, t_3)\}$ in the descending temporal order. For each temporal neighbor $v_j \in \mathcal{N}_K(v_i, t)$, we represent its interaction with the target node $v_i$ at time $t'$ using a combination of edge features $\mathbf{e}_{ij}(t') \in \mathbb{R}^{d_e}$, time-encoding $\boldsymbol{\psi}(t - t') \in \mathbb{R}^{d_t}$, and node features $\mathbf{x}_i, \mathbf{x}_j \in \mathbb{R}^{d_n}$, which is denoted as $\mathbf{u}_{ij}(t') = [\mathbf{e}_{ij}(t') \| \boldsymbol{\psi}(t - t') \| \mathbf{x}_i \| \mathbf{x}_j]$. Then, we define $\mathbb{H}_i(t) = \{\mathbf{u}_{ij}(t') \mid (v_j, t') \in \mathcal{N}_K^t(v_i)\}$ as the set of all features that represent the temporal neighbors of node $v_i$ at time $t$, where all vectors in $\mathbb{H}_i(t)$ are ordered by the descending temporal order. For example, the interactions between the temporal neighbors of node $v_4$ at time $t > t_6$ to its root node in Figure 2 are $\mathbb{H}_4(t) = \{\mathbf{u}_{4,5}(t_6), \mathbf{u}_{4,2}(t_4), \mathbf{u}_{4,2}(t_3)\}$, where $\mathbf{u}_{4,5}(t_6) = [\mathbf{e}_{4,5}(t_6) \| \boldsymbol{\psi}(t - t_6) \| \mathbf{x}_4 \| \mathbf{x}_5]$, $\mathbf{u}_{4,2}(t_4) = [\mathbf{e}_{4,2}(t_4) \| \boldsymbol{\psi}(t - t_4) \| \mathbf{x}_4 \| \mathbf{x}_2]$, and $\mathbf{u}_{4,2}(t_3) = [\mathbf{e}_{4,2}(t_3) \| \boldsymbol{\psi}(t - t_3) \| \mathbf{x}_4 \| \mathbf{x}_2]$. The time-encoding function in SToNe is defined as $\boldsymbol{\psi}(t - t') = \cos((t - t')\mathbf{w})$, where $\mathbf{w} = [\alpha^{-(i-1)/\beta}]_{i=1}^{d_t}$ is a fixed $d_t$-dimensional vector and $\alpha = \beta = \sqrt{d_t}$. Notice that a similar time-encoding function is used in other temporal graph learning methods, e.g., Kumar et al. (2019); Xu et al. (2020a); Rossi et al. (2020); Wang et al. (2021c); Cong et al. (2023).

**Encoding features via GNN.** SToNe is a GNN-based method with trainable parameters $\boldsymbol{\theta} = \{\boldsymbol{\alpha}, \mathbf{W}^{(1)}, \mathbf{W}^{(2)}\}$, where $\boldsymbol{\alpha} \in \mathbb{R}^K$, $\mathbf{W}^{(1)} \in \mathbb{R}^{d_{\text{hid}} \times d_{\text{in}}}$, and $\mathbf{W}^{(2)} \in \mathbb{R}^{d_{\text{out}} \times d_{\text{hid}}}$. Here $K$ is the maximum temporal neighbor size we consider, $d_{\text{in}} = d_e + d_t + 2d_n$, and $d_{\text{hid}}, d_{\text{out}}$ are the dimensions of hidden and output representations. The representation of node $v_i$ at time $t$ is computed by

$$\mathbf{h}_i(t) = \mathbf{W}^{(2)} \text{LayerNorm}(\mathbf{z}_i(t)), \mathbf{z}_i(t) = \sigma\Big(\sum_{k=1}^{|\mathbb{H}_i(t)|} \alpha_k \mathbf{W}^{(1)}[\mathbb{H}_i(t)]_k\Big) + \sum_{k=1}^{|\mathbb{H}_i(t)|} [\mathbb{H}_i(t)]_k, \tag{1}$$

If the temporal neighbor size of node $v_i$ is less than $K$, i.e., $|\mathbb{H}_i(t)| < K$, we only need to update the first $|\mathbb{H}_i(t)|$ entries of the vector $\boldsymbol{\alpha}$. Notice that the number of parameters in SToNe is $(K + d_{\text{hid}} d_{\text{in}} + d_{\text{hid}} d_{\text{out}})$, which is usually fewer than other TGL algorithms given the same hidden dimension size. We will report the number of parameters and its computational cost in the experiment section.

**Link prediction via MLP.** When the downstream task is future link prediction, we predict whether an interaction between nodes $v_i, v_j$ occurs at time $t$ by applying a 2-layer MLP model on $[\mathbf{h}_i(t) \| \mathbf{h}_j(t)] \in \mathbb{R}^{2d_{\text{out}}}$. It is worth noting that the same link classifier is used in almost all the existing temporal graph learning methods Kumar et al. (2019); Xu et al. (2020a); Rossi et al. (2020); Wang et al. (2021c); Souza et al. (2022); Zhou et al. (2022); Cong et al. (2023), including SToNe.

## 4.2 COMPARISON TO EXISTING METHODS

**Comparison to TGAT.** GNN-based method TGAT uses 2-hop uniformly sampled neighbors and aggregates the information using a 2-layer GAT Veličković et al. (2017). The neighbor aggregation weights in TGAT are estimated by self-attention. In contrast, SToNe uses 1-hop most recent neighbors and directly learns the neighbor aggregation weights $\boldsymbol{\alpha}$ as shown in Eq. 1. Moreover, self-attention in TGAT can be thought of as weighted average of neighbor information, while SToNe can be thought of as sum aggregation, which can better distinguish different temporal neighbors, and it is especially helpful when node and edge features are lacking.

**Comparison to TGN.** Memory&GNN-based method TGN uses 1-hop most recent temporal neighbors and applies a self-attention module to the sampled temporal neighbors' features that are stored inside the memory blocks. In fact, SToNe can be thought of as a special case of TGN, where we use the features in $\mathbb{H}_i(t)$ instead of the memory blocks and directly learn the neighbor aggregation weight $\alpha$ instead of using the self-attention aggregation as shown in Eq. 1.

**Comparison to GraphMixer.** SToNe could be think of as a simplified version of Cong et al. (2023) that addresses the high computation cost associated with the MLP-mixer used for temporal aggregation Tolstikhin et al. (2021). Instead of relying on the MLP-mixer, we introduce the aggregation vector $\alpha$ and employ linear functions parameterized by $\mathbf{W}^{(1)}$ and $\mathbf{W}^{(2)}$ for aggregation. Additionally, we do not explicitly model the graph structure through Cong et al. (2023)'s node-encoder. Instead, we implicitly capture the node features within the temporal interactions present in $\mathbb{H}(t)$. Our experiments demonstrate that SToNe significantly reduces the model complexity (Table 2) while achieving comparable performance (Figure 4 and Table 1).

**Temporal graph construction.** Most of the TGL methods Kumar et al. (2019); Xu et al. (2020a); Rossi et al. (2020); Sankar et al. (2020); Wang et al. (2021c) implements temporal graphs as directed graph data structure with information only flowing from source to destination nodes. However, we consider the temporal graph as an bi-directed graph data structure by assuming that information also flow from destination to source nodes for SToNe. This means that the "most recent 1-hop neighbors" sampled for the two nodes on the "bi-directed" temporal graph should be similar if two nodes are frequently connected in recent timestamps. Such similarity provides information on whether two nodes are frequently connected in recent timestamps, which is essential for temporal graph link prediction Cong et al. (2023). For example, let us assume nodes $v_i, v_j$ interacts at time $t_1, t_2$ in the temporal order. Then, given any timestamp $t > t_2$ and a large enough $K$, the 1-hop temporal neighbors on the bi-directed graph is $\mathcal{N}_K^t(v_i) = \{(v_j, t_1), (v_j, t_2)\}$ and $\mathcal{N}_K^t(v_j) = \{(v_i, t_1), (v_i, t_2)\}$, while on directed graph is $\mathcal{N}_K^t(v_i) = \{(v_j, t_1), (v_j, t_2)\}$ but $\mathcal{N}_K^t(v_j) = \emptyset$. Intuitively, if two nodes are frequently connected in recent timestamps, they are also likely to be connected in the near future. In the experiment section, we show that changing the temporal graph from bi-directed to directed can negatively impact the feature-label alignment and model performance.

## 5    EXPERIMENTS

We compare SToNe with several TGL algorithms under the transductive learning setting. We conduct experiments on 6 real-world datasets, i.e., Reddit, Wiki, MOOC, LastFM, GDELT, and UCI. Similar to many existing works, we also use the $70\%/15\%/15\%$ chronological splits for the train/validation/test sets. We re-run the official released implementations on benchmark datasets and repeat 6 times with different random seeds. Please refer to Appendix A for experiment setup details.

### 5.1    EXPERIMENT RESULTS

**Comparison on average precision.** We compare the average precision score with baseline methods in Table 1. We observe that SToNe could achieve compatible or even better performance on most datasets. In particular, the performance of SToNe outperforms most of the baselines with a large margin on the LastFM and UCI dataset. This is potentially because these two datasets lack node/edge features and have larger average time-gap (see dataset statistics in Table 3). Since baseline methods rely on RNN or self-attention to process the node/edge features, they implicitly assume that the features exists and are "smooth" at adjacent timestamps, which could generalize poorly when the assumptions are violated. PINT addresses this issue by pre-processing the dataset and generating its own positional encoding as the augment features, therefore it achieves better performance than SToNe on UCI dataset. However, computing this positional encoding is time-consuming and does not always perform well on other datasets. For example, PINT requires more than 400 hours to compute the positional features on GDELT, which is infeasible. Additionally, the performance of SToNe closely matches that of GraphMixer, but with significantly lower model complexity, which will be demonstrated in Table 2. SToNe exhibits improved and more consistent performance on UCI dataset, because SToNe is less susceptible to overfitting on small dataset.

**Comparison on generalization performance.** To validate the generalization performance of SToNe, we compare the average precision score and the generalization gap in Figure 4. The general-

Table 1: Comparison on the average precision score for link prediction. We highlight the best score in red, the second best score in blue, and the third best in green.

|  | Reddit | Wiki | MOOC | LastFM | GDELT | UCI |
|---|---|---|---|---|---|---|
| JODIE | 99.75 ± 0.02 | 98.94 ± 0.06 | 98.99 ± 0.04 | 79.41 ± 3.68 | 98.48 ± 0.01 | 73.93 ± 2.77 |
| TGAT | 99.56 ± 0.04 | 98.69 ± 0.10 | 99.28 ± 0.07 | 75.16 ± 0.10 | 96.46 ± 0.04 | 82.49 ± 0.67 |
| TGSRec | 95.21 ± 0.08 | 91.64 ± 0.12 | 83.62 ± 0.34 | 76.91 ± 0.87 | 97.03 ± 0.61 | 76.64 ± 0.54 |
| TGN | 98.83 ± 0.01 | 99.61 ± 0.05 | 99.63 ± 0.06 | 91.04 ± 2.18 | 98.33 ± 0.09 | 79.91 ± 1.59 |
| APAN | 99.36 ± 0.17 | 98.99 ± 0.14 | 99.02 ± 0.11 | 73.18 ± 7.72 | 98.01 ± 0.26 | 61.73 ± 5.96 |
| PINT | 99.03 ± 0.01 | 98.78 ± 0.10 | 87.24 ± 0.73 | 88.06 ± 0.71 | - | 96.22 ± 0.08 |
| CAW-attn | 98.51 ± 0.02 | 97.95 ± 0.03 | 63.07 ± 0.82 | 76.31 ± 0.10 | 95.06 ± 0.11 | 92.16 ± 0.19 |
| DySAT | 98.55 ± 0.01 | 96.64 ± 0.03 | 98.76 ± 0.11 | 76.28 ± 0.04 | 98.17 ± 0.01 | 80.43 ± 0.36 |
| GraphMixer | 99.93 ± 0.01 | 99.85 ± 0.01 | 99.91 ± 0.01 | 96.31 ± 0.02 | 98.89 ± 0.02 | 92.39 ± 2.15 |
| SToNe | 99.89 ± 0.00 | 99.85 ± 0.05 | 99.88 ± 0.04 | 95.74 ± 0.13 | 99.11 ± 0.03 | 94.60 ± 0.31 |

ization gap is defined as the absolute difference between the training and validation average precision scores. Our results show that SToNe, similar to GraphMixer, consistently achieves a higher average precision score faster than other baselines, has a smaller generalization gap, and has relatively stable performance across different epochs. This suggests that SToNe has better generalization performance compared to the baselines. In particular on the UCI dataset, SToNe is less prone to overfit and its generalization gap increases more slowly than all other baseline methods.

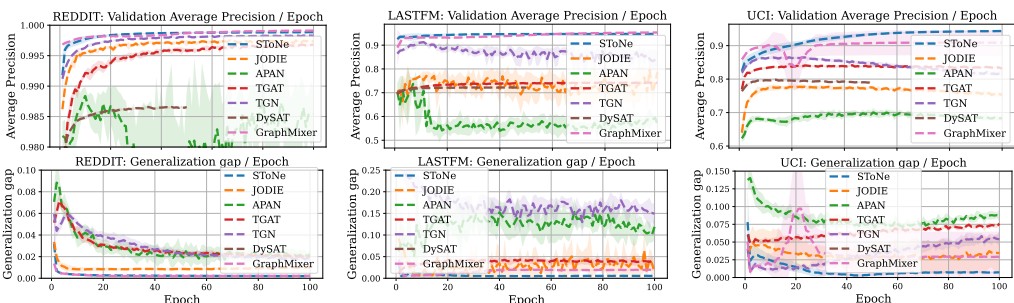

Figure 4: Comparison of the average prevision of validation set and generalization gap of different methods on real-world datasets.

**Comparison on model complexity.** We compare the number of parameters (including trainable weight parameters and memory block size) and wall-clock time per epoch during the training phase in Table 2. Our results show that SToNe has fewer parameters than all the baselines, and its computation time is also faster than most baseline methods. Note that some baselines also require significant time for data preprocessing, which is not included in Table 2. For example, PINT takes more than 84 hours to pre-compute the positional encoding on the Reddit dataset, which is significantly longer than its per epoch training time. In contrast, SToNe does not require computing augmented features based on the temporal graph and therefore does not have this pre-processing time.

Table 2: Comparison of the number of parameters ($\times 10^5$ parameters) and wall-clock time (second) of single epoch of training in the format "*Number of parameters (Wall-clock time)*".

|  | JODIE | TGAT | TGSRec | TGN | APAN | PINT | CAW | DySAT | GraphMixer | SToNe |
|---|---|---|---|---|---|---|---|---|---|---|
| Reddit | 11.6 (5s) | 2.1 (15s) | 5.1 (538s) | 14.1 (8s) | 12.3 (13s) | 17.1 (436s) | 43.3 (1,930s) | 4.3 (33s) | 23.3 (12s) | 0.58 (8s) |
| Wiki | 9.8 (2s) | 2.1 (4s) | 4.8 (157s) | 12.3 (2s) | 10.6 (4s) | 17.9 (93s) | 43.3 (282s) | 4.3 (6s) | 19.8 (3s) | 0.58 (2s) |
| MOOC | 7.8 (4s) | 1.4 (8s) | 3.9 (656s) | 10.0 (5s) | 8.4 (9s) | 10.1 (157s) | 43.3 (653s) | 2.3 (16d) | 15.3 (7s) | 0.41 (5s) |
| LastFM | 2.6 (11s) | 1.4 (28s) | 2.3 (1,810s) | 4.8 (15s) | 3.2 (28s) | 4.9 (440s) | 43.3 (1,832s) | 2.3 (41s) | 5.2 (21s) | 0.41 (16s) |

**More experiment results.** Due to the space limit, we defer more experiment results to Appendix B, such as performance evaluation under different metrics (e.g., AUC, RecallK, and MRR), ablation study on the effect of neighbor selection (e.g., recent vs randomly sampled neighbors), etc.

## 6 CONCLUSION

In this paper, we study the generalization ability of various TGL algorithms. We reveal the relationship between the generalization error to "*the number of layers/steps*" and "*the feature-label alignment*". Guided by our analysis, we propose SToNe. Extensive experiments are conducted on real-world datasets to show that SToNe enjoys a smaller generalization error, better performance, and lower complexity. These results provide a deeper insight into TGL from a theoretical perspective and are beneficial to design practically efficacious and simpler TGL algorithms for future studies.

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

## A EXPERIMENT SETUP DETAILS

### A.1 HARDWARE SPECIFICATION AND ENVIRONMENT

Our experiments are conducted on a single machine with an Intel i9-10850K processor, an Nvidia RTX 3090 GPU, and 64GB of RAM. The experiments are implemented in Python 3.8 using PyTorch 1.12.1 on CUDA 11.6 and the temporal graph learning framework Zhou et al. (2022). To ensure a fair comparison, all experiments are repeated 6 times with random seeds $\{0, 1, 2, 3, 4, 5\}$.

### A.2 DETAILS ON DATASETS

We conduct experiments on the following 6 datasets, where the detailed dataset statistics are summarized in Table 3. The download links for all datasets can be found in the code repository.

- Reddit dataset consists of one month of posts made by users on subreddits. The link feature is extracted by converting the text of each post into a feature vector. Reddit dataset has been previously used in existing works such as Kumar et al. (2019); Xu et al. (2020a); Rossi et al. (2020); Souza et al. (2022); Wang et al. (2021c); Zhou et al. (2022).

- Wiki dataset consists of one month of edits made by edits on Wikipedia pages. The link feature is extracted by converting the edit test into an LIWC-feature vector. Wiki dataset has been previously used in existing works such as Kumar et al. (2019); Xu et al. (2020a); Rossi et al. (2020); Souza et al. (2022); Wang et al. (2021c); Zhou et al. (2022).

- LastFM dataset consists of one month of who-listen-to-which song information. LastFM dataset has been previously used in existing works such as Kumar et al. (2019); Souza et al. (2022); Zhou et al. (2022).

- MOOC dataset consists of actions done by students on a MOOC online course. MOOC dataset has been previously used in existing works such as Kumar et al. (2019); Souza et al. (2022); Zhou et al. (2022).

- GDELT dataset is a temporal knowledge graph dataset that originates from the Event Database. The Event Database records events from around the world as reported in news articles. It has been previously used in works such as Zhou et al. (2022); Cong et al. (2023). We follow Cong et al. (2023) to subsample 1 temporal link per 100 continuous temporal link because the original dataset is too big to fit into CPU RAM memory for single-machine training.

- UCI dataset is a publicly available communication network dataset that includes email interactions between core employees and messages sent between peer users on an online social network platform. It has been previously used in works such as Sankar et al. (2020); Souza et al. (2022).

Table 3: Dataset statistic.

|  | $|\mathcal{V}|$ | $|\mathcal{E}|$ | Avg time-gap | $\dim(\mathbf{x}_i^{\text{node}})$ | $\dim(\mathbf{x}_{ij}^{\text{link}})$ | Node | Link | Time |
|---|---|---|---|---|---|---|---|---|
| Reddit | 10,984 | 672,447 | 4 | 0 | 172 | ✗ | ✓ | ✓ |
| Wiki | 9,227 | 157,474 | 17 | 0 | 172 | ✗ | ✓ | ✓ |
| MOOC | 7,144 | 411,749 | 3.6 | 0 | 0 | ✗ | ✗ | ✓ |
| LastFM | 1,980 | 1,293,103 | 106 | 0 | 0 | ✗ | ✗ | ✓ |
| GDELT | 8,831 | 1,912,909 | 0.1 | 413 | 186 | ✓ | ✓ | ✓ |
| UCI | 1,900 | 59,835 | 279 | 0 | 0 | ✗ | ✓ | ✓ |

### A.3 DETAILS ON BASELINE IMPLEMENTATIONS

The implementations of JODIE, DySAT, TGAT, TGN, and APAN are obtained from the TGL framework Zhou et al. (2022) at TGL-code[1]. This framework's implementation has been found to achieve better overall scores than the original implementations of these baselines.

The implementation of CAWs-attn is obtained from their official implementation at CAW-code[2].

The implementation of TGSRec is obtained from TGSRec-code[3].

The implementation of PINT is obtained from PINT-code[4].

The implementation of GraphMixer is obtained from GraphMixer-code[5].

We follow their instructions that are provided in the code repository for hyper-parameter selection. We directly test with their official implementations by changing our data structure to their required structure and use their default hyper-parameters.

### A.4 DETAILS ON STONE IMPLEMENTATIONS

We implement the proposed method SToNe under the TGL framework Zhou et al. (2022) and use the same hyper-parameters on all datasets (e.g., learning rate 0.0001, weight decay $10^{-6}$, batch size 600, hidden dimension 100) to ensure a fair comparison. The models are trained until the validation error did not decrease for 20 epochs. One of the most important dataset-dependent hyper-parameter is the number of temporal graph neighbors $K$. The selection of $K$ can be found in our repository.

---

[1] https://github.com/amazon-research/tgl
[2] https://github.com/snap-stanford/CAW
[3] https://github.com/DyGRec/TGSRec
[4] https://github.com/AaltoPML/PINT
[5] https://github.com/CongWeilin/GraphMixer

# B MORE EXPERIMENT RESULTS

## B.1 TRANSDUCTIVE LEARNING WITH AUC AS EVALUATION METRIC

AUC (Under the ROC Curve) is one of the most widely accepted evaluation metrics for link prediction, which has been used in many existing works Xu et al. (2020a); Rossi et al. (2020). In the following, we compare the AUC score of SToNe with baselines in Table 4. We observe that SToNe performs better than the baselines on most datasets. While our performance is slightly lower than PINT on the UCI dataset, PINT requires a significant amount of time to pre-compute the positional features as augmented features (about 3 hours for the UCI dataset) and the use of positional encoding in PINT does not always perform well on other datasets. Besides, we do not report the results of PINT on the GDELT dataset because it requires more than 400 hours to compute the positional features for training. Additionally, the performance of SToNe closely matches that of GraphMixer, but with significantly lower model complexity, which will be demonstrated in Table 2. SToNe exhibits improved and more consistent performance compared on UCI dataset, which is attributed to its small dataset size and SToNe is less susceptible to overfitting on small dataset. Please also refer to our discussion on computational time and average precision score in Section 5.1 for more details.

Table 4: Comparison on the AUC score for link prediction. We highlight the best score in red, the second best score in blue, and the third best in green.

|  | Reddit | Wiki | MOOC | LastFM | GDELT | UCI |
|---|---|---|---|---|---|---|
| JODIE | $99.78 \pm 0.01$ | $99.10 \pm 0.05$ | $99.46 \pm 0.07$ | $80.48 \pm 5.84$ | $98.66 \pm 0.01$ | $76.15 \pm 3.28$ |
| TGAT | $99.59 \pm 0.04$ | $98.78 \pm 0.08$ | $99.53 \pm 0.03$ | $78.49 \pm 0.09$ | $97.21 \pm 0.02$ | $83.30 \pm 0.40$ |
| TGSRec | $94.74 \pm 0.20$ | $91.32 \pm 0.19$ | $80.70 \pm 2.31$ | $76.66 \pm 1.54$ | $96.72 \pm 0.42$ | $81.00 \pm 0.58$ |
| TGN | $98.86 \pm 0.01$ | $99.62 \pm 0.05$ | $99.77 \pm 0.04$ | $92.54 \pm 2.24$ | $98.55 \pm 0.05$ | $75.32 \pm 2.34$ |
| APAN | $99.50 \pm 0.11$ | $99.12 \pm 0.11$ | $99.43 \pm 0.08$ | $78.15 \pm 6.99$ | $98.39 \pm 0.15$ | $64.39 \pm 7.42$ |
| PINT | $99.03 \pm 0.01$ | $98.78 \pm 0.10$ | $87.24 \pm 0.73$ | $88.06 \pm 0.71$ | - | $94.22 \pm 0.58$ |
| CAW-attn | $98.30 \pm 0.01$ | $97.89 \pm 0.02$ | $63.95 \pm 0.81$ | $72.93 \pm 0.54$ | $95.13 \pm 0.11$ | $92.10 \pm 0.10$ |
| DySAT | $98.43 \pm 0.01$ | $96.87 \pm 0.03$ | $99.25 \pm 0.05$ | $73.93 \pm 0.08$ | $98.38 \pm 0.01$ | $78.70 \pm 0.05$ |
| GraphMixer | $99.94 \pm 0.01$ | $99.82 \pm 0.01$ | $99.93 \pm 0.01$ | $97.38 \pm 0.02$ | $98.89 \pm 0.02$ | $91.74 \pm 2.69$ |
| SToNe | $99.91 \pm 0.00$ | $99.85 \pm 0.05$ | $99.91 \pm 0.02$ | $97.16 \pm 0.06$ | $99.26 \pm 0.02$ | $93.14 \pm 0.67$ |

## B.2 ABLATION STUDY ON THE EFFECT OF NEIGHBOR SELECTION

We study the effect of model input selection on feature-label alignment and average precision scores. As shown in Figure 5, changing the default setting of "recent 1-hop neighbors" in SToNe to either "recent 2-hop neighbors" or "random 1-hop neighbor" increases feature-label alignment score, which has a inverse correlation to the generalization ability, and decreases average precision scores.

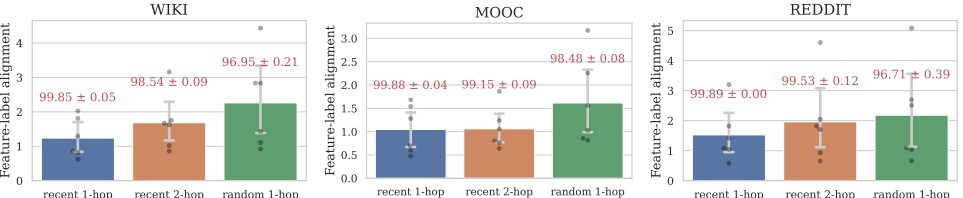

Figure 5: Comparison of feature-label alignment (y-axis) and average precision score (in red text) of different model input selection (x-axis) on real-world datasets.

## B.3 DIRECTED VERSUS BI-DIRECTED GRAPH.

We study the impact of using directed/bi-directed graph data structure on the feature-label alignment (FLA) and average precision score. As shown in Figure 6, changing from a bi-directed to a directed graph will increase the FLA score and lead to a significant decrease in model performance. This is because "the recent 1-hop neighbors" sampled for the two nodes on the bi-directed graph could provide information on whether two nodes are frequently connected in the last few timestamps, which is essential for temporal graph link prediction. However, such type of information is missing if using directed graph as discussed in the last paragraph of Section 4.2.

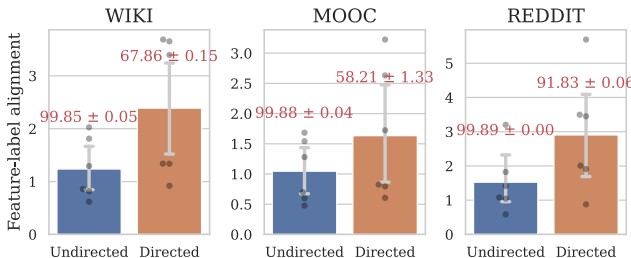

Figure 6: Compare the feature-label alignment (y-axis) and average precision (red text) with directed/undirected temporal graph.

Table 5: Comparison on the Recall@1, Recall@5 and, MRR. For exch row, the best score is in red text, second best score is in blue text, and the first best score is in green.

|  |  | JODIE | DySAT | TGAT | TGN | GraphMixer | SToNe |
|---|---|---|---|---|---|---|---|
| Reddit | Recall@1 | $0.8773 \pm 0.0121$ | $0.8454 \pm 0.0007$ | $0.8428 \pm 0.0130$ | $0.9294 \pm 0.0028$ | $0.9999 \pm 0.0000$ | $0.9998 \pm 0.0001$ |
|  | Recall@5 | $0.9712 \pm 0.0012$ | $0.9362 \pm 0.0006$ | $0.9531 \pm 0.0077$ | $0.9823 \pm 0.0012$ | $1.0000 \pm 0.0000$ | $1.0000 \pm 0.0000$ |
|  | MRR | $0.8425 \pm 0.0152$ | $0.8153 \pm 0.0016$ | $0.8219 \pm 0.0116$ | $0.9072 \pm 0.0040$ | $0.9965 \pm 0.0000$ | $0.9958 \pm 0.0002$ |
| Wiki | Recall@1 | $0.8347 \pm 0.0051$ | $0.7689 \pm 0.0043$ | $0.7139 \pm 0.0150$ | $0.8526 \pm 0.0050$ | $0.9954 \pm 0.0002$ | $0.9955 \pm 0.0002$ |
|  | Recall@5 | $0.9286 \pm 0.0033$ | $0.8763 \pm 0.0080$ | $0.8915 \pm 0.0090$ | $0.9293 \pm 0.0026$ | $0.9971 \pm 0.0002$ | $0.9980 \pm 0.0001$ |
|  | MRR | $0.8145 \pm 0.0048$ | $0.7450 \pm 0.0084$ | $0.6999 \pm 0.0128$ | $0.8343 \pm 0.0063$ | $0.9875 \pm 0.0004$ | $0.9827 \pm 0.0005$ |
| MOOC | Recall@1 | $0.8637 \pm 0.0130$ | $0.8337 \pm 0.0059$ | $0.9066 \pm 0.0072$ | $0.9314 \pm 0.0175$ | $0.9994 \pm 0.0001$ | $0.9994 \pm 0.0001$ |
|  | Recall@5 | $0.9981 \pm 0.0002$ | $0.9981 \pm 0.0002$ | $0.9927 \pm 0.0013$ | $0.9932 \pm 0.0009$ | $0.9999 \pm 0.0001$ | $0.9999 \pm 0.0000$ |
|  | MRR | $0.7675 \pm 0.0151$ | $0.7451 \pm 0.0058$ | $0.8234 \pm 0.0083$ | $0.8602 \pm 0.0263$ | $0.9911 \pm 0.0004$ | $0.9910 \pm 0.0006$ |
| LastFM | Recall@1 | $0.1684 \pm 0.0589$ | $0.2773 \pm 0.0035$ | $0.1770 \pm 0.0041$ | $0.2746 \pm 0.1279$ | $0.9940 \pm 0.0013$ | $0.9951 \pm 0.0002$ |
|  | Recall@5 | $0.2982 \pm 0.0766$ | $0.3886 \pm 0.0050$ | $0.2617 \pm 0.0043$ | $0.4312 \pm 0.2001$ | $0.9997 \pm 0.0002$ | $1.0000 \pm 0.0000$ |
|  | MRR | $0.2071 \pm 0.0536$ | $0.2892 \pm 0.0074$ | $0.1998 \pm 0.0042$ | $0.2995 \pm 0.1278$ | $0.9620 \pm 0.0015$ | $0.9592 \pm 0.0011$ |
| GDELT | Recall@1 | $0.7533 \pm 0.0053$ | $0.3503 \pm 0.0138$ | $0.0182 \pm 0.0053$ | $0.7595 \pm 0.0047$ | $0.9115 \pm 0.0028$ | $0.9117 \pm 0.0027$ |
|  | Recall@5 | $0.9333 \pm 0.0011$ | $0.8239 \pm 0.0068$ | $0.0576 \pm 0.0134$ | $0.9318 \pm 0.0023$ | $0.9891 \pm 0.0011$ | $0.9899 \pm 0.0009$ |
|  | MRR | $0.7108 \pm 0.0076$ | $0.4146 \pm 0.0103$ | $0.0868 \pm 0.0061$ | $0.7302 \pm 0.0044$ | $0.8622 \pm 0.0037$ | $0.8627 \pm 0.0032$ |
| UCI | Recall@1 | $0.5736 \pm 0.0068$ | $0.6039 \pm 0.0878$ | $0.1872 \pm 0.0292$ | $0.4732 \pm 0.0434$ | $0.7989 \pm 0.1119$ | $0.8995 \pm 0.0069$ |
|  | Recall@5 | $0.7814 \pm 0.0022$ | $0.6682 \pm 0.0889$ | $0.3036 \pm 0.0339$ | $0.6384 \pm 0.0396$ | $0.8571 \pm 0.0771$ | $0.9435 \pm 0.0034$ |
|  | MRR | $0.5572 \pm 0.0087$ | $0.5656 \pm 0.0816$ | $0.2296 \pm 0.0253$ | $0.4799 \pm 0.0354$ | $0.7677 \pm 0.1093$ | $0.8628 \pm 0.0087$ |

### B.4 TRANSDUCTIVE LEARNING WITH RECALL@K AND MRR AS EVALUATION METRICS

Recall@K and MRR (mean reciprocal rank) are popular evaluation metrics commonly used in the real-world recommendation system. Higher values indicate better model performance. Our implementation of Recall@K and MRR follows the implementation in Hu et al. (2020); Cong et al. (2023): we first sample 100 negative destination nodes for each source node of a temporal link node pair, then our goal is to rank the positive temporal link node pairs higher than 100 negative destination nodes. As shown in Table 5, SToNe and GraphMixer has higher Recall@K and MRR scores than other baselines, indicating that SToNe has more confidence in the estimated categories. This might because SToNe has better generalization ability and higher confidence in its predictions when using the knowledge learned from the training set.

### B.5 ABLATION STUDY ON THE EFFECT OF USING SELF-ATTENTION IN STONE

In this section, we explore the effect of replacing the aggregation weight $\alpha$ with the self-attention aggregation Shi et al. (2020) implemented by PyTorch Geometric Fey & Lenssen (2019), we name it as "*SToNe (self-attention)*".

First of all, we compare the average precision score and AUC score of "*SToNe*" and "*SToNe (self-attention)*" in Table 6. We observe that "*SToNe*" could achieve superior performance than "*SToNe (self-attention)*" on all datasets. In particular, using self-attention with SToNe results in a larger variance on LastFM and UCI datasets. This is potentially because these two datasets lack node/edge features and have larger average time-gap (dataset statistic in Table 3). Since "*SToNe (self-attention)*" is relying on self-attention to process the node/edge features, it implicitly assumes node features exist and are "smooth" at adjacent timestamps. As a result, "*SToNe (self-attention)*" could generalize poorly when the assumptions are violated.

Moreover, to validate the generalization performance, we compare the average precision score and the generalization gap in Figure 7. The generalization gap is defined as the absolute difference be-

Table 6: Comparison on the Average precision and AUC score for link prediction.

|  |  | Reddit | Wiki | MOOC | LastFM | GDELT | UCI |
|---|---|---|---|---|---|---|---|
| Average Precision | SToNe (self-attention) | $99.37 \pm 0.01$ | $98.99 \pm 0.01$ | $99.42 \pm 0.01$ | $91.79 \pm 4.02$ | $96.20 \pm 0.02$ | $90.24 \pm 0.73$ |
|  | SToNe | $\mathbf{99.89} \pm 0.00$ | $\mathbf{99.85} \pm 0.05$ | $\mathbf{99.88} \pm 0.04$ | $\mathbf{95.74} \pm 0.13$ | $\mathbf{97.40} \pm 0.02$ | $\mathbf{94.60} \pm 0.31$ |
| AUC Score | SToNe (self-attention) | $99.37 \pm 0.01$ | $98.92 \pm 0.02$ | $99.58 \pm 0.01$ | $93.33 \pm 4.23$ | $95.56 \pm 0.01$ | $88.22 \pm 0.98$ |
|  | SToNe | $\mathbf{99.91} \pm 0.00$ | $\mathbf{99.85} \pm 0.05$ | $\mathbf{99.91} \pm 0.02$ | $\mathbf{97.16} \pm 0.06$ | $\mathbf{96.96} \pm 0.01$ | $\mathbf{93.14} \pm 0.67$ |

tween the training and validation average precision scores. Our results show that "*SToNe*" could achieve a higher average precision score than "*SToNe (self-attention)*" and has a smaller generalization gap. This suggests that using aggregation weight $\alpha$ instead of using self-attention in SToNe could lead to better generalization performance, which is expected because self-attention has higher model complexity and could be harder to train.

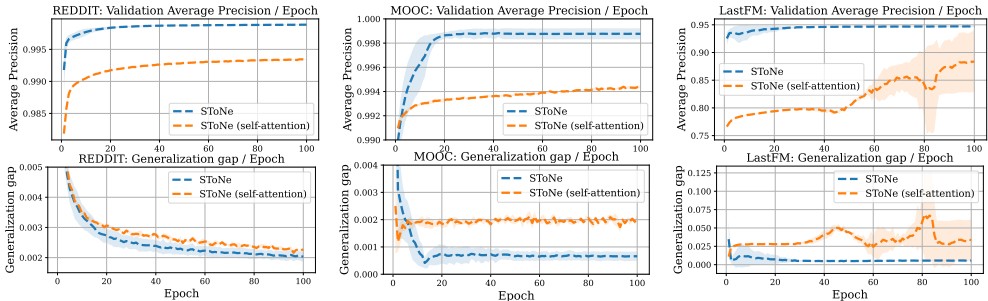

Figure 7: Comparison of the average prevision of the validation set and the generalization gap of different methods on real-world datasets.

### B.6 CONDUCT EXPERIMENTS UNDER THE INDUCTIVE LEARNING SETTING

Please notice that the discussion in our paper is mainly focusing on transductive learning. For the completeness, we also report some preliminary inductive learning results. Our inductive learning setting is the same as Xu et al. (2020a). More specifically, the inductive node sets are all nodes that does not belonging to the training set nodes, i.e., $\mathcal{V}_{\text{inductive}} = \mathcal{V} \setminus \mathcal{V}_{\text{train}}$. In the inductive setting, negative nodes are only selected from $\mathcal{V}_{\text{inductive}}$. A test set link is considered as the inductive link if at least one of its associated nodes belong to $\mathcal{V}_{\text{inductive}}$, i.e., $\mathcal{E}_{\text{inductive}} = \{(v_i, v_j) \mid (v_i \in \mathcal{V}_{\text{inductive}} \vee v_j \in \mathcal{V}_{\text{inductive}}) \wedge (v_i, v_j) \in \mathcal{E}_{\text{test}}\}$. Inductive learning performance is evaluated only on the inductive edges. Dataset statistics are summarized in Table 7.

Table 7: Dataset statistics for inductive learning settings.

|  | $|\mathcal{V}_{\text{inductive}}|$ | $|\mathcal{E}_{\text{inductive}}|$ | $|\mathcal{E}_{\text{test}}|$ |
|---|---|---|---|
| Reddit | 134 | 4,704 | 100,867 |
| Wiki | 1,210 | 5,732 | 26,621 |
| MOOC | 342 | 8,645 | 61,763 |
| UCI | 391 | 4,876 | 8,976 |

We compare with baselines under the inductive learning setting with TGL framework for a fair comparison. As shown in the table, the performance of our method also works well on the inductive learning setting. Moreover, by combining with the results in Table 1, we found that changing from transductive to inductive learning does not affect our model performance much, which is potentially because our method has a simpler neural architecture and a stronger inductive bias for link prediction task.

### B.7 PRELIMINARY RESULTS OF NODE CLASSIFICATION

The design of SToNe is mainly focusing on the link prediction. For the completeness, we also report some preliminary node classification results using average precision in the table bellow. For this experiment, we use the identical network structure as the link prediction model. The evaluation

Table 8: Inductive learning setting average precision.

| Inductive average precision | Reddit | Wiki | MOOC | UCI |
|---|---|---|---|---|
| SToNe | $99.75 \pm 0.00$ | $99.70 \pm 0.05$ | $99.81 \pm 0.04$ | $90.94 \pm 0.33$ |
| TGN | $98.88 \pm 0.02$ | $98.88 \pm 0.06$ | $99.72 \pm 0.08$ | $83.27 \pm 2.01$ |
| JODIE | $96.06 \pm 0.03$ | $98.59 \pm 0.07$ | $86.90 \pm 0.06$ | $59.92 \pm 3.19$ |
| TGAT | $93.28 \pm 0.05$ | $95.82 \pm 0.12$ | $95.57 \pm 0.11$ | $67.65 \pm 0.73$ |
| DySAT | $90.83 \pm 0.02$ | $96.45 \pm 0.05$ | $98.65 \pm 0.19$ | $80.48 \pm 2.98$ |

strategy follows the framework in Zhou et al. (2022). It is worth noting that since our paper primarily focused on link prediction task, the performance of SToNe is not optimal. We plan to make it as an interesting future direction and enhance model performance by introducing the inductive bias of the node classification task.

Table 9: Node classification average precision.

| | SToNe | GraphMixder | JODIE | TGAT | TGN | DySAT |
|---|---|---|---|---|---|---|
| Reddit | 70.66 | 70.51 | 70.89 | 63.63 | 64.79 | 62.69 |
| Wiki | 85.78 | 83.93 | 80.63 | 85.30 | 87.13 | 85.30 |

## C  GENERALIZATION BOUND OF GNN-BASED METHOD

Recall that we compute the representation of node $v_i$ at time $t$ by applying GNN on the temporal graph that only considers the temporal edges with timestamp before time $t$. The GNN has trainable weight parameters $\boldsymbol{\theta} = \{\mathbf{W}^{(1)}, \ldots, \mathbf{W}^{(L)}\}$ and binary hyper-parameter $\alpha \in \{0, 1\}$ that controls whether residual connection is used.

**Representation computation.**   The final prediction on node $v_i$ is computed as $f_i(\boldsymbol{\theta}) = \mathbf{W}^{(L)}\mathbf{h}_i^{(L-1)}$, where the hidden representation $\mathbf{h}_i^{(L-1)}$ is computed by

$$\mathbf{h}_i^{(\ell)} = \sigma\big(\mathbf{W}^{(\ell)} \sum\nolimits_{j \in \mathcal{N}(i)} P_{ij}\mathbf{h}_j^{(\ell-1)}\big) + \alpha\mathbf{h}_i^{(\ell-1)} \in \mathbb{R}^m,$$

$$\mathbf{h}_i^{(1)} = \sigma\big(\mathbf{W}^{(1)} \sum\nolimits_{j \in \mathcal{N}(i)} P_{ij}\mathbf{x}_j\big) \in \mathbb{R}^m.$$

Here $\sigma(\cdot)$ is the activation function, $P_{ij}$ is the aggregation weight used for propagating information from node $v_j$ to $v_i$, and $\mathcal{N}(i)$ is the set of all neighbors of node $v_i$ in the temporal graph. For parameter dimension, we have $\mathbf{W}^{(1)} \in \mathbb{R}^{m \times d}$, $\mathbf{W}^{(\ell)} \in \mathbb{R}^{m \times m}$ for $2 \leq \ell \leq L - 1$, and $\mathbf{W}^{(L)} \in \mathbb{R}^{1 \times m}$, where $m$ is the hidden dimension and $d$ is the input dimension.

**Gradient computation.** The gradient of weight matrix $\mathbf{W}^{(\ell)}$, $\forall \ell \in [L-1]$ is computed by

$$\frac{\partial f_i(\boldsymbol{\theta})}{\partial \mathbf{W}^{(\ell)}} = \sum_{i_{L-2} \in \mathcal{N}(i_{L-1})} \sum_{i_{L-3} \in \mathcal{N}(i_{L-2})} \cdots \sum_{i_\ell \in \mathcal{N}(i_{\ell+1})} P_{i_{L-1},i_{L-2}} P_{i_{L-2},i_{L-3}} \ldots P_{i_{\ell+1},i_\ell} \cdot \mathbf{G}^\ell(i_\ell, \ldots, i_{L-1}).$$

Here $\mathbf{G}^\ell(i_\ell, \ldots, i_{L-1}) \in \mathbb{R}^{m \times m}$ is defined as

$$\mathbf{G}^\ell(i_\ell, \ldots, i_{L-1}) = \Big[\mathbf{W}^{(L)}\big(\mathbf{D}_{i_{L-1}}^{(L-1)}\mathbf{W}^{(L-1)} + \alpha\mathbf{I}_m\big) \ldots \big(\mathbf{D}_{i_{\ell+1}}^{(\ell+1)}\mathbf{W}^{(\ell+1)} + \alpha\mathbf{I}_m\big)\mathbf{D}_{i_\ell}^{(\ell)}\Big]^\top (\tilde{\mathbf{z}}_{i_\ell}^{(\ell-1)})^\top,$$

where $\mathbf{D}_{i,\ell}^{(\ell)} = \mathrm{diag}(\sigma'(\mathbf{z}_i^{(\ell)})) \in \mathbb{R}^{m \times m}$ is a diagonal matrix and $\tilde{\mathbf{z}}_{i_\ell}^{(\ell-1)}$ is the aggregation of neighbors' representations

$$\tilde{\mathbf{z}}_{i_\ell}^{(\ell-1)} = \sum_{i_{\ell-1} \in \mathcal{N}(i_\ell)} P_{i_\ell, i_{\ell-1}} \mathbf{h}_{i_{\ell-1}}^{(\ell-1)} \in \mathbb{R}^m, \ \forall \ell \in \{1, 2, \ldots, L-1\}.$$

Finally, the gradient with respect to the final layer weight matrix $\mathbf{W}^{(L)}$ is computed as

$$\frac{\partial f_i(\boldsymbol{\theta})}{\partial \mathbf{W}^{(L)}} = \mathbf{h}_i^{(L-1)}.$$

Please refer to Appendix G for detailed derivation of the gradients for the weight parameters in an $L$-layer GNN.

### C.1  PROOF SCRATCH

In the following, we summarize the proof on the generalization bound of GNN-based method as following:

- Firstly, we show in Lemma 1 that if two weight parameters are close to each other, then the node representation computed on these two parameters are also close.

- Secondly, we show that if two set of weight parameters are close to each other, then the neural network outputs computed on these weight parameters are almost linear in Lemma 2, and the computed loss is almost convex in Lemma 3.

- Thirdly, we show in Lemma 4 that with high probability, the gradient of neural network can be upper bounded, and this upper bound is dependent on the neural architecture.

- Thirdly, based on our previous results in Lemma 1, Lemma 3, and Lemma 4, we show in Lemma 5 that the difference between the cumulative loss over training to the loss computed on optimal solution could be upper bounded.

- Finally, we show in Lemma 6 that the expected 0-1 error is upper bounded. Then, by using our definition on the neural tangent random feature and feature-label alignment, we conclude the proof.

### C.2 USEFUL LEMMAS

For the ease of presentation, we introduce the following two definitions which will be used when introducing our lemmas.

**Definition 2** ($\omega$-neighborhood Cao & Gu (2019)). *For any $\widetilde{\boldsymbol{\theta}} = \{\widetilde{\mathbf{W}}^{(1)}, \ldots, \widetilde{\mathbf{W}}^{(L)}\}$, we define its $\omega$-neighborhood as*

$$\mathcal{B}(\boldsymbol{\theta}, \omega) = \{\widetilde{\boldsymbol{\theta}} \mid \|\widetilde{\mathbf{W}}^{(\ell)} - \mathbf{W}^{(\ell)}\|_\mathrm{F} \leq \omega, \; \ell \in [L]\}.$$

**Definition 3** (Neural tangent random feature Cao & Gu (2019)). *Let $\boldsymbol{\theta}_0$ be generated via the initialization. We define the neural tangent random feature function class as*

$$\mathcal{F}(\boldsymbol{\theta}_0, R) = \{f(\boldsymbol{\theta}_0) + \langle \nabla_\theta f(\boldsymbol{\theta}_0), \boldsymbol{\theta} \rangle \mid \boldsymbol{\theta} \in \mathcal{B}(\mathbf{0}, Rm^{-1/2})\},$$

*where $R > 0$ measures the size of the function class and $m$ is the hidden dimension of the neural network.*

In the following, we show that if the input weight parameters $\boldsymbol{\theta} = \{\mathbf{W}^{(1)}, \ldots, \mathbf{W}^{(L)}\}$ and $\widetilde{\boldsymbol{\theta}} = \{\widetilde{\mathbf{W}}^{(1)}, \ldots, \widetilde{\mathbf{W}}^{(L)}\}$ are close, the hidden representation of graph neural networks computed on $\boldsymbol{\theta}$ and $\widetilde{\boldsymbol{\theta}}$ does not change too much.

---

**Lemma 1.** *Let $\rho$ be the Lipschitz constant of the activation function and $m$ is the hidden dimension. Then with $\omega = \mathcal{O}(1/((3\rho + 1)\tau)^{(L-1)})$ and assuming $\widetilde{\boldsymbol{\theta}} \in \mathcal{B}(\boldsymbol{\theta}, \omega)$, we have $\|\widetilde{\mathbf{h}}_i^{(\ell)} - \mathbf{h}_i^{(\ell)}\|_2 = \mathcal{O}(1)$ with probability at least $1 - 2\ell \exp(-m/2) - \ell \exp(-\Omega(m))$.*

---

Please note that the smaller the distance $\omega$, the closer the representation $\|\widetilde{\mathbf{h}}_i^{(\ell)} - \mathbf{h}_i^{(\ell)}\|_2$. In particular, according to the proof of Lemma 1, we have $\|\widetilde{\mathbf{h}}_i^{(\ell)} - \mathbf{h}_i^{(\ell)}\|_2 \leq \mathcal{O}(\epsilon)$ by selecting $\omega = \mathcal{O}(\epsilon/((3\rho + 1)\tau)^{(L-1)})$ for any small $\epsilon > 0$. This conclusion will be later used in Lemma 5.

*Proof of Lemma 1.* When $\ell = 1$, we have for any node $v_i \in \mathcal{V}$

$$
\begin{aligned}
\|\widetilde{\mathbf{h}}_i^{(1)} - \mathbf{h}_i^{(1)}\|_2 &= \left\|\sigma\Big(\widetilde{\mathbf{W}}^{(1)} \sum_{j \in \mathcal{N}(i)} P_{ij}\mathbf{h}_j^{(0)}\Big) - \sigma\Big(\mathbf{W}^{(1)} \sum_{j \in \mathcal{N}(i)} P_{ij}\mathbf{h}_j^{(0)}\Big)\right\|_2 \\
&\underset{(a)}{\leq} \rho\|\widetilde{\mathbf{W}}^{(1)} - \mathbf{W}^{(1)}\|_2 \Big\| \sum_{j \in \mathcal{N}(i)} P_{ij}\mathbf{x}_j\Big\|_2 \\
&\underset{(b)}{\leq} \rho\omega\Big(\sum_{j \in \mathcal{N}(i)} P_{ij}\|\mathbf{x}_j\|_2\Big) \\
&\underset{(c)}{\leq} \rho\tau \cdot \omega = \mathcal{O}(1),
\end{aligned}
$$

where inequality (a) is due to the Lipschitz continuity of activation function, inequality (b) is due to the $\omega$-neighborhood definition $\widetilde{\mathbf{W}} \in \mathcal{B}(\mathbf{W}, \omega)$, and inequality (c) is due to Assumption 1 and Assumption 3.

Similarly, when $\ell \in \{2, \ldots, L - 1\}$, we have $\forall i \in \mathcal{V}$

$$
\begin{aligned}
\|\widetilde{\mathbf{h}}_i^{(\ell)} - \mathbf{h}_i^{(\ell)}\|_2 &\underset{(a)}{\leq} \left\|\sigma\Big(\widetilde{\mathbf{W}}^{(\ell)} \sum_{j \in \mathcal{N}(i)} P_{ij}\widetilde{\mathbf{h}}_j^{(\ell-1)}\Big) - \sigma\Big(\mathbf{W}^{(\ell)} \sum_{j \in \mathcal{N}(i)} P_{ij}\mathbf{h}_j^{(\ell-1)}\Big)\right\|_2 + \alpha\|\widetilde{\mathbf{h}}_i^{(\ell-1)} - \mathbf{h}_i^{(\ell-1)}\|_2 \\
&\underset{(b)}{\leq} \rho\|\widetilde{\mathbf{W}}^{(\ell)} - \mathbf{W}^{(\ell)}\|_2 \Big\| \sum_{j \in \mathcal{N}(i)} P_{ij}\widetilde{\mathbf{h}}_j^{(\ell-1)}\Big\|_2 + \rho\|\mathbf{W}^{(\ell)}\|_2\Big(\sum_{j \in \mathcal{N}(i)} P_{ij}\|\widetilde{\mathbf{h}}_j^{(\ell-1)} - \mathbf{h}_j^{(\ell-1)}\|_2\Big) \\
&\quad + \|\widetilde{\mathbf{h}}_i^{(\ell-1)} - \mathbf{h}_i^{(\ell-1)}\|_2 \\
&\underset{(c)}{\leq} \rho\omega\Big(\Big\|\sum_{j \in \mathcal{N}(i)} P_{ij}\widetilde{\mathbf{h}}_j^{(\ell-1)} - \sum_{j \in \mathcal{N}(i)} P_{ij}\mathbf{h}_j^{(\ell-1)}\Big\|_2 + \Big\|\sum_{j \in \mathcal{N}(i)} P_{ij}\mathbf{h}_j^{(\ell-1)}\Big\|_2\Big) \\
&\quad + \rho\|\mathbf{W}^{(\ell)}\|_2\Big(\sum_{j \in \mathcal{N}(i)} P_{ij}\|\widetilde{\mathbf{h}}_j^{(\ell-1)} - \mathbf{h}_j^{(\ell-1)}\|_2\Big) + \|\widetilde{\mathbf{h}}_i^{(\ell-1)} - \mathbf{h}_i^{(\ell-1)}\|_2 \\
&\leq \rho\omega\tau \cdot \max_{j \in \{i\} \cup \mathcal{N}(i)} \|\mathbf{h}_j^{(\ell-1)}\|_2 + \Big(\rho\tau(\|\mathbf{W}^{(\ell)}\|_2 + \omega) + 1\Big) \cdot \max_{j \in \{i\} \cup \mathcal{N}(i)} \|\widetilde{\mathbf{h}}_j^{(\ell-1)} - \mathbf{h}_j^{(\ell-1)}\|_2,
\end{aligned}
$$

where the inequality (a) and (c) are due to $\|\mathbf{A} + \mathbf{B}\|_2 \leq \|\mathbf{A}\|_2 + \|\mathbf{B}\|_2$, the inequality (b) is due to the Lipschitz continuity of activation function and $\widetilde{\mathbf{W}} \in \mathcal{B}(\mathbf{W}, \omega)$.

By Proposition 19, we know that with probability at least $1 - 2\exp(-m/2)$ we have $\|\mathbf{W}^{(\ell)}\|_2 \leq 3$ for all $\ell \in [L-1]$.

By Lemma 21, we know that with probability at least $1 - \exp(-\Omega(m))$ we have $\|\mathbf{h}_i^{(\ell)}\|_2 = \Theta(1)$.

Then, we have with probability at least $1 - 2\ell\exp(-m/2) - \ell\exp(-\Omega(m))$ for any $i \in \mathcal{V}$

$$
\begin{aligned}
\|\widetilde{\mathbf{h}}_i^{(\ell)} - \mathbf{h}_i^{(\ell)}\|_2 &\leq \left(\rho\tau(3+\omega) + 1\right) \cdot \max_{j \in \{i\} \cup \mathcal{N}(i)} \|\widetilde{\mathbf{h}}_j^{(\ell-1)} - \mathbf{h}_j^{(\ell-1)}\|_2 + \rho\omega\tau \cdot \max_{j \in \{i\} \cup \mathcal{N}(i)} \|\mathbf{h}_j^{(\ell-1)}\|_2 \\
&\leq \left(\rho\tau(3+\omega) + 1\right) \cdot \max_{j \in \mathcal{V}} \|\widetilde{\mathbf{h}}_j^{(\ell-1)} - \mathbf{h}_j^{(\ell-1)}\|_2 + \rho\omega\tau \cdot \max_{j \in \mathcal{V}} \|\mathbf{h}_j^{(\ell-1)}\|_2 \\
&\overset{(a)}{=} \left(\rho\tau(3+\omega) + 1\right) \cdot \max_{j \in \mathcal{V}} \|\widetilde{\mathbf{h}}_j^{(\ell-1)} - \mathbf{h}_j^{(\ell-1)}\|_2 + \rho\omega\tau \cdot \Theta(1) \\
&\leq \rho\omega\tau \cdot \Theta(1) \cdot \frac{((3+\omega)\rho\tau + 1)^{\ell-1} - 1}{(3+\omega)\rho\tau} + \rho\omega\tau \cdot ((3+\omega)\rho\tau + 1)^{\ell-1} \\
&\leq ((3+\omega)\rho\tau + 1)^{\ell-1}\left(\frac{\rho\omega\tau \cdot \Theta(1)}{(3+\omega)\rho\tau} + \rho\omega\tau\right) \\
&\leq ((3+\omega)\rho\tau + 1)^{\ell-1} \cdot \omega \cdot \left(\Theta(1) + \rho\tau\right),
\end{aligned}
$$

where the equality (a) is due to $\max_{j \in \mathcal{V}} \|\mathbf{h}_j^{(\ell)}\|_2 = \Theta(1)$.

By setting $\omega = 1/((3\rho + 1)\tau)^{L-1}$ we have the above equation upper bounded by $\mathcal{O}(1)$. $\qquad\square$

Then, in the next lemma, we show that if the initialization of two set of weight parameters $\boldsymbol{\theta} = \{\mathbf{W}^{(1)}, \ldots, \mathbf{W}^{(L)}\}$ and $\widetilde{\boldsymbol{\theta}} = \{\widetilde{\mathbf{W}}^{(1)}, \ldots, \widetilde{\mathbf{W}}^{(L)}\}$ are close, the neural network output $f_i(\boldsymbol{\theta})$ will be almost linear with respect to its weight parameters.

---

**Lemma 2.** *Let $\boldsymbol{\theta}, \widetilde{\boldsymbol{\theta}} \in \mathcal{B}(\boldsymbol{\theta}_0, \omega)$ with $\omega = \mathcal{O}\left(1/((3\rho+1)\tau)^{(L-1)}\right)$. Then, for any node $v_i \in \mathcal{V}$ in the graph, with probability at least $1 - 2(L-1)\exp(-m/2) - L\exp(-\Omega(m)) - 2/m$, we have*
$$\epsilon_{lin} := |f_i(\widetilde{\boldsymbol{\theta}}) - f_i(\boldsymbol{\theta}) - \langle\nabla f_i(\boldsymbol{\theta}), \widetilde{\boldsymbol{\theta}} - \boldsymbol{\theta}\rangle| = \mathcal{O}(1),$$
*where $f_i(\boldsymbol{\theta})$ is the prediction on the $L$-hop subgraph centered on the root node $v_i$.*

---

Please note that the smaller the distance $\omega$, the more the model output close to linear. In particular, according to the proof of Lemma 2 and the proof of Lemma 1, we have $\epsilon_{\text{lin}} \leq \mathcal{O}(\epsilon)$ by selecting $\omega = \mathcal{O}(\epsilon/((3\rho + 1)\tau)^{(L-1)})$ for any small $\epsilon > 0$. This conclusion will be later used in Lemma 5.

*Proof of Lemma 2.* According to the forward and backward propagation rules as we recapped at the beginning of this section, we have

$$\left| f_i(\widetilde{\boldsymbol{\theta}}) - f_i(\boldsymbol{\theta}) - \langle \nabla f_i(\boldsymbol{\theta}), \widetilde{\boldsymbol{\theta}} - \boldsymbol{\theta} \rangle \right|$$

$$= \left| f_i(\widetilde{\boldsymbol{\theta}}) - f_i(\boldsymbol{\theta}) - \sum_{\ell=1}^{L} \left\langle \frac{\partial f_i(\boldsymbol{\theta})}{\partial \mathbf{W}^{(\ell)}}, \widetilde{\mathbf{W}}^{(\ell)} - \mathbf{W}^{(\ell)} \right\rangle \right|$$

$$\le \left| f_i(\widetilde{\boldsymbol{\theta}}) - f_i(\boldsymbol{\theta}) - \left\langle \frac{\partial f_i(\boldsymbol{\theta})}{\partial \mathbf{W}^{(L)}}, \widetilde{\mathbf{W}}^{(L)} - \mathbf{W}^{(L)} \right\rangle \right| + \sum_{\ell=1}^{L-1} \left| \left\langle \frac{\partial f_i(\boldsymbol{\theta})}{\partial \mathbf{W}^{(\ell)}}, \widetilde{\mathbf{W}}^{(\ell)} - \mathbf{W}^{(\ell)} \right\rangle \right|$$

$$= \left| \widetilde{\mathbf{W}}^{(L)} (\widetilde{\mathbf{h}}_i^{(L-1)} - \mathbf{h}_i^{(L-1)})^\top \right|$$

$$+ \sum_{\ell=1}^{L-1} \Bigg[ \sum_{i_{L-2} \in \mathcal{N}(i_{L-1})} \sum_{i_{L-3} \in \mathcal{N}(i_{L-2})} \cdots \sum_{i_\ell \in \mathcal{N}(i_{\ell+1})} P_{i_{L-1}, i_{L-2}} P_{i_{L-2}, i_{L-3}} \cdots P_{i_{\ell+1}, i_\ell}$$

$$\left| \mathbf{W}^{(L)} \Big( \mathbf{D}_{i_{L-1}}^{(L-1)} \mathbf{W}^{(L-1)} + \alpha_{L-2} \mathbf{I}_m \Big) \cdots \Big( \mathbf{D}_{i_{\ell+1}}^{(\ell+1)} \mathbf{W}^{(\ell+1)} + \alpha_\ell \mathbf{I}_m \Big) \mathbf{D}_{i_\ell}^{(\ell)} (\widetilde{\mathbf{W}}^{(\ell)} - \mathbf{W}^{(\ell)}) \tilde{\mathbf{z}}_{i_\ell}^{(\ell-1)} \right| \Bigg]$$

$$\le \| \widetilde{\mathbf{W}}^{(L)} \|_2 \left\| \widetilde{\mathbf{h}}_i^{(L-1)} - \mathbf{h}_i^{(L-1)} \right\|_2$$

$$+ \sum_{\ell=1}^{L-1} \Bigg[ \sum_{i_{L-2} \in \mathcal{N}(i_{L-1})} \sum_{i_{L-3} \in \mathcal{N}(i_{L-2})} \cdots \sum_{i_\ell \in \mathcal{N}(i_{\ell+1})} P_{i_{L-1}, i_{L-2}} P_{i_{L-2}, i_{L-3}} \cdots P_{i_{\ell+1}, i_\ell}$$

$$\| \mathbf{W}^{(L)} \|_2 \underbrace{\left\| \Big( \mathbf{D}_{i_{L-1}}^{(L-1)} \mathbf{W}^{(L-1)} + \alpha_{L-2} \mathbf{I}_m \Big) \cdots \Big( \mathbf{D}_{i_{\ell+1}}^{(\ell+1)} \mathbf{W}^{(\ell+1)} + \alpha_\ell \mathbf{I}_m \Big) \mathbf{D}_{i_\ell}^{(\ell)} \right\|_2}_{(a)} \| \tilde{\mathbf{z}}_{i_\ell}^{(\ell-1)} \|_2 \left\| \widetilde{\mathbf{W}}^{(\ell)} - \mathbf{W}^{(\ell)} \right\|_2 \Bigg],$$

$$\underbrace{\hspace{10cm}}_{(b)}$$

where $\tilde{\mathbf{z}}_{i_\ell}^{(\ell-1)} = \sum_{i_{\ell-1} \in \mathcal{N}(i_\ell)} P_{i_\ell, i_{\ell-1}} \mathbf{h}_{i_{\ell-1}}^{(\ell-1)}$ and it has bounded $\ell_2$-norm

$$\| \tilde{\mathbf{z}}_{i_\ell}^{(\ell-1)} \|_2 \le \tau \max_{i_{\ell-1} \in \mathcal{N}(i_\ell)} \| \mathbf{h}_{i_{\ell-1}}^{(\ell-1)} \|_2 = \tau \Theta(1).$$

Since the derivative of activation function is bounded, we have $\| \mathbf{D}_{i,\ell} \|_2 \le \rho$ and $\| \mathbf{D}_i^{(\ell)} \mathbf{W}^{(\ell)} + \alpha_\ell \mathbf{I}_m \|_2 \le 3\rho + 1$.

Therefore, we know that the term (a) in the above equation could be upper bounded by $\rho(3\rho + 1)^{L-1-\ell}$.

Besides, we know that $\| \mathbf{W}^{(L)} \|_2 \le \sqrt{2}$ according to Lemma 20.

Finally, by plugging the results back, we can upper bound the term (b) in the above equation by

$$(b) \le C_\ell = \sqrt{2} \rho (3\rho + 1)^{L-\ell-1} \tau \Theta(1) \omega$$

and

$$\sum_{i_{L-2} \in \mathcal{N}(i_{L-1})} \sum_{i_{L-3} \in \mathcal{N}(i_{L-2})} \cdots \sum_{i_\ell \in \mathcal{N}(i_{\ell+1})} \Big( P_{i_{L-1}, i_{L-2}} P_{i_{L-2}, i_{L-3}} \cdots P_{i_{\ell+1}, i_\ell} C_\ell \Big)$$

$$= \sum_{i_{L-2} \in \mathcal{N}(i_{L-1})} P_{i_{L-1}, i_{L-2}} \Big( \sum_{i_{L-3} \in \mathcal{N}(i_{L-2})} P_{i_{L-2}, i_{L-3}} \cdots \Big( \sum_{i_\ell \in \mathcal{N}(i_{\ell+1})} P_{i_{\ell+1}, i_\ell} C_\ell \Big) \cdots \Big)$$

$$= \sqrt{2} \rho (3\rho + 1)^{L-\ell-1} \tau^{L-\ell} \Theta(1) \omega.$$

As a result, we have

$$|f_i(\widetilde{\boldsymbol{\theta}}) - f_i(\boldsymbol{\theta}) - \langle \nabla f_i(\boldsymbol{\theta}), \widetilde{\boldsymbol{\theta}} - \boldsymbol{\theta} \rangle| \le (\sqrt{2} + \omega) \cdot \mathcal{O}(1) + \sqrt{2} \rho \omega \cdot \Theta(1) \cdot \sum_{\ell=1}^{L-1} (3\rho + 1)^{L-\ell-1} \tau^{L-\ell}$$

$$\le (\sqrt{2} + \omega) \cdot \mathcal{O}(1) + \sqrt{2} \rho \omega \cdot \Theta(1) \cdot \Big( \frac{((3\rho + 1)\tau)^{L-1} - 1}{3\rho} \Big)$$

$$\le \mathcal{O}(1),$$

where the last inequality holds by selecting $\omega = 1/((3\rho + 1)\tau)^{L-1}$. $\qquad\square$

Let us define the logistic loss as

$$\text{loss}_i(\boldsymbol{\theta}) = \psi(y_i f_i(\boldsymbol{\theta})), \psi(x) = \log(1 + \exp(-x)).$$

Then, the following lemma shows that $\text{loss}_i(\boldsymbol{\theta})$ is almost a convex function of $\boldsymbol{\theta}$ for any $v_i \in \mathcal{V}$ if the initialization of two set of parameters are close to each other.

---

**Lemma 3.** *Let* $\boldsymbol{\theta}, \widetilde{\boldsymbol{\theta}} \in \mathcal{B}(\boldsymbol{\theta}_0, \omega)$ *with* $\omega = 1/((3\rho + 1)\tau)^{L-1}$ *for any* $v_i \in \mathcal{V}$, *it holds that*

$$loss_i(\widetilde{\boldsymbol{\theta}}) \geq loss_i(\boldsymbol{\theta}) + \langle \nabla_\theta loss_i(\boldsymbol{\theta}), \widetilde{\boldsymbol{\theta}} - \boldsymbol{\theta} \rangle - \epsilon_{lin}$$

*with probability at least* $1 - 2(L - 1)\exp(-m/2) - L\exp(-\Omega(m)) - 2/m$, *where*

$$\epsilon_{lin} = \left| \left\langle \nabla_\theta f_i(\boldsymbol{\theta}), \widetilde{\boldsymbol{\theta}} - \boldsymbol{\theta} \right\rangle - f_i(\widetilde{\boldsymbol{\theta}}) + f_i(\boldsymbol{\theta}) \right| = \mathcal{O}(1)$$

*according to Lemma 2.*

---

*Proof of Lemma 3.* The proof follows the proof of Lemma 9 in Zhu et al. (2022).

By the convexity of $\psi(\cdot)$, we know that $\psi(b) \geq \psi(a) + \psi'(a)(b - a)$. Therefore, we have

$$\begin{aligned}
\text{loss}_i(\widetilde{\boldsymbol{\theta}}) - \text{loss}_i(\boldsymbol{\theta}) &= \psi(y_i f_i(\widetilde{\boldsymbol{\theta}})) - \psi(y_i f_i(\boldsymbol{\theta})) \\
&\geq \psi'(y_i f_i(\boldsymbol{\theta}))\left( y_i f_i(\widetilde{\boldsymbol{\theta}}) - y_i f_i(\boldsymbol{\theta}) \right) \\
&= \psi'(y_i f_i(\boldsymbol{\theta})) y_i \left( f_i(\widetilde{\boldsymbol{\theta}}) - f_i(\boldsymbol{\theta}) \right).
\end{aligned}$$

By using the chain rule, we have

$$\left\langle \nabla_\theta \text{loss}_i(\boldsymbol{\theta}), \widetilde{\boldsymbol{\theta}} - \boldsymbol{\theta} \right\rangle = \psi'(y_i f_i(\boldsymbol{\theta})) \cdot y_i \left\langle \nabla_\theta f_i(\boldsymbol{\theta}), \widetilde{\boldsymbol{\theta}} - \boldsymbol{\theta} \right\rangle.$$

By combining the above equations, we have

$$\begin{aligned}
\text{loss}_i(\widetilde{\boldsymbol{\theta}}) - \text{loss}_i(\boldsymbol{\theta}) &\geq \psi'(y_i f_i(\boldsymbol{\theta})) y_i \left( f_i(\widetilde{\boldsymbol{\theta}}) - f_i(\boldsymbol{\theta}) \right) \\
&= \psi'(y_i f_i(\boldsymbol{\theta})) y_i \left\langle \nabla_\theta f_i(\boldsymbol{\theta}), \widetilde{\boldsymbol{\theta}} - \boldsymbol{\theta} \right\rangle - \psi'(y_i f_i(\boldsymbol{\theta})) y_i \left( \left\langle \nabla_\theta f_i(\boldsymbol{\theta}), \widetilde{\boldsymbol{\theta}} - \boldsymbol{\theta} \right\rangle - f_i(\widetilde{\boldsymbol{\theta}}) + f_i(\boldsymbol{\theta}) \right) \\
&\geq \left\langle \nabla_\theta \text{loss}_i(\boldsymbol{\theta}), \widetilde{\boldsymbol{\theta}} - \boldsymbol{\theta} \right\rangle - \left| \left\langle \nabla_\theta f_i(\boldsymbol{\theta}), \widetilde{\boldsymbol{\theta}} - \boldsymbol{\theta} \right\rangle - f_i(\widetilde{\boldsymbol{\theta}}) + f_i(\boldsymbol{\theta}) \right|,
\end{aligned}$$

where the inequality is due to $|\psi'(y_i f_i(\boldsymbol{\theta})) y_i| \leq 1$. $\qquad\square$

Moreover, by the gradient computation, we know that the gradient of the neural network function can be upper bounded.

---

**Lemma 4.** *For any* $v_i \in \mathcal{V}$ *with probability at least* $1 - 2(L - \ell)\exp(-m/2) - \ell \exp(-\Omega(m)) - 2/m$, *it holds that*

$$\left\| \frac{\partial f_i(\boldsymbol{\theta})}{\partial \mathbf{W}^{(\ell)}} \right\|_2, \left\| \frac{\partial loss_i(\boldsymbol{\theta})}{\partial \mathbf{W}^{(\ell)}} \right\|_2 \leq \Theta\left( ((3\rho + 1)\tau)^{L-\ell} \right)$$

---

*Proof of Lemma 4.* For $\ell = L$, we have for any $v_i \in \mathcal{V}$

$$\begin{aligned}
\left\| \frac{\partial f_i(\boldsymbol{\theta})}{\partial \mathbf{W}^{(L)}} \right\|_2 &= \|\mathbf{h}_i^{(L-1)}\|_2 \\
&\leq \max_{j \in \mathcal{V}} \|\mathbf{h}_j^{(L-1)}\|_2 = \Theta(1).
\end{aligned}$$

For $\ell \in [L-1]$, we have for any $v_i \in \mathcal{V}$

$$\left\| \frac{\partial f_i(\boldsymbol{\theta})}{\partial \mathbf{W}^{(\ell)}} \right\|_2 \leq \sum_{i_{L-2} \in \mathcal{N}(i_{L-1})} \sum_{i_{L-3} \in \mathcal{N}(i_{L-2})} \cdots \sum_{i_\ell \in \mathcal{N}(i_{\ell+1})} P_{i_{L-1},i_{L-2}} P_{i_{L-2},i_{L-3}} \cdots P_{i_{\ell+1},i_\ell} \left\| \mathbf{G}^\ell(i_\ell, \ldots, i_{L-1}) \right\|_2.$$

To upper bound $\left\| \mathbf{G}^\ell(i_\ell, \ldots, i_{L-1}) \right\|_2$, we have

$$
\begin{aligned}
\left\| \mathbf{G}^\ell(i_\ell, \ldots, i_{L-1}) \right\|_2 &= \left\| \left[ \mathbf{W}^{(L)} \left( \mathbf{D}_{i_{L-1}}^{(L-1)} \mathbf{W}^{(L-1)} + \alpha_{L-2} \mathbf{I}_m \right) \cdots \left( \mathbf{D}_{i_{\ell+1}}^{(\ell+1)} \mathbf{W}^{(\ell+1)} + \alpha_\ell \mathbf{I}_m \right) \mathbf{D}_{i_\ell}^{(\ell)} \right]^\top (\tilde{\mathbf{z}}_{i_\ell}^{(\ell-1)})^\top \right\|_2 \\
&\leq \left\| \mathbf{W}^{(L)} \right\|_2 \left\| \left( \mathbf{D}_{i_{L-1}}^{(L-1)} \mathbf{W}^{(L-1)} + \alpha_{L-2} \mathbf{I}_m \right) \cdots \left( \mathbf{D}_{i_{\ell+1}}^{(\ell+1)} \mathbf{W}^{(\ell+1)} + \alpha_\ell \mathbf{I}_m \right) \mathbf{D}_{i_\ell}^{(\ell)} \right\|_2 \left\| \tilde{\mathbf{z}}_{i_\ell}^{(\ell-1)} \right\|_2 \\
&\leq \sqrt{2} \rho \tau (3\rho + 1)^{L-\ell-1} \Theta(1),
\end{aligned}
$$

where the last inequality follows the proof of Lemma 2.

By combining the above results, we have

$$
\begin{aligned}
\left\| \frac{\partial f_i(\boldsymbol{\theta})}{\partial \mathbf{W}^{(\ell)}} \right\|_2 &\leq \sqrt{2} \rho((3\rho + 1)\tau)^{L-\ell-1} \Theta(1) \\
&\leq \Theta((3\rho + 1)\tau)^{L-\ell}.
\end{aligned}
$$

Moreover, the above inequality also implies

$$
\begin{aligned}
\left\| \frac{\partial \mathrm{loss}_i(\boldsymbol{\theta})}{\partial \mathbf{W}^{(\ell)}} \right\|_2 &= |\psi'(y_i f_i(\boldsymbol{\theta})) \cdot y_i| \cdot \left\| \frac{\partial f_i(\boldsymbol{\theta})}{\partial \mathbf{W}^{(\ell)}} \right\|_2 \\
&\leq \left\| \frac{\partial f_i(\boldsymbol{\theta})}{\partial \mathbf{W}^{(\ell)}} \right\|_2,
\end{aligned}
$$

where the last inequality is due to $|\psi'(y_i f_i(\boldsymbol{\theta})) \cdot y_i| \leq 1$. $\qquad \square$

In the following, we show that the cumulative loss can be upper bounded under small changes on the weight parameters.

---

**Lemma 5.** *For any $\epsilon, \delta, R > 0$, there exists*

$$m^\star = \mathcal{O} \left( \frac{((3\rho + 1)\tau)^{4(L-1)} L^2 R^4}{4\epsilon^4} \right) \log(1/\delta),$$

*such that if $m \geq m^\star$, then with probability at least $1 - \delta$ over the randomness of $\boldsymbol{\theta}_0$, for any $\boldsymbol{\theta}_\star \in \mathcal{B}(\boldsymbol{\theta}_0, Rm^{-1/2})$, with $\eta = \frac{\epsilon}{mL((3\rho+1)\tau)^{2(L-1)}}$ and $N = \frac{L^2 R^2 ((3\rho+1)\tau)^{2(L-1)}}{2\epsilon^2}$, the cumulative loss can be upper bounded by*

$$\frac{1}{N} \sum_{i=1}^N \mathrm{loss}_i(\boldsymbol{\theta}_{i-1}) \leq \frac{1}{N} \sum_{i=1}^N \mathrm{loss}_i(\boldsymbol{\theta}^\star) + \mathcal{O}(\epsilon).$$

---

*Proof of Lemma 5.* Let us define $\boldsymbol{\theta}^\star$ as the optimal solution that could minimize the cumulative loss over $N$ epochs, where at each epoch only a single data point is used as defined in Algorithm 1

$$\boldsymbol{\theta}^\star = \arg\min_{\boldsymbol{\theta} \in \mathcal{B}(\boldsymbol{\theta}_0, \omega)} \sum_{i=1}^N \mathrm{loss}_i(\boldsymbol{\theta}).$$

Without loss of generality, let us assume the epoch loss $\mathrm{loss}_i(\boldsymbol{\theta}_{i-1})$ is computed on the $i$-th node. Then, in the following, we try to show $\boldsymbol{\theta}_0, \boldsymbol{\theta}_1, \ldots, \boldsymbol{\theta}_{N-1} \in \mathcal{B}(\boldsymbol{\theta}_0, \omega)$, where $\omega = \epsilon/((3\rho + 1)\tau)^{L-1}$

First of all, it is clear that $\boldsymbol{\theta}_0 \in \mathcal{B}(\boldsymbol{\theta}_0, \omega)$. Then, to show $\boldsymbol{\theta}_n \in \mathcal{B}(\boldsymbol{\theta}_0, \omega)$ for any $n \in \{1, \ldots, N-1\}$, we use our previous conclusion on the upper bound of gradient $\|\partial \mathrm{loss}_i(\boldsymbol{\theta})/\partial \mathbf{W}^{(\ell)}\| \leq \Theta((3\rho +$

$1)\tau)^{L-\ell}$ in Lemma 4 and have

$$
\begin{aligned}
\|\mathbf{W}_n^{(\ell)} - \mathbf{W}_0^{(\ell)}\|_2 &\leq \sum_{i=1}^{n} \|\mathbf{W}_i^{(\ell)} - \mathbf{W}_{i-1}^{(\ell)}\|_2 \\
&= \sum_{i=1}^{n} \eta \Big\| \frac{\partial \mathrm{loss}_{i-1}(\boldsymbol{\theta}_{i-1})}{\partial \mathbf{W}_{i-1}^{(\ell)}} \Big\|_2 \\
&\leq \Theta\Big( \eta N ((3\rho + 1)\tau)^{L-\ell} \Big).
\end{aligned}
$$

By plugging in the choice of $\eta = \frac{\epsilon}{mL((3\rho+1)\tau)^{2(L-1)}}$ and $N = \frac{L^2 R^2 ((3\rho+1)\tau)^{2(L-1)}}{2\epsilon^2}$, we have

$$
\begin{aligned}
\|\mathbf{W}_n^{(\ell)} - \mathbf{W}_0^{(\ell)}\|_2 &\leq \Theta\Big( \eta N ((3\rho + 1)\tau)^{L-\ell-1} \Big) \\
&= \Theta\Big( \frac{\epsilon}{mL((3\rho+1)\tau)^{2(L-1)}} \frac{L^2 R^2 ((3\rho+1)\tau)^{2(L-1)}}{2\epsilon^2} ((3\rho+1)\tau)^{L-\ell} \Big) \\
&= \Theta\Big( \frac{LR^2}{2m\epsilon} ((3\rho+1)\tau)^{L-\ell} \Big).
\end{aligned}
$$

After changing norm from $\ell_2$-norm to Frobenius-norm, we have

$$
\|\mathbf{W}_n^{(\ell)} - \mathbf{W}_0^{(\ell)}\|_\mathrm{F} \leq \sqrt{m} \times \Theta\Big( \frac{LR^2}{2m\epsilon} ((3\rho+1)\tau)^{L-\ell} \Big), \tag{2}
$$

By plugging in the selection of hidden dimension $m$, we have

$$
\|\mathbf{W}_n^{(\ell)} - \mathbf{W}_0^{(\ell)}\|_\mathrm{F} \leq \frac{\epsilon}{((3\rho+1)\tau)^{L-1}},
$$

which means $\boldsymbol{\theta}_0, \boldsymbol{\theta}_1, \ldots, \boldsymbol{\theta}_{N-1} \in \mathcal{B}(\boldsymbol{\theta}_0, \omega)$ for $\omega = \epsilon/((3\rho+1)\tau)^{L-1}$

Then, our next step is to bound $\mathrm{loss}_i(\boldsymbol{\theta}_i) - \mathrm{loss}_i(\boldsymbol{\theta}_\star)$. By Lemma 3, we know that

$$
\begin{aligned}
\mathrm{loss}_{i+1}(\boldsymbol{\theta}_i) - \mathrm{loss}_{i+1}(\boldsymbol{\theta}_\star) &\leq \Big\langle \nabla_{\boldsymbol{\theta}} \mathrm{loss}_{i+1}(\boldsymbol{\theta}_i), \boldsymbol{\theta}_i - \boldsymbol{\theta}_\star \Big\rangle + \epsilon_{\mathrm{lin}} \\
&= \sum_{\ell=1}^{L} \Big\langle \frac{\partial \mathrm{loss}_{i+1}(\boldsymbol{\theta}_i)}{\partial \mathbf{W}^{(\ell)}}, \mathbf{W}_i^{(\ell)} - \mathbf{W}_\star^{(\ell)} \Big\rangle + \epsilon_{\mathrm{lin}} \\
&= \frac{1}{\eta} \sum_{\ell=1}^{L} \Big\langle \mathbf{W}_i^{(\ell)} - \mathbf{W}_{i+1}^{(\ell)}, \mathbf{W}_i^{(\ell)} - \mathbf{W}_\star^{(\ell)} \Big\rangle + \epsilon_{\mathrm{lin}} \\
&\leq \frac{1}{2\eta} \sum_{\ell=1}^{L} \Big( \|\mathbf{W}_i^{(\ell)} - \mathbf{W}_{i+1}^{(\ell)}\|_\mathrm{F}^2 + \|\mathbf{W}_i^{(\ell)} - \mathbf{W}_\star^{(\ell)}\|_\mathrm{F}^2 - \|\mathbf{W}_{i+1}^{(\ell)} - \mathbf{W}_\star^{(\ell)}\|_\mathrm{F}^2 \Big) + \epsilon_{\mathrm{lin}}.
\end{aligned}
$$

Then, our next step is to upper bound each term on the right hand size of inequality.

(1) According to the proof of Lemma 2, we have $\epsilon_{\mathrm{lin}} \leq \mathcal{O}(\epsilon)$ by selecting $\omega = \mathcal{O}(\epsilon/((3\rho + 1)\tau)^{(L-1)})$.

(2) Recall that $\|\mathbf{W}_i^{(\ell)} - \mathbf{W}_{i+1}^{(\ell)}\|_\mathrm{F}^2$ could be upper bounded by

$$
\begin{aligned}
\|\mathbf{W}_i^{(\ell)} - \mathbf{W}_{i+1}^{(\ell)}\|_\mathrm{F}^2 &\leq \eta^2 \Big\| \frac{\partial \mathrm{loss}_{i+1}(\boldsymbol{\theta}_i)}{\partial \mathbf{W}^{(\ell)}} \Big\|_\mathrm{F}^2 \\
&\leq \eta^2 m \Big\| \frac{\partial \mathrm{loss}_{i+1}(\boldsymbol{\theta}_i)}{\partial \mathbf{W}^{(\ell)}} \Big\|_2^2 \\
&\leq \Theta\Big( m\eta^2 ((3\rho+1)\tau)^{2(L-\ell)} \Big).
\end{aligned}
$$

(3) By finite sum $\|\mathbf{W}_i^{(\ell)} - \mathbf{W}_\star^{(\ell)}\|_{\mathrm{F}}^2 - \|\mathbf{W}_{i+1}^{(\ell)} - \mathbf{W}_\star^{(\ell)}\|_{\mathrm{F}}^2$ from $i = 0$ to $N - 1$, we have

$$\frac{1}{N} \sum_{i=0}^{N-1} (\|\mathbf{W}_i^{(\ell)} - \mathbf{W}_\star^{(\ell)}\|_{\mathrm{F}}^2 - \|\mathbf{W}_{i+1}^{(\ell)} - \mathbf{W}_\star^{(\ell)}\|_{\mathrm{F}}^2) = \frac{1}{N}\|\mathbf{W}_0^{(\ell)} - \mathbf{W}_\star^{(\ell)}\|_{\mathrm{F}}^2 \underbrace{- \frac{1}{N}\|\mathbf{W}_{N+1}^{(\ell)} - \mathbf{W}_\star^{(\ell)}\|_{\mathrm{F}}^2}_{\leq 0}$$

$$\leq \frac{R^2}{mN},$$

where the inequality is due to $\boldsymbol{\theta}_\star \in \mathcal{B}(\boldsymbol{\theta}_0, Rm^{-1/2})$.

Finally, by combining the results above, we have

$$\frac{1}{N} \sum_{i=1}^{N} \mathrm{loss}_i(\boldsymbol{\theta}_{i-1}) - \frac{1}{N} \sum_{i=1}^{N} \mathrm{loss}_i(\boldsymbol{\theta}_\star) \leq \frac{R^2 L}{2m\eta N} + \sum_{\ell=1}^{L} \Theta\left(\frac{m\eta}{2}(3\rho\tau + 1)^{2(L-1)}\right) + \mathcal{O}(\epsilon).$$

By selecting $\eta = \frac{\epsilon}{mL((3\rho+1)\tau)^{2(L-1)}}$ and $N = \frac{L^2 R^2 ((3\rho+1)\tau)^{2(L-1)}}{2\epsilon^2}$, we have

$$\frac{R^2 L}{2m\eta N} = \frac{R^2 L}{2m} \times \frac{mL((3\rho+1)\tau)^{2(L-1)}}{\epsilon} \times \frac{2\epsilon^2}{L^2 R^2 (3\rho\tau + 1)^{2(L-1)}} = \epsilon,$$

$$\sum_{\ell=1}^{L} \Theta\left(\frac{m\eta}{2}((3\rho+1)\tau)^{2(L-\ell)}\right) \leq L \times \Theta\left(m\eta \cdot ((3\rho+1)\tau)^{2(L-1)}\right)$$

$$= \Theta\left(m \cdot \frac{\epsilon}{mL((3\rho+1)\tau)^{2(L-1)}}((3\rho+1)\tau)^{2(L-1)}\right) = \Theta(\epsilon).$$

Therefore, combining the results above, we have

$$\frac{1}{N} \sum_{i=1}^{N} \mathrm{loss}_i(\boldsymbol{\theta}_{i-1}) - \frac{1}{N} \sum_{i=1}^{N} \mathrm{loss}_i(\boldsymbol{\theta}_\star) \leq \mathcal{O}(\epsilon)$$

$\square$

By plugging in the selection of $\epsilon = \frac{LR((3\rho+1)\tau)^{(L-1)}}{\sqrt{2N}}$ to Lemma 5, we have

> **Corollary 1.** *For any $\delta > 0$ and $R > 0$, there exists $m^\star = \mathcal{O}\left(N^2/L^2\right)\log(1/\delta)$, such that if $m \geq m^\star$, then with probability at least $1 - \delta$ over the randomness of $\boldsymbol{\theta}_0$, for any $\boldsymbol{\theta}_\star \in \mathcal{B}(\boldsymbol{\theta}_0, Rm^{-1/2})$, with the selection of learning rate $\eta = \frac{R}{m\sqrt{2N}((3\rho+1)\tau)^{(L-1)}}$, the cumulative loss can be upper bounded by*
>
> $$\frac{1}{N} \sum_{i=1}^{N} loss_i(\boldsymbol{\theta}_{i-1}) \leq \frac{1}{N} \sum_{i=1}^{N} loss_i(\boldsymbol{\theta}^\star) + \mathcal{O}\left(\frac{LR((3\rho+1)\tau)^{(L-1)}}{\sqrt{N}}\right).$$

In the following, we present the expected 0-1 error bound of multi-layer GNN, which consists of two terms: (1) the expected 0-1 error with the neural tangent random feature function and (2) the standard large-deviation error term.

**Lemma 6.** *For any $\delta \in (0, 1/e]$ and $R > 0$, there exists $m^\star = \mathcal{O}\left(N^2/L^2\right)\log(1/\delta)$, such that if $m \geq m^\star$, then with probability at least $1 - \delta$ over the randomness of $\boldsymbol{\theta}_0$, with the selection of learning rate $\eta = \frac{R}{m\sqrt{2N}((3\rho+1)\tau)^{(L-1)}}$, we have*

$$\mathbb{E}\left[loss_N^{0-1}(\widetilde{\boldsymbol{\theta}})|\mathcal{D}_1^{N-1}\right]$$

$$\leq \frac{4}{N} \inf_{f \in \mathcal{F}(\boldsymbol{\theta}_0, R)} \sum_{i=1}^{N} \psi(y_i f_i) + \mathcal{O}\left(\frac{LR((3\rho+1)\tau)^{(L-1)}}{\sqrt{N}}\right) + \mathcal{O}\left(\sqrt{\frac{\log(1/\delta)}{N}}\right) + \Delta,$$

*where $\mathcal{F}(\boldsymbol{\theta}_0, R)$ is the neural tangent random feature function class, $\widetilde{\boldsymbol{\theta}}$ is uniformly selected from $\{\boldsymbol{\theta}_0, \ldots, \boldsymbol{\theta}_{N-1}\}$, $\mathcal{D}_1^{N-1}$ is the sequence of data points sampled before the $N$-th iteration, and the expectation is computed on the uniform selection of weight parameters $\widetilde{\boldsymbol{\theta}}$ and the condition sampling of $N$-th iteration data examples.*

*Proof of Lemma 6.* The proof is based on the proof of Theorem 3.3 in Cao & Gu (2019).

Let us recall that the 0-1 loss is defined as

$$loss_i^{0-1}(\boldsymbol{\theta}) = \mathbf{1}\{y_i f_i(\boldsymbol{\theta}) < 0\}, \ \forall i \in [N]$$

Since the cross entropy loss $\psi(\cdot)$ satisfies $\mathbf{1}\{z \leq 0\} \leq 4\psi(z)$, we have $loss_i^{0-1}(\boldsymbol{\theta}) \leq 4loss_i(\boldsymbol{\theta})$. Then, we have with probability at least $1 - \delta$

$$\frac{1}{N}\sum_{i=1}^{N} loss_i^{0-1}(\boldsymbol{\theta}_{i-1}) \leq \frac{4}{N}\sum_{i=1}^{N} loss_i(\boldsymbol{\theta}^\star) + \mathcal{O}\left(\frac{LR((3\rho+1)\tau)^{(L-1)}}{\sqrt{N}}\right)$$

$$\leq \frac{4}{N}\sum_{i=1}^{N} \psi(y_i f_i(\boldsymbol{\theta}_\star)) + \mathcal{O}\left(\frac{LR((3\rho+1)\tau)^{(L-1)}}{\sqrt{N}}\right)$$

Let us define $F_i(\boldsymbol{\theta}_0, \boldsymbol{\theta}^\star) := f_i(\boldsymbol{\theta}_0) + \langle \nabla_\theta f_i(\boldsymbol{\theta}_0), \boldsymbol{\theta}^\star - \boldsymbol{\theta}_0 \rangle$. Since $\psi(\cdot)$ is 1-Lipschitz continuous, we have

$$\psi(y_i f_i(\boldsymbol{\theta})) - \psi(y_i F_i(\boldsymbol{\theta}_0, \boldsymbol{\theta}^\star)) \leq y_i \Big( f_i(\boldsymbol{\theta}) - F_i(\boldsymbol{\theta}_0, \boldsymbol{\theta}^\star) \Big)$$

$$= y_i \Big( f_i(\boldsymbol{\theta}) - f_i(\boldsymbol{\theta}_0) + \langle \nabla_\theta f_i(\boldsymbol{\theta}_0), \boldsymbol{\theta}_0 - \boldsymbol{\theta}^\star \rangle \Big) \leq \mathcal{O}(1),$$

where the last inequality is due to Lemma 2.

By combining the results above, we have

$$\frac{1}{N}\sum_{i=1}^{N} loss_i^{0-1}(\boldsymbol{\theta}_{i-1}) \leq \frac{4}{N}\sum_{i=1}^{N} \psi(y_i F_i(\boldsymbol{\theta}_0, \boldsymbol{\theta}^\star)) + \mathcal{O}\left(\frac{LR((3\rho+1)\tau)^{(L-1)}}{\sqrt{N}}\right)$$

$$\leq \frac{4}{N} \inf_{f \in \mathcal{F}(\boldsymbol{\theta}_0, R)} \sum_{i=1}^{N} \psi(y_i f_i) + \mathcal{O}\left(\frac{LR((3\rho+1)\tau)^{(L-1)}}{\sqrt{N}}\right),$$

where the second inequality is due to the definition of neural tangent random feature

$$\mathcal{F}(\boldsymbol{\theta}_0, R) = \{f(\boldsymbol{\theta}_0) + \langle \nabla_\theta f(\boldsymbol{\theta}_0), \boldsymbol{\theta} \rangle \mid \boldsymbol{\theta} \in \mathcal{B}(\mathbf{0}, Rm^{-1/2})\}$$

and $\boldsymbol{\theta}^\star \in \mathcal{B}(\boldsymbol{\theta}_0, Rm^{-1/2})$.

By Proposition 1, we have with probability at least $1 - \delta$,

$$\frac{1}{N}\sum_{i=1}^{N} \mathbb{E}\left[loss_i^{0-1}(\boldsymbol{\theta}_{i-1})|\mathcal{D}_1^{i-1}\right] \leq \frac{4}{N} \inf_{f \in \mathcal{F}(\boldsymbol{\theta}_0, R)} \sum_{i=1}^{N} \psi(y_i f_i) + \mathcal{O}\left(\frac{LR((3\rho+1)\tau)^{(L-1)}}{\sqrt{N}}\right) + \mathcal{O}\left(\sqrt{\frac{\log(1/\delta)}{N}}\right)$$

By using Assumption 4, we have

$$
\mathbb{E}\left[\text{loss}_N^{0-1}(\boldsymbol{\theta})|\mathcal{D}_1^{N-1}\right] = \frac{1}{N}\sum_{i=1}^{N}\mathbb{E}\left[\text{loss}_N^{0-1}(\boldsymbol{\theta}_{i-1})|\mathcal{D}_1^{N-1}\right]
$$

$$
\leq \frac{4}{N}\inf_{f\in\mathcal{F}(\boldsymbol{\theta}_0,R)}\sum_{i=1}^{N}\psi(y_if_i) + \mathcal{O}\Big(\frac{LR((3\rho+1)\tau)^{(L-1)}}{\sqrt{N}}\Big) + \mathcal{O}\Big(\sqrt{\frac{\log(1/\delta)}{N}}\Big) + \Delta
$$

where $\widetilde{\boldsymbol{\theta}}$ is uniformly sampled from $\{\boldsymbol{\theta}_1,\ldots,\boldsymbol{\theta}_{N-1}\}$. $\qquad\square$

## C.3 PROOF OF THEOREM 2

In the following, we show that the expected error is bounded by $\sqrt{\mathbf{y}^\top(\mathbf{JJ}^\top)^{-1}\mathbf{y}}$ and is proportional to $L((3\rho+1)\tau)^{L-1}/\sqrt{N}$.

> **Theorem 2** (Multi-layer GNN-based method). *For any $\delta \in (0, 1/e]$ and $R > 0$, there exists there exists $m^\star = \mathcal{O}\left(N^2/L^2\right)\log(1/\delta)$, such that if $m \geq m^\star$, then with probability at least $1 - \delta$ over the randomness of $\boldsymbol{\theta}_0$ with step size $\eta = \frac{R}{m\sqrt{2N}((3\rho+1)\tau)^{(L-1)}}$ we have*
>
> $$
> \mathbb{E}\left[loss_N^{0-1}(\widetilde{\boldsymbol{\theta}})|\mathcal{D}_1^{N-1}\right] \leq \mathcal{O}\Big(\frac{LR((3\rho+1)\tau)^{(L-1)}}{\sqrt{N}}\Big) + \mathcal{O}\Big(\sqrt{\frac{\log(1/\delta)}{N}}\Big) + \Delta,
> $$
>
> *where $R = \mathcal{O}(\sqrt{\mathbf{y}^\top(\mathbf{JJ}^\top)^{-1}\mathbf{y}})$, $\widetilde{\boldsymbol{\theta}}$ is uniformly selected from $\{\boldsymbol{\theta}_0,\ldots,\boldsymbol{\theta}_{N-1}\}$, $\mathcal{D}_1^{N-1}$ is the sequence of data points sampled before the $N$-th iteration, and the expectation is computed on the uniform selection of weight parameters $\widetilde{\boldsymbol{\theta}}$ and the condition sampling of $N$-th iteration data examples.*

The proof of Theorem 2 follows the proof of Corollary 3.10 in Cao & Gu (2019).

To begin with, let us recall from Lemma 6 that

$$
\mathbb{E}\left[\text{loss}_N^{0-1}(\boldsymbol{\theta})|\mathcal{D}_1^{N-1}\right]
$$
$$
\leq \frac{4}{N}\inf_{f\in\mathcal{F}(\boldsymbol{\theta}_0,R)}\sum_{i=1}^{N}\psi(y_if_i) + \mathcal{O}\Big(\frac{LR((3\rho+1)\tau)^{(L-1)}}{\sqrt{N}}\Big) + \mathcal{O}\left(\sqrt{\frac{\log(1/\delta)}{N}}\right) + \Delta.
$$

Our goal is to show that $\exists f' \in \mathcal{F}(\boldsymbol{\theta}_0, R)$ such that $\psi(y_if_i') = \log(1+\exp(-y_if_i)) \leq \frac{1}{\sqrt{N}}, \forall v_i \in \mathcal{V}$, which implies

$$
y_if_i' \geq -\log(\exp(N^{-1/2}) - 1), \forall v_i \in \mathcal{V}.
$$

Let us define $B = -\log(\exp(N^{-1/2}) - 1)$ and $B' = \max_{i\in\mathcal{V}}|f_i(\boldsymbol{\theta}_0)|$ for notation simplicity.

By the definition of $\mathcal{F}(\boldsymbol{\theta}_0, R)$, we know that $\exists\, \boldsymbol{\theta} \in \mathcal{B}(\boldsymbol{\theta}_0, Rm^{-1/2})$ such that

$$
y_if_i' = y_i\Big(f_i(\boldsymbol{\theta}_0) + \langle\nabla_\theta f_i(\boldsymbol{\theta}_0), \boldsymbol{\theta}\rangle\Big) = y_if_i(\boldsymbol{\theta}_0) + y_i\langle\nabla_\theta f_i(\boldsymbol{\theta}_0), \boldsymbol{\theta}\rangle
$$
$$
\geq -B' + y_i\langle\nabla_\theta f_i(\boldsymbol{\theta}_0), \boldsymbol{\theta}\rangle \geq B,
$$

where the inequality holds because

$$
-\max_{i\in\mathcal{V}}|f_i(\boldsymbol{\theta}_0)| \leq y_if_i(\boldsymbol{\theta}_0) \leq \max_{i\in\mathcal{V}}|f_i(\boldsymbol{\theta}_0)| \text{ and } \langle\nabla_\theta f_i(\boldsymbol{\theta}_0), \boldsymbol{\theta}\rangle = y_i(B + B').
$$

For notation simplicity, we define

$$
\hat{y}_i = \langle\nabla_\theta f_i(\boldsymbol{\theta}_0), \boldsymbol{\theta}\rangle,
$$
$$
\hat{y}_i = (B + B')y_i,
$$
$$
M = md + m + m^2(L - 2).
$$

Besides, let us denote the stack of gradient is

$$\mathbf{J} = \begin{bmatrix} \text{vec}(\nabla_\theta f_1(\boldsymbol{\theta}_0)) \\ \text{vec}(\nabla_\theta f_1(\boldsymbol{\theta}_1)) \\ \vdots \\ \text{vec}(\nabla_\theta f_1(\boldsymbol{\theta}_N)) \end{bmatrix} \in \mathbb{R}^{N \times M}, \ M = |\boldsymbol{\theta}|$$

Besides, let us define $\mathbf{J} = \mathbf{P}\boldsymbol{\Lambda}\mathbf{Q}^\top$ as the singular value decomposition of $\mathbf{J}$, where $\mathbf{P} \in \mathbb{R}^{N \times N}, \mathbf{Q} \in \mathbb{R}^{M \times M}$ have orthonormal columns, and $\boldsymbol{\Lambda} \in \mathbb{R}^{N \times M}$ is the singular value matrix.

Let us define $\mathbf{w} = \mathbf{Q}\boldsymbol{\Lambda}^{-1}\mathbf{P}^\top \hat{\mathbf{y}}$, then multiplying both sides by $\mathbf{J}$ we have

$$\mathbf{J}\mathbf{w} = (\mathbf{P}\boldsymbol{\Lambda}\mathbf{Q}^\top)(\mathbf{Q}\boldsymbol{\Lambda}^{-1}\mathbf{P}^\top)\hat{\mathbf{y}} = \hat{\mathbf{y}}.$$

Since $\|\hat{\mathbf{y}}\|_2^2 = \|(B + B')\mathbf{y}\|_2^2$, we have

$$\begin{aligned} \|\mathbf{w}\|_2^2 &= \|\mathbf{Q}\boldsymbol{\Lambda}^{-1}\mathbf{P}^\top \hat{\mathbf{y}}\|_2^2 \\ &= \hat{\mathbf{y}}^\top \mathbf{P}\boldsymbol{\Lambda}^{-1}\mathbf{Q}^\top \mathbf{Q}\boldsymbol{\Lambda}^{-1}\mathbf{P}^\top \hat{\mathbf{y}} \\ &= \hat{\mathbf{y}}^\top \mathbf{P}\boldsymbol{\Lambda}^{-2}\mathbf{P}^\top \hat{\mathbf{y}} \\ &= \hat{\mathbf{y}}^\top (\mathbf{J}\mathbf{J}^\top)^{-1}\hat{\mathbf{y}} \\ &= (B + B')^2 \cdot \mathbf{y}^\top (\mathbf{J}\mathbf{J}^\top)^{-1}\mathbf{y}. \end{aligned}$$

Let $\boldsymbol{\theta}$ be the parameters reshaped from $\mathbf{w}$, then we have

$$\|\boldsymbol{\theta}\|_\mathrm{F} \leq \mathcal{O}\left(\sqrt{\mathbf{y}^\top (\mathbf{J}\mathbf{J}^\top)^{-1}\mathbf{y}}\right) \Rightarrow \boldsymbol{\theta} \in \mathcal{B}\left(\mathbf{0}, \mathcal{O}\left(\sqrt{\mathbf{y}^\top (\mathbf{J}\mathbf{J}^\top)^{-1}\mathbf{y}}\right)\right),$$

which concludes our proof.

## D  GENERALIZATION BOUND OF RNN-BASED METHOD

Recall that we compute the representation of node $v_i$ at time $t$ by applying a multi-step RNN onto a sequence of temporal events $\{\mathbf{v}_1, \ldots, \mathbf{v}_{L-1}\}$ that are constructed at the target node. The temporal event features $\mathbf{v}_\ell$ are pre-computed on the temporal graph.

**Representation computation.** The RNN has trainable parameters $\boldsymbol{\theta} = \{\mathbf{W}^{(1)}, \mathbf{W}^{(2)}, \mathbf{W}^{(3)}\}$ and $\alpha \in \{0, 1\}$ is binary hyper-parameter that controls whether a residual connection is used. Then, the final prediction on node $v_i$ is computed as $f_i(\boldsymbol{\theta}) = \mathbf{W}^{(4)} \mathbf{h}_{L-1}$, where the hidden representation $\mathbf{h}_{L-1}$ is recursively compute by

$$\mathbf{h}_\ell = \sigma\Big(\kappa\big(\mathbf{W}^{(1)} \mathbf{h}_{\ell-1} + \mathbf{W}^{(2)} \mathbf{x}_\ell\big)\Big) + \alpha \mathbf{h}_{\ell-1} \in \mathbb{R}^m.$$

Here $\sigma(\cdot)$ is the activation function, $\mathbf{h}_0 = \mathbf{0}_m$ is initialized as all-zero vector, and $\mathbf{x}_\ell = \mathbf{W}^{(0)} \mathbf{v}_\ell$. We normalize the hidden representation by $\kappa = 1/\sqrt{2}$ so that $\|\mathbf{h}_\ell\|_2^2$ does not grow exponentially with respect to the number of steps $L$. For weight parameters, we have $\mathbf{W}^{(1)}, \mathbf{W}^{(2)} \in \mathbb{R}^{m \times m}$, $\mathbf{W}^{(3)} \in \mathbb{R}^{1 \times m}$ as the trainable parameters, but $\mathbf{W}^{(0)} \in \mathbb{R}^{m \times d}$ is non-trainable.

**Gradient computation.** The gradient with respect to each weight matrix is computed by

$$\frac{\partial f_i(\boldsymbol{\theta})}{\partial \mathbf{W}^{(1)}} = \sum_{\ell=1}^{L-1} \kappa \left[\mathbf{W}^{(3)} \left(\kappa \mathbf{D}_{L-1} \mathbf{W}^{(1)} + \alpha \mathbf{I}_m\right) \ldots \left(\kappa \mathbf{D}_{\ell+1} \mathbf{W}^{(1)} + \alpha \mathbf{I}_m\right) \mathbf{D}_\ell\right]^\top \mathbf{h}_\ell^\top \in \mathbb{R}^{m \times m}$$

$$\frac{\partial f_i(\boldsymbol{\theta})}{\partial \mathbf{W}^{(2)}} = \sum_{\ell=1}^{L-1} \kappa \left[\mathbf{W}^{(3)} \left(\kappa \mathbf{D}_{L-1} \mathbf{W}^{(1)} + \alpha \mathbf{I}_m\right) \ldots \left(\kappa \mathbf{D}_{\ell+1} \mathbf{W}^{(1)} + \alpha \mathbf{I}_m\right) \mathbf{D}_\ell\right]^\top \mathbf{x}_\ell^\top \in \mathbb{R}^{m \times d}$$

$$\frac{\partial f_i(\boldsymbol{\theta})}{\partial \mathbf{W}^{(3)}} = [\mathbf{h}_{L-1}]^\top \in \mathbb{R}^{1 \times m}$$

### D.1  USEFUL LEMMAS

In the following, we show that if the input weights $\boldsymbol{\theta} = \{\mathbf{W}^{(1)}, \mathbf{W}^{(2)}, \mathbf{W}^{(3)}\}$ and $\widetilde{\boldsymbol{\theta}} = \{\widetilde{\mathbf{W}}^{(1)}, \widetilde{\mathbf{W}}^{(2)}, \widetilde{\mathbf{W}}^{(3)}\}$ are close, the output of RNN's hidden representation computed with $\boldsymbol{\theta}, \widetilde{\boldsymbol{\theta}}$ does not change too much.

> **Lemma 7.** *Let $\rho$ be the maximum Lipschitz constant of the activation function and $m$ is the hidden dimension. Then with $\omega = \mathcal{O}(1/(1 + 3\kappa\rho)^{(L-1)})$ and assuming $\widetilde{\boldsymbol{\theta}} \in \mathcal{B}(\boldsymbol{\theta}, \omega)$, we have $\|\widetilde{\mathbf{h}}_\ell - \mathbf{h}_\ell\|_2 = \mathcal{O}(1)$ with probability at least $1 - 2\ell \exp(-m/2) - \ell \exp(-\Omega(m))$.*

Please note that the smaller the distance $\omega$, the closer the representation $\|\widetilde{\mathbf{h}}_\ell - \mathbf{h}_\ell\|_2$. In particular, according to the proof of Lemma 7, we have $\|\widetilde{\mathbf{h}}_\ell - \mathbf{h}_\ell\|_2 \leq \mathcal{O}(\epsilon)$ by selecting $\omega = \mathcal{O}(\epsilon/(1 + 3\kappa\rho)^{(L-1)})$ for any small $\epsilon > 0$. This conclusion will be later used in Lemma 11.

*Proof of Lemma 7.* When $\ell = 1$, we have

$$\|\widetilde{\mathbf{h}}_1 - \mathbf{h}_1\|_2 = \left\|\sigma\left(\widetilde{\mathbf{W}}^{(1)} \mathbf{h}_0 + \widetilde{\mathbf{W}}^{(2)} \mathbf{x}_1\right) - \sigma\left(\mathbf{W}^{(1)} \mathbf{h}_0 + \mathbf{W}^{(2)} \mathbf{x}_1\right)\right\|_2$$

$$\underset{(a)}{\leq} \rho \|\widetilde{\mathbf{W}}^{(2)} - \mathbf{W}^{(2)}\|_2 \|\mathbf{x}_1\|_2$$

$$\underset{(b)}{\leq} \rho\omega = \mathcal{O}(1),$$

where the inequality (a) is due to the Lipschitz continuity of the activation function, the inequality (b) is due to $\omega$-neighborhood definition $\widetilde{\mathbf{W}} \in \mathcal{B}(\mathbf{W}, \omega)$, $\mathbf{h}_0 = \mathbf{0}_m$ is an all-zero vector and Assumption 1.

Similarly, when $\ell \in \{2, \ldots, L-1\}$, we have

$$
\begin{aligned}
\|\widetilde{\mathbf{h}}_\ell - \mathbf{h}_\ell\|_2 &\underset{(a)}{\leq} \left\| \sigma\left( \kappa \widetilde{\mathbf{W}}^{(1)} \widetilde{\mathbf{h}}_{\ell-1} + \kappa \widetilde{\mathbf{W}}^{(2)} \mathbf{x}_\ell \right) - \sigma\left( \kappa \mathbf{W}^{(1)} \mathbf{h}_{\ell-1} + \kappa \mathbf{W}^{(2)} \mathbf{x}_\ell \right) \right\|_2 + \|\widetilde{\mathbf{h}}_{\ell-1} - \mathbf{h}_{\ell-1}\|_2 \\
&\underset{(b)}{\leq} \rho\kappa \|\widetilde{\mathbf{W}}^{(2)} - \mathbf{W}^{(2)}\|_2 \|\mathbf{x}_\ell\|_2 + \rho\kappa \|\mathbf{W}^{(1)}\|_2 \|\widetilde{\mathbf{h}}_{\ell-1} - \mathbf{h}_{\ell-1}\|_2 \\
&\quad + \rho\kappa \|\widetilde{\mathbf{W}}^{(1)} - \mathbf{W}^{(1)}\|_2 \|\widetilde{\mathbf{h}}_{\ell-1}\|_2 + \|\widetilde{\mathbf{h}}_{\ell-1} - \mathbf{h}_{\ell-1}\|_2 \\
&\underset{(c)}{\leq} \left( 1 + \rho\kappa \|\mathbf{W}^{(1)}\|_2 \right) \|\widetilde{\mathbf{h}}_{\ell-1} - \mathbf{h}_{\ell-1}\|_2 + \rho\kappa\omega \|\mathbf{x}_\ell\|_2 + \rho\kappa\omega \|\widetilde{\mathbf{h}}_{\ell-1}\|_2,
\end{aligned}
$$

where the inequalities (a) and (c) are due to $\|\mathbf{A} + \mathbf{B}\|_2 \leq \|\mathbf{A}\|_2 + \|\mathbf{B}\|_2$, the inequalities (b) and (c) are due to the Lipschitz continuity of activation function and $\widetilde{\mathbf{W}} \in \mathcal{B}(\mathbf{W}, \omega)$.

By Proposition 19, we know that with probability at least $1 - 2\exp(-m/2)$ we have $\|\mathbf{W}^{(1)}\|_2 \leq 3$.

Meanwhile, by using similar proof strategy of Lemma 21, we know that with probability at least $1 - \exp(-\Omega(m))$ we have $\|\mathbf{h}_\ell\|_2 = \Theta(1)$.

Then, by combining the results above, we know that with probability at least $1 - 2\ell \exp(-m/2) - \ell \exp(-\Omega(m))$ we have

$$
\begin{aligned}
\|\widetilde{\mathbf{h}}_\ell - \mathbf{h}_\ell\|_2 &\leq (1 + 3\kappa\rho) \|\widetilde{\mathbf{h}}_{\ell-1} - \mathbf{h}_{\ell-1}\|_2 + \rho\omega \cdot \Theta(1) \\
&\leq (1 + 3\kappa\rho)^2 \|\widetilde{\mathbf{h}}_{\ell-2} - \mathbf{h}_{\ell-2}\|_2 + \rho\omega \cdot \Theta(1) \cdot (1 + (1 + 3\kappa\rho)) \\
&\leq \frac{(1 + 3\kappa\rho)^{\ell-1} - 1}{3\kappa\rho} \cdot \rho\omega \cdot \Theta(1),
\end{aligned}
$$

By setting $\omega = 1/(1 + 3\kappa\rho)^{L-1}$ we have the above equation upper bounded by $\mathcal{O}(1)$. $\qquad \square$

Then, in the next lemma, we show that if the initialization of two sets of weight parameters are close, the neural network output $f_i(\boldsymbol{\theta})$ is almost linear with respect to its weight parameters.

---

**Lemma 8.** *Let $\boldsymbol{\theta}, \widetilde{\boldsymbol{\theta}} \in \mathcal{B}(\boldsymbol{\theta}_0, \omega)$ with $\omega = \mathcal{O}\left(1/(1 + 3\kappa\rho)^{(L-1)}\right)$. Then, for any node $v_i \in \mathcal{V}$ in the graph, with probability at least $1 - 2(L-1)\exp(-m/2) - L\exp(-\Omega(m)) - 2/m$, we have*
$$
\epsilon_{lin} = |f_i(\widetilde{\boldsymbol{\theta}}) - f_i(\boldsymbol{\theta}) - \langle \nabla f_i(\boldsymbol{\theta}), \widetilde{\boldsymbol{\theta}} - \boldsymbol{\theta} \rangle| = \mathcal{O}(1),
$$
*where $f_i(\boldsymbol{\theta})$ is the prediction on the sequence of sampled temporal event starting from the node $v_i$.*

---

Please note that the smaller the distance $\omega$, the more the model output close to linear. In particular, according to the proof of Lemma 8 and proof of Lemma 7, we have $\epsilon_{\text{lin}} \leq \mathcal{O}(\epsilon)$ by selecting $\omega = \mathcal{O}\left(\epsilon/(1 + 3\kappa\rho)^{(L-1)}\right)$ for any small $\epsilon > 0$. This conclusion will be later used in Lemma 11.

*Proof of Lemma 8.* According to the forward and backward propagation rules as we recapped at the beginning of this section, we have

$$|f_i(\widetilde{\boldsymbol{\theta}}) - f_i(\boldsymbol{\theta}) - \langle \nabla f_i(\boldsymbol{\theta}), \widetilde{\boldsymbol{\theta}} - \boldsymbol{\theta}\rangle|$$

$$\leq \underbrace{\left|\widetilde{\mathbf{W}}^{(3)}(\widetilde{\mathbf{h}}_{L-1} - \mathbf{h}_{L-1})\right|}_{(a)}$$

$$+ \underbrace{\kappa \sum_{\ell=1}^{L-1} \|\mathbf{W}^{(3)}\|_2 \left\|\left(\kappa\mathbf{D}_{L-1}\mathbf{W}^{(1)} + \alpha\mathbf{I}_m\right)\dots\left(\kappa\mathbf{D}_{\ell+1}\mathbf{W}^{(1)} + \alpha\mathbf{I}_m\right)\right\|_2 \|\mathbf{D}_\ell\|_2 \|\widetilde{\mathbf{W}}^{(1)} - \mathbf{W}^{(1)}\|_2 \|\mathbf{h}_\ell\|_2}_{(b)}$$

$$+ \underbrace{\kappa \sum_{\ell=1}^{L-1} \|\mathbf{W}^{(3)}\|_2 \left\|\left(\kappa\mathbf{D}_{L-1}\mathbf{W}^{(1)} + \alpha\mathbf{I}_m\right)\dots\left(\kappa\mathbf{D}_{\ell+1}\mathbf{W}^{(1)} + \alpha\mathbf{I}_m\right)\right\|_2 \|\mathbf{D}_\ell\|_2 \|\widetilde{\mathbf{W}}^{(1)} - \mathbf{W}^{(1)}\|_2 \|\mathbf{x}_\ell\|_2}_{(c)}.$$

By Lemma 7 and Lemma 20, we know that the term (a) in the above equation could be upper bounded by

$$(a) \leq \sqrt{2} \cdot \mathcal{O}(1)$$

Besides, since the derivative of activation function is bounded, we have $\|\mathbf{D}_\ell\|_2 \leq \rho$, $\forall \ell \in [L-1]$.

Therefore, we know that $\|\mathbf{D}_\ell\mathbf{W}^{(i)} + \alpha\mathbf{I}_m\|_2 \leq 3\rho + 1$ for any $i \in \{1, 2, 3\}$ and we can upper bound the term (b) in the above equation by

$$(b), (c) \leq \left(1 + (1 + 3\kappa\rho) + (1 + 3\kappa\rho)^2 + \dots + (1 + 3\kappa\rho)^{L-2}\right) \cdot \sqrt{2}\kappa\rho\omega \cdot \mathcal{O}(1)$$

$$= \frac{(1 + 3\kappa\rho)^{L-1} - 1}{3\kappa\rho} \cdot \sqrt{2}\kappa\rho\omega \cdot \mathcal{O}(1).$$

As a result, we have

$$|f_i(\widetilde{\boldsymbol{\theta}}) - f_i(\boldsymbol{\theta}) - \langle \nabla f_i(\boldsymbol{\theta}), \widetilde{\boldsymbol{\theta}} - \boldsymbol{\theta}\rangle| \leq \sqrt{2} \cdot \mathcal{O}(1) + 2 \times \frac{(1 + 3\kappa\rho)^{L-1} - 1}{3\kappa\rho} \cdot \sqrt{2}\kappa\rho\omega \cdot \mathcal{O}(1)$$

$$\leq \mathcal{O}(1),$$

where the last inequality holds by selecting $\omega = 1/(1 + 3\kappa\rho)^{L-1}$. $\qquad\square$

Let us define the logistic regression objective function as

$$\text{loss}_i(\boldsymbol{\theta}) = \psi(y_i f_i(\boldsymbol{\theta})), \psi(x) = \log(1 + \exp(-x)).$$

Then, the following lemma shows that $\text{loss}_i(\boldsymbol{\theta})$ is almost a convex function of $\boldsymbol{\theta}$ if the initialization of two sets of parameters are close.

---

**Lemma 9.** *Let $\boldsymbol{\theta}, \widetilde{\boldsymbol{\theta}} \in \mathcal{B}(\boldsymbol{\theta}_0, \omega)$ with $\omega = 1/(1 + 3\kappa\rho)^{L-1}$, it holds that*

$$loss_i(\widetilde{\boldsymbol{\theta}}) \geq loss_i(\boldsymbol{\theta}) + \langle \nabla_\theta loss_i(\boldsymbol{\theta}), \widetilde{\boldsymbol{\theta}} - \boldsymbol{\theta}\rangle - \epsilon_{lin}$$

*with probability at least $1 - 2(L-1)\exp(-m/2) - L\exp(-\Omega(m)) - 2/m$, where*

$$\epsilon_{lin} = |f_i(\widetilde{\boldsymbol{\theta}}) - f_i(\boldsymbol{\theta}) - \langle \nabla f_i(\boldsymbol{\theta}), \widetilde{\boldsymbol{\theta}} - \boldsymbol{\theta}\rangle|$$

*according to Lemma 8.*

---

*Proof of Lemma 9.* The proof follows the proof of Lemma 3. $\qquad\square$

Moreover, by the gradient computation, we know that the gradient of the neural network function can be upper bounded.

**Lemma 10.** *For any node $v_i \in \mathcal{V}$, with probability at least $1 - 2(L - \ell) \exp(-m/2) - \ell \exp(-\Omega(m)) - 2/m$, it holds that for $\ell = 1, 2, 3$*

$$\left\| \frac{\partial f_i(\boldsymbol{\theta})}{\partial \mathbf{W}^{(\ell)}} \right\|_2, \left\| \frac{\partial loss_i(\boldsymbol{\theta})}{\partial \mathbf{W}^{(\ell)}} \right\|_2 \leq \Theta\left((1 + 3\kappa\rho)^{L-1}\right).$$

*Proof of Lemma 10.* The $\ell_2$-norm of the gradient with respect to $\mathbf{W}^{(3)}$ is upper bounded by

$$\left\| \frac{\partial f_i(\boldsymbol{\theta})}{\partial \mathbf{W}^{(3)}} \right\|_2 = \|\mathbf{h}_{L-1}\|_2 = \Theta(1).$$

The $\ell_2$-norm of the gradient with respect to $\mathbf{W}^{(1)}$ is upper bounded by

$$
\begin{aligned}
\left\| \frac{\partial f_i(\boldsymbol{\theta})}{\partial \mathbf{W}^{(1)}} \right\|_2 &\leq \sum_{\ell=1}^{L} \kappa \left\| \left( \mathbf{W}^{(3)} \left(\kappa \mathbf{D}_{L-1} \mathbf{W}^{(1)} + \alpha \mathbf{I}_m\right) \ldots \left(\kappa \mathbf{D}_{\ell+1} \mathbf{W}^{(1)} + \alpha \mathbf{I}_m\right) \mathbf{D}_\ell \right)^\top (\mathbf{h}_\ell)^\top \right\|_2 \\
&\leq \left( 1 + (1 + 3\kappa\rho) + (1 + 3\kappa\rho)^2 + \ldots + (1 + 3\kappa\rho)^{L-2} \right) \cdot \sqrt{2}\kappa\rho \cdot \mathcal{O}(1) \\
&= \frac{(1 + 3\kappa\rho)^{L-1} - 1}{3\kappa\rho} \cdot \sqrt{2}\kappa\rho \cdot \mathcal{O}(1) \\
&= \mathcal{O}\left((1 + 3\kappa\rho)^{L-1}\right).
\end{aligned}
$$

The $\ell_2$-norm of the gradient with respect to $\mathbf{W}^{(2)}$ is upper bounded by

$$
\begin{aligned}
\left\| \frac{\partial f_i(\boldsymbol{\theta})}{\partial \mathbf{W}^{(2)}} \right\|_2 &\leq \sum_{\ell=1}^{L} \kappa \left\| \left( \mathbf{W}^{(3)} \left(\kappa \mathbf{D}_{L-1} \mathbf{W}^{(1)} + \alpha \mathbf{I}_m\right) \ldots \left(\kappa \mathbf{D}_{\ell+1} \mathbf{W}^{(1)} + \alpha \mathbf{I}_m\right) \mathbf{D}_\ell \right)^\top (\mathbf{x}_\ell)^\top \right\|_2 \\
&\leq \left( 1 + (1 + 3\kappa\rho) + (1 + 3\kappa\rho)^2 + \ldots + (1 + 3\kappa\rho)^{L-2} \right) \cdot \sqrt{2}\kappa\rho \cdot \mathcal{O}(1) \\
&= \frac{(1 + 3\kappa\rho)^{L-1} - 1}{3\kappa\rho} \cdot \sqrt{2}\kappa\rho \cdot \mathcal{O}(1) \\
&= \mathcal{O}\left((1 + 3\kappa\rho)^{L-1}\right).
\end{aligned}
$$

Moreover, since $|\psi'(y_i f_i(\boldsymbol{\theta})) \cdot y_i| \leq 1$, we have

$$\left\| \frac{\partial loss_i(\boldsymbol{\theta})}{\partial \mathbf{W}^{(\ell)}} \right\|_2 = |\psi'(y_i f_i(\boldsymbol{\theta})) \cdot y_i| \cdot \left\| \frac{\partial f_i(\boldsymbol{\theta})}{\partial \mathbf{W}^{(\ell)}} \right\|_2 \leq \left\| \frac{\partial f_i(\boldsymbol{\theta})}{\partial \mathbf{W}^{(\ell)}} \right\|_2.$$

$\square$

In the following, we show that the cumulative loss can be upper bounded under small changes on the weight parameters.

**Lemma 11.** *For any $\epsilon, \delta, R > 0$, there exists*

$$m^\star = \mathcal{O}\left( \frac{(1 + 3\kappa\rho)^{4(L-1)} L^2 R^4}{4\epsilon^4} \right) \log(1/\delta),$$

*such that if $m \geq m^\star$, then with probability at least $1 - \delta$ over the randomness of $\boldsymbol{\theta}_0$, for any $\mathcal{B}(\boldsymbol{\theta}_0, Rm^{-1/2})$, with $\eta = \frac{\epsilon}{mL(1+3\kappa\rho)^{2(L-1)}}$ and $N = \frac{L^2 R^2 (1+3\kappa\rho)^{2(L-1)}}{2\epsilon^2}$, the cumulative loss can be upper bounded by*

$$\frac{1}{N} \sum_{i=1}^{N} loss_i(\boldsymbol{\theta}_{i-1}) \leq \frac{1}{N} \sum_{i=1}^{N} loss_i(\boldsymbol{\theta}^\star) + \mathcal{O}(\epsilon).$$

*Proof of Lemma 11.* Let us define $\boldsymbol{\theta}^{\star}$ as the optimal solution that could minimize the cumulative loss over $N$ epochs, where at each epoch $\mathcal{S}_i(t) = \{\mathbf{v}_1, \ldots, \mathbf{v}_{L-1}\}$ is constructed to compute $f_i(\boldsymbol{\theta})$

$$\boldsymbol{\theta}^{\star} = \underset{\boldsymbol{\theta} \in \mathcal{B}(\boldsymbol{\theta}_0, \omega)}{\arg\min} \sum_{i=1}^{N} \mathrm{loss}_i(\boldsymbol{\theta}).$$

Without loss of generality, let us assume the epoch loss $\mathrm{loss}_i(\boldsymbol{\theta})$ is computed on the $i$-th node. Then, in the following, we try to show $\boldsymbol{\theta}_0, \boldsymbol{\theta}_1, \ldots, \boldsymbol{\theta}_N \in \mathcal{B}(\boldsymbol{\theta}_0, \omega)$, where $\omega = 1/(1 + 3\kappa\rho)^{L-1}$.

First of all, it is clear that $\boldsymbol{\theta}_0 \in \mathcal{B}(\boldsymbol{\theta}_0, \omega)$. Then, to show $\boldsymbol{\theta}_n \in \mathcal{B}(\boldsymbol{\theta}_0, \omega)$ for any $n \in \{1, \ldots, N\}$, we use our previous conclusion on the upper bound of gradient $\|\partial\mathrm{loss}_i(\boldsymbol{\theta})/\partial\mathbf{W}^{(\ell)}\| \leq \Theta(1 + 3\kappa\rho)^{L-1}$ in Lemma 10, we have

$$\|\mathbf{W}_n^{(\ell)} - \mathbf{W}_0^{(\ell)}\|_2 \leq \sum_{i=1}^{n} \|\mathbf{W}_i^{(\ell)} - \mathbf{W}_{i-1}^{(\ell)}\|_2$$
$$= \sum_{i=1}^{n} \eta \left\| \frac{\partial\mathrm{loss}_{i-1}(\boldsymbol{\theta}_{i-1})}{\partial\mathbf{W}_{i-1}^{(\ell)}} \right\|_2$$
$$\leq \Theta\left(\eta N(1 + 3\kappa\rho)^{L-1}\right).$$

By plugging in the choice of $\eta = \frac{\epsilon}{mL(1+3\kappa\rho)^{2(L-1)}}$ and $N = \frac{L^2 R^2 (1+3\kappa\rho)^{2(L-1)}}{2\epsilon^2}$, we have

$$\|\mathbf{W}_n^{(\ell)} - \mathbf{W}_0^{(\ell)}\|_2 \leq \Theta\left(\eta N(1 + 3\kappa\rho)^{L-\ell-1}\right)$$
$$= \Theta\left(\frac{\epsilon}{m(1+3\kappa\rho)^{2(L-1)}} \frac{LR^2(1+3\kappa\rho)^{2(L-1)}}{2\epsilon^2}(1+3\kappa\rho)^{L-1}\right)$$
$$= \Theta\left(\frac{LR^2}{2m\epsilon}(1+3\kappa\rho)^{L-1}\right).$$

Changing norm from $\ell_2$-norm to Frobenius-norm, we have

$$\|\mathbf{W}_n^{(\ell)} - \mathbf{W}_0^{(\ell)}\|_{\mathrm{F}} \leq \sqrt{m} \times \Theta\left(\frac{LR^2}{2m\epsilon}(1+3\kappa\rho)^{L-1}\right), \tag{3}$$

By plugging in the selection of hidden dimension $m$, we have

where the last inequality holds if

$$\|\mathbf{W}_n^{(\ell)} - \mathbf{W}_0^{(\ell)}\|_{\mathrm{F}} \leq \frac{\epsilon}{(1+3\kappa\rho)^{L-1}}$$

which means $\boldsymbol{\theta}_1, \ldots, \boldsymbol{\theta}_N, \boldsymbol{\theta}_\star \in \mathcal{B}(\boldsymbol{\theta}_0, \omega)$ where $\omega = \epsilon/(1 + 3\kappa\rho)^{L-1}$.

Then, our next step is to bound $\mathrm{loss}_i(\boldsymbol{\theta}_i) - \mathrm{loss}_i(\boldsymbol{\theta}_\star)$. By Lemma 3, we know that

$$\mathrm{loss}_{i+1}(\boldsymbol{\theta}_i) - \mathrm{loss}_{i+1}(\boldsymbol{\theta}_\star) \leq \langle \nabla_{\boldsymbol{\theta}}\mathrm{loss}_{i+1}(\boldsymbol{\theta}_i), \boldsymbol{\theta}_i - \boldsymbol{\theta}_\star \rangle + \epsilon_{\mathrm{lin}}$$
$$= \sum_{\ell=1}^{L} \left\langle \frac{\partial\mathrm{loss}_{i+1}(\boldsymbol{\theta}_i)}{\partial\mathbf{W}^{(\ell)}}, \mathbf{W}_i^{(\ell)} - \mathbf{W}_\star^{(\ell)} \right\rangle + \epsilon_{\mathrm{lin}}$$
$$= \frac{1}{\eta} \sum_{\ell=1}^{L} \left\langle \mathbf{W}_i^{(\ell)} - \mathbf{W}_{i+1}^{(\ell)}, \mathbf{W}_i^{(\ell)} - \mathbf{W}_\star^{(\ell)} \right\rangle + \epsilon_{\mathrm{lin}}$$
$$\leq \frac{1}{2\eta} \sum_{\ell=1}^{L} \left( \|\mathbf{W}_i^{(\ell)} - \mathbf{W}_{i+1}^{(\ell)}\|_{\mathrm{F}}^2 + \|\mathbf{W}_i^{(\ell)} - \mathbf{W}_\star^{(\ell)}\|_{\mathrm{F}}^2 - \|\mathbf{W}_{i+1}^{(\ell)} - \mathbf{W}_\star^{(\ell)}\|_{\mathrm{F}}^2 \right) + \epsilon_{\mathrm{lin}}.$$

Then, our next step is to upper bound each term on the right hand size of inequality.

(1) According to the proof of Lemma 8, we have $\epsilon_{\mathrm{lin}} \leq O(\epsilon)$ by selecting $\omega = \epsilon/(1 + 3\kappa\rho)^{L-1}$.

(2) Recall that $\|\mathbf{W}_i^{(\ell)} - \mathbf{W}_{i+1}^{(\ell)}\|_F^2$ could be upper bounded by

$$\|\mathbf{W}_i^{(\ell)} - \mathbf{W}_{i+1}^{(\ell)}\|_F^2 \leq \eta^2 \left\|\frac{\partial \text{loss}_i(\boldsymbol{\theta}_i)}{\partial \mathbf{W}^{(\ell)}}\right\|_F^2$$

$$\leq \eta^2 m \left\|\frac{\partial \text{loss}_i(\boldsymbol{\theta}_i)}{\partial \mathbf{W}^{(\ell)}}\right\|_2^2$$

$$\leq \Theta\left(m\eta^2(1+3\kappa\rho)^{2(L-\ell)}\right).$$

(3) By finite sum $\|\mathbf{W}_i^{(\ell)} - \mathbf{W}_\star^{(\ell)}\|_F^2 - \|\mathbf{W}_{i+1}^{(\ell)} - \mathbf{W}_\star^{(\ell)}\|_F^2$ for $i = 1, \ldots, N$, we have

$$\frac{1}{N}\sum_{i=1}^{N}(\|\mathbf{W}_i^{(\ell)} - \mathbf{W}_\star^{(\ell)}\|_F^2 - \|\mathbf{W}_{i+1}^{(\ell)} - \mathbf{W}_\star^{(\ell)}\|_F^2) = \frac{1}{N}\|\mathbf{W}_0^{(\ell)} - \mathbf{W}_\star^{(\ell)}\|_F^2 \underbrace{-\frac{1}{N}\|\mathbf{W}_{N+1}^{(\ell)} - \mathbf{W}_\star^{(\ell)}\|_F^2}_{\leq 0}$$

$$\leq \frac{R^2}{mN},$$

where the inequality is due to $\boldsymbol{\theta}_\star \in \mathcal{B}(\boldsymbol{\theta}_0, Rm^{-1/2})$.

Finally, by plugging the results above, we have

$$\frac{1}{N}\sum_{i=1}^{N}\text{loss}_i(\boldsymbol{\theta}_i) - \frac{1}{N}\sum_{i=1}^{N}\text{loss}_i(\boldsymbol{\theta}_\star) \leq \frac{R^2 L}{2m\eta N} + \sum_{\ell=1}^{L}\Theta\left(\frac{m\eta}{2}(1+3\kappa\rho)^{2(L-1)}\right) + \mathcal{O}(\epsilon).$$

By selecting $\eta = \frac{\epsilon}{mL(1+3\kappa\rho)^{2(L-1)}}$ and $N = \frac{L^2 R^2(1+3\kappa\rho)^{2(L-1)}}{2\epsilon^2}$, we have

$$\frac{R^2 L}{2m\eta N} = \frac{R^2 L}{2m} \times \frac{m(1+3\kappa\rho)^{2(L-1)}}{\epsilon} \times \frac{2\epsilon^2}{LR^2(3\rho\tau+1)^{2(L-1)}} = \epsilon,$$

$$\sum_{\ell=1}^{L}\Theta\left(\frac{m\eta}{2}(1+3\kappa\rho)^{2(L-\ell)}\right) \leq L \times \Theta\left(m\eta \cdot (1+3\kappa\rho)^{2(L-1)}\right)$$

$$= \Theta\left(m \cdot \frac{\epsilon}{m(1+3\kappa\rho)^{2(L-1)}}(1+3\kappa\rho)^{2(L-1)}\right) = \Theta(\epsilon).$$

Therefore, we have

$$\frac{1}{N}\sum_{i=1}^{N}\text{loss}_i(\boldsymbol{\theta}_i) - \frac{1}{N}\sum_{i=1}^{N}\text{loss}_i(\boldsymbol{\theta}_\star) \leq \mathcal{O}(\epsilon).$$

$\square$

By plugging in the selection of $\epsilon = \frac{LR(1+3\kappa\rho)^{L-1}}{\sqrt{2N}}$ to Lemma 11, we have

> **Corollary 2.** *For any $\delta, R > 0$, there exists $m^\star = \mathcal{O}(N^2/L^2)\log(1/\delta)$, such that if $m \geq m^\star$, then with probability at least $1 - \delta$ over the randomness of $\boldsymbol{\theta}_0$, for any $\boldsymbol{\theta}_\star \in \mathcal{B}(\boldsymbol{\theta}_0, Rm^{-1/2})$, with the selection of learning rate $\eta = \frac{R}{m\sqrt{2N}(1+3\kappa\rho)^{(L-1)}}$, the cumulative loss can be upper bounded by*
>
> $$\frac{1}{N}\sum_{i=1}^{N}loss_i(\boldsymbol{\theta}_{i-1}) \leq \frac{1}{N}\sum_{i=1}^{N}loss_i(\boldsymbol{\theta}^\star) + \mathcal{O}\left(\frac{LR(1+3\kappa\rho)^{L-1}}{\sqrt{N}}\right).$$

In the following, we present the expected 0-1 error bound of multi-step RNN, which consists of two terms: (1) the expected 0-1 error with the neural tangent random feature function and (2) the standard large-deviation error term.

**Lemma 12.** *For any $\delta \in (0, 1/e]$ and $R > 0$, there exists $m^\star = \mathcal{O}(N^2/L^2)\log(1/\delta)$, such that if $m \geq m^\star$, then with probability at least $1 - \delta$ over the randomness of $\boldsymbol{\theta}_0$ with the selection of learning rate $\eta = \frac{R}{m\sqrt{2N}(1+3\kappa\rho)^{(L-1)}}$, we have*

$$\mathbb{E}[\ell_N^{0-1}(\widetilde{\boldsymbol{\theta}})|\mathcal{D}_1^{N-1}]$$

$$\leq \frac{4}{N} \inf_{f \in \mathcal{F}(\boldsymbol{\theta}_0, R)} \sum_{i=1}^{N} \psi(y_i f_i) + \mathcal{O}\left(\sqrt{\frac{\log(1/\delta)}{N}}\right) + \mathcal{O}\left(\frac{LR(1+3\kappa\rho)^{(L-1)}}{\sqrt{N}}\right) + \Delta,$$

*where $\mathcal{F}(\boldsymbol{\theta}_0, R)$ is the neural tangent random feature function class, $\widetilde{\boldsymbol{\theta}}$ is uniformly selected from $\{\boldsymbol{\theta}_0, \dots, \boldsymbol{\theta}_{N-1}\}$, $\mathcal{D}_1^{N-1}$ is the sequence of data points sampled before the $N$-th iteration, and the expectation is computed on the uniform selection of weight parameters $\widetilde{\boldsymbol{\theta}}$ and the condition sampling of $N$-th iteration data examples.*

### D.2 PROOF OF THEOREM 3

In the following, we show that the expected error is bounded by $\sqrt{\mathbf{y}^\top(\mathbf{J}\mathbf{J}^\top)^{-1}\mathbf{y}}$ and is proportional to $L(3\rho\tau + 1)^{L-1}/\sqrt{N}$. Finally, to obtain the results in the form of Theorem 1, we just need to set $\kappa = \sqrt{2}$.

**Theorem 3** (Multi-steps RNN). *For any $\delta \in (0, 1/e]$ and $R > 0$, there exists $m^\star = \mathcal{O}(N^2/L^2)\log(1/\delta)$, such that if $m \geq m^\star$, then with probability at least $1 - \delta$ over the randomness of $\boldsymbol{\theta}_0$ with step size $\eta = \frac{R}{m\sqrt{2N}(1+3\rho/\sqrt{2})^{(L-1)}}$, we have*

$$\mathbb{E}[\ell_N^{0-1}(\widetilde{\boldsymbol{\theta}})|\mathcal{D}_1^{N-1}] \leq \mathcal{O}\left(\frac{LR((1+3\rho/\sqrt{2}))^{(L-1)}}{\sqrt{N}}\right) + \mathcal{O}\left(\sqrt{\frac{\log(1/\delta)}{N}}\right) + \Delta,$$

*where $R = \mathcal{O}(\sqrt{\mathbf{y}^\top(\mathbf{J}\mathbf{J}^\top)^{-1}\mathbf{y}})$, $\widetilde{\boldsymbol{\theta}}$ is uniformly selected from $\{\boldsymbol{\theta}_0, \dots, \boldsymbol{\theta}_{N-1}\}$, $\mathcal{D}_1^{N-1}$ is the sequence of data points sampled before the $N$-th iteration, and the expectation is computed on the uniform selection of weight parameters $\widetilde{\boldsymbol{\theta}}$ and the condition sampling of $N$-th iteration data examples.*

The proof of Theorem 3 follows the proof of Theorem 2 and Lemma 12.

# E    GENERALIZATION BOUND OF MEMORY-BASED METHOD

Recall that the representation of node $v_i$ at time $t$ is computed by applying weight parameters on the memory block $\mathbf{s}_i$. Let us define $\boldsymbol{\theta} = \{\mathbf{W}^{(1)}, \ldots, \mathbf{W}^{(4)}\}$ as the parameters to optimize.

**Representation computation.** The final prediction of node $v_i$ is computed by $f_i(\boldsymbol{\theta}) = \mathbf{W}^{(4)}\mathbf{s}_i(t)$ and $\mathbf{s}_i(t) \in \mathbb{R}^m$ is updated whenever node $v_i$ interacts with other nodes by

$$\mathbf{s}_i(t) = \sigma\Big(\kappa\big(\mathbf{W}^{(1)}\mathbf{s}_i^+(h_i^t) + \mathbf{W}^{(2)}\mathbf{s}_j^+(h_j^t) + \mathbf{W}^{(3)}\mathbf{e}_{ij}(t)\big)\Big),$$

where $\sigma(\cdot)$ is the activation function, $h_i^t$ is the latest timestamp that node $v_i$ interacts with other nodes before time $t$, $\mathbf{s}_i(0) = \mathbf{W}^{(0)}\mathbf{x}_i$, and $\mathbf{s}_i^+(t) = \text{StopGrad}(\mathbf{s}_i(t))$ is the memory block of node $v_i$ at time $t$. We normalize hidden representation by $\kappa = 1/\sqrt{3}$ so that $\|\mathbf{s}_i(t)\|_2^2$ does not grow exponentially with time $t$. For weight parameters, we have $\mathbf{W}^{(1)}, \mathbf{W}^{(2)} \in \mathbb{R}^{m \times m}$, $\mathbf{W}^{(3)} \in \mathbb{R}^{m \times d}$, $\mathbf{W}^{(4)} \in \mathbb{R}^m$ as the trainable parameters, but $\mathbf{W}^{(0)} \in \mathbb{R}^{m \times d}$ is non-trainable.

**Gradient computation.** Let us define $\mathbf{D}_i(t) = \text{diag}(\sigma'(\mathbf{z}_i(t))) \in \mathbb{R}^{m \times m}$ as a diagonal matrix. Then the gradient with respect to each weight matrix is computed by

$$\frac{\partial f_i^t(\boldsymbol{\theta})}{\partial \mathbf{W}^{(1)}} = \frac{\partial f_i^t(\boldsymbol{\theta})}{\partial \mathbf{s}_i(t)}\frac{\partial \mathbf{s}_i(t)}{\partial \mathbf{W}^{(1)}} = \kappa[\mathbf{s}_i^+(h_i(t))\mathbf{W}^{(4)}\mathbf{D}_i(t)]^\top \in \mathbb{R}^{m \times m},$$

$$\frac{\partial f_i^t(\boldsymbol{\theta})}{\partial \mathbf{W}^{(2)}} = \frac{\partial f_i^t(\boldsymbol{\theta})}{\partial \mathbf{s}_i(t)}\frac{\partial \mathbf{s}_i(t)}{\partial \mathbf{W}^{(2)}} = \kappa[\mathbf{s}_j^+(h_j(t))\mathbf{W}^{(4)}\mathbf{D}_i(t)]^\top \in \mathbb{R}^{m \times m},$$

$$\frac{\partial f_i^t(\boldsymbol{\theta})}{\partial \mathbf{W}^{(3)}} = \kappa[\mathbf{e}_{ij}(t)\mathbf{W}^{(4)}\mathbf{D}_i(t)]^\top \in \mathbb{R}^{m \times d},$$

$$\frac{\partial f_i^t(\boldsymbol{\theta})}{\partial \mathbf{W}^{(4)}} = [\mathbf{s}_i(t)]^\top \in \mathbb{R}^m.$$

## E.1    USEFUL LEMMAS

In the following, we first show that if the input weights are close, then given the same input data, the output of each neuron with any activation function does not change too much. Please notice that we do not need to consider the change of input memory blocks when using different weight parameters, i.e., the difference between $\mathbf{s}_i^+(t)$ and $\tilde{\mathbf{s}}_i^+(t)$. This is because this lemma is used to show the linearity of model output in the over-parameterized network after weight perturbation in Lemma 14, and it does not affect the memory blocks due to stop gradient.

> **Lemma 13.** *Let $\rho$ be the Lipschitz constant of the activation function and $m$ is the hidden dimension. Then with $\omega = \mathcal{O}(1/\rho)$ and assuming $\widetilde{\boldsymbol{\theta}} \in \mathcal{B}(\boldsymbol{\theta}, \omega)$, then we have $\|\widetilde{\mathbf{s}}_i(t) - \mathbf{s}_i(t)\|_2 = \mathcal{O}(1)$ with probability at least $1 - 6\exp(-m/2) - \exp(-\Omega(m))$.*

Please note that the smaller the distance $\omega$, the closer the representation $\|\widetilde{\mathbf{s}}_i(t) - \mathbf{s}_i(t)\|_2$. In particular, according to the proof of Lemma 13, we have $\|\widetilde{\mathbf{s}}_i(t) - \mathbf{s}_i(t)\|_2 \leq \mathcal{O}(\epsilon)$ by selecting $\omega = \mathcal{O}(\epsilon/\rho)$ for any small $\epsilon > 0$. This conclusion will be later used in Lemma 17.

*Proof of Lemma 13.* We can upper bound $\|\widetilde{\mathbf{s}}_i(t) - \mathbf{s}_i(t)\|_2$ by

$$\begin{aligned}
\|\widetilde{\mathbf{s}}_i(t) - \mathbf{s}_i(t)\|_2 &= \|\sigma(\kappa\widetilde{\mathbf{z}}_i(t)) - \sigma(\kappa\mathbf{z}_i(t))\|_2 \\
&\leq \kappa\rho\|\widetilde{\mathbf{z}}_i(t) - \mathbf{z}_i(t)\|_2 \\
&\leq \kappa\rho\|\widetilde{\mathbf{W}}^{(1)}\mathbf{s}_i^+(h_i(t)) - \mathbf{W}^{(1)}\mathbf{s}_i^+(h_i(t))\|_2 + \kappa\rho\|\widetilde{\mathbf{W}}^{(2)}\mathbf{s}_j^+(h_j(t)) - \mathbf{W}^{(2)}\mathbf{s}_j^+(h_j(t))\|_2 \\
&\quad + \kappa\rho\|\widetilde{\mathbf{W}}^{(3)}\mathbf{e}_{ij}(t) - \mathbf{W}^{(3)}\mathbf{e}_{ij}(t)\|_2 \\
&\leq \kappa\rho\|\mathbf{s}_i^+(h_i(t))\|_2\|\widetilde{\mathbf{W}}^{(1)} - \mathbf{W}^{(1)}\|_2 + \kappa\rho\|\mathbf{s}_j^+(h_j(t))\|_2\|\widetilde{\mathbf{W}}^{(2)} - \mathbf{W}^{(2)}\|_2 \\
&\quad + \kappa\rho\,\|\widetilde{\mathbf{W}}^{(3)} - \mathbf{W}^{(3)}\|_2\|\mathbf{e}_{ij}(t)\|_2.
\end{aligned}$$

Recall that $\widetilde{\boldsymbol{\theta}} \in \mathcal{B}(\boldsymbol{\theta}, \omega)$ and our assumption that $\|\mathbf{e}_{ij}(t)\|_2 = 1$, we have

$$\|\widetilde{\mathbf{s}}_i(t) - \mathbf{s}_i(t)\|_2 \leq \kappa\rho\omega \left(\|\mathbf{s}_i^+(h_i(t))\|_2 + \|\mathbf{s}_j^+(h_j(t))\|_2 + 1\right)$$
$$= \kappa\rho\omega \left(\|\mathbf{s}_i(h_i(t))\|_2 + \|\mathbf{s}_j(h_j(t))\|_2 + 1\right),$$

where the equality holds because $\mathbf{s}_i^+(t) = \text{StopGrad}(\mathbf{s}_i(t))$.

From Lemma 21, we know that $\|\mathbf{s}_i(h_i(t))\|_2 = \Theta(1)$, $\forall i \in [N]$. As a result, we have

$$\|\widetilde{\mathbf{s}}_i(t) - \mathbf{s}_i(t)\|_2 \leq \kappa\rho\omega \cdot (2\Theta(1) + 1).$$

Therefore, by selecting $\omega = 1/\rho$, we have $\|\mathbf{s}_i^+(t) - \mathbf{s}_i(t)\|_2 \leq \mathcal{O}(1)$.

$\square$

Then, in the next lemma, we show that if the initialization of two sets of weight parameters are close, the neural network output $f_i(\boldsymbol{\theta})$ is almost linear with respect to its weight parameters.

---

**Lemma 14.** *Let* $\boldsymbol{\theta}, \widetilde{\boldsymbol{\theta}} \in \mathcal{B}(\boldsymbol{\theta}_0, \omega)$ *with* $\omega = \mathcal{O}\left(1/\rho\right)$. *Then, for any node* $v_i \in \mathcal{V}$ *in the graph, with probability at least* $1 - 6\exp(-m/2) - \exp(-\Omega(m)) - 2/m$ *we have*

$$\epsilon_{lin} = |f_i^t(\widetilde{\boldsymbol{\theta}}) - f_i^t(\boldsymbol{\theta}) - \langle\nabla f_i^t(\boldsymbol{\theta}), \widetilde{\boldsymbol{\theta}} - \boldsymbol{\theta}\rangle| = \mathcal{O}(1),$$

*where* $f_i^t(\boldsymbol{\theta})$ *is the prediction on node* $v_i$ *at the* $t$-*th step.*

---

Please note that the smaller the distance $\omega$, the more the model output close to linear. In particular, according to the proof of Lemma 14 and the proof of Lemma 13, we have $\epsilon_{\text{lin}} \leq \mathcal{O}(\epsilon)$ by selecting $\omega = \mathcal{O}(\epsilon/\rho)$ for any small $\epsilon > 0$. This conclusion will be later used in Lemma 17.

*Proof of Lemma 14.* According to the forward and backward propagation rules as we recapped at the beginning of this section, we have

$$|f_i^t(\widetilde{\boldsymbol{\theta}}) - f_i^t(\boldsymbol{\theta}) - \langle\nabla f_i^t(\boldsymbol{\theta}), \widetilde{\boldsymbol{\theta}} - \boldsymbol{\theta}\rangle|$$
$$= \left|\widetilde{\mathbf{W}}^{(4)}\widetilde{\mathbf{s}}_i(t) - \mathbf{W}^{(4)}\mathbf{s}_i(t) - (\widetilde{\mathbf{W}}^{(4)} - \mathbf{W}^{(4)})\mathbf{s}_i(t)\right| + \left|\mathbf{W}^{(4)}\mathbf{D}_i(t)(\widetilde{\mathbf{W}}^{(1)} - \mathbf{W}^{(1)})\mathbf{s}_i^+(h_i(t))\right|$$
$$+ \left|\mathbf{W}^{(4)}\mathbf{D}_i(t)(\widetilde{\mathbf{W}}^{(2)} - \mathbf{W}^{(2)})\mathbf{s}_j^+(h_j(t))\right| + \left|\mathbf{W}^{(4)}\mathbf{D}_i(t)(\widetilde{\mathbf{W}}^{(3)} - \mathbf{W}^{(3)})\mathbf{e}_{ij}(t)\right|$$
$$= \left|\widetilde{\mathbf{W}}^{(4)}(\widetilde{\mathbf{s}}_i(t) - \mathbf{s}_i(t))\right| + \kappa\left|\mathbf{W}^{(4)}\mathbf{D}_i(t)(\widetilde{\mathbf{W}}^{(1)} - \mathbf{W}^{(1)})\mathbf{s}_i^+(h_i(t))\right|$$
$$+ \kappa\left|\mathbf{W}^{(4)}\mathbf{D}_i(t)(\widetilde{\mathbf{W}}^{(2)} - \mathbf{W}^{(2)})\mathbf{s}_j^+(h_j(t))\right| + \kappa\left|\mathbf{W}^{(4)}\mathbf{D}_i(t)(\widetilde{\mathbf{W}}^{(3)} - \mathbf{W}^{(3)})\mathbf{e}_{ij}(t)\right|$$
$$\leq \sqrt{2}\|\widetilde{\mathbf{s}}_i(t) - \mathbf{s}_i(t)\|_2 + \sqrt{2}\kappa\|\mathbf{D}_i(t)\|_2\|\widetilde{\mathbf{W}}^{(1)} - \mathbf{W}^{(1)}\|_2\|\mathbf{s}_i(h_i(t))\|_2$$
$$+ \sqrt{2}\kappa\|\mathbf{D}_i(t)\|_2\|\widetilde{\mathbf{W}}^{(2)} - \mathbf{W}^{(2)}\|_2\|\mathbf{s}_j(h_j(t))\|_2 + \sqrt{2}\kappa\|\mathbf{D}_i(t)\|_2\|\widetilde{\mathbf{W}}^{(3)} - \mathbf{W}^{(3)}\|_2\|\mathbf{e}_{ij}(t)\|_2.$$

Since the derivative of activation function is bounded, we have $\|\mathbf{D}_i(t)\|_2 \leq \rho$ and therefore

$$|f_i^t(\widetilde{\boldsymbol{\theta}}) - f_i^t(\boldsymbol{\theta}) - \langle\nabla f_i^t(\boldsymbol{\theta}), \widetilde{\boldsymbol{\theta}} - \boldsymbol{\theta}\rangle| \leq \sqrt{2}\|\widetilde{\mathbf{s}}_i(t) - \mathbf{s}_i(t)\|_2 + \sqrt{2}\kappa\rho\|\widetilde{\mathbf{W}}^{(1)} - \mathbf{W}^{(1)}\|_2\|\mathbf{s}_i(h_i(t))\|_2$$
$$+ \sqrt{2}\kappa\rho\|\widetilde{\mathbf{W}}^{(2)} - \mathbf{W}^{(2)}\|_2\|\mathbf{s}_j(h_j(t))\|_2 + \sqrt{2}\kappa\rho\|\widetilde{\mathbf{W}}^{(3)} - \mathbf{W}^{(3)}\|_2\|\mathbf{e}_{ij}(t)\|_2$$
$$\overset{(s)}{=} \sqrt{2}\|\widetilde{\mathbf{s}}_i(t) - \mathbf{s}_i(t)\|_2 + \sqrt{2}\kappa\rho\|\widetilde{\mathbf{W}}^{(1)} - \mathbf{W}^{(1)}\|_2\Theta(1)$$
$$+ \sqrt{2}\kappa\rho\|\widetilde{\mathbf{W}}^{(2)} - \mathbf{W}^{(2)}\|_2\Theta(1) + \sqrt{2}\kappa\rho\|\widetilde{\mathbf{W}}^{(3)} - \mathbf{W}^{(3)}\|_2,$$

where the equality (a) is due to $\|\mathbf{s}_i(h_i(t))\|_2 = \|\mathbf{s}_i(h_i(t))\|_2 = \Theta(1)$.

By Lemma 13 that $\|\widetilde{\mathbf{s}}_i(t) - \mathbf{s}_i(t)\|_2 \leq \mathcal{O}(1)$, we know

$$|f_i^t(\widetilde{\boldsymbol{\theta}}) - f_i^t(\boldsymbol{\theta}) - \langle\nabla f_i^t(\boldsymbol{\theta}), \widetilde{\boldsymbol{\theta}} - \boldsymbol{\theta}\rangle| \leq \mathcal{O}(1).$$

$\square$

Let us define

$$\text{loss}_i(\boldsymbol{\theta}) = \psi(y_i f_i(\boldsymbol{\theta})), \psi(x) = \log(1 + \exp(-x)).$$

Then, the following lemma shows that $\text{loss}_i(\boldsymbol{\theta})$ is almost a convex function of $\boldsymbol{\theta}$ for any $i \in [N]$ if the initialization of two sets of parameters are close.

---

**Lemma 15.** *Let $\boldsymbol{\theta}, \widetilde{\boldsymbol{\theta}} \in \mathcal{B}(\boldsymbol{\theta}_0, \omega)$ with $\omega = \mathcal{O}(1/\rho)$ for any $i \in [N]$, it holds that*

$$loss_i(\widetilde{\boldsymbol{\theta}}) \geq loss_i(\boldsymbol{\theta}) + \langle \nabla_\theta loss_i(\boldsymbol{\theta}), \widetilde{\boldsymbol{\theta}} - \boldsymbol{\theta} \rangle - \epsilon_{lin},$$

*with probability at least $1 - 6 \exp(-m/2) - \exp(-\Omega(m)) - 2/m$ where*

$$\epsilon_{lin} = |f_i^t(\widetilde{\boldsymbol{\theta}}) - f_i^t(\boldsymbol{\theta}) - \langle \nabla f_i^t(\boldsymbol{\theta}), \widetilde{\boldsymbol{\theta}} - \boldsymbol{\theta} \rangle|$$

*according to Lemma 14*

---

*Proof of Lemma 15.* The proof is the same as the proof of Lemma 3. $\square$

Moreover, by the gradient computation, we know that the gradient of the neural network function can be upper bounded.

---

**Lemma 16.** *For any $i \in [N]$ with probability at least $1 - 6 \exp(-m/2) - \exp(-\Omega(m)) - 2/m$, it holds that*

$$\left\| \frac{\partial f_i(\boldsymbol{\theta})}{\partial \mathbf{W}^{(k)}} \right\|_2, \left\| \frac{\partial loss_i(\boldsymbol{\theta})}{\partial \mathbf{W}^{(k)}} \right\|_2 \leq \begin{cases} \Theta\left(\sqrt{2}\kappa\rho\right) & \text{if } k = 1, 2 \\ \sqrt{2}\kappa\rho & \text{if } k = 3 \\ \Theta\left(1\right) & \text{if } k = 4 \end{cases}$$

---

*Proof of Lemma 16.* Recall the definition of the gradient

$$\left\| \frac{\partial f_i^t(\boldsymbol{\theta})}{\partial \mathbf{W}^{(1)}} \right\|_2 = \kappa \left\| \mathbf{s}_i^+(h_i(t))\mathbf{W}^{(4)}\mathbf{D}_i(t) \right\|_2 \leq \sqrt{2}\kappa\rho \cdot \Theta(1),$$

$$\left\| \frac{\partial f_i^t(\boldsymbol{\theta})}{\partial \mathbf{W}^{(2)}} \right\|_2 = \kappa \left\| \mathbf{s}_j^+(h_j(t))\mathbf{W}^{(4)}\mathbf{D}_i(t) \right\|_2 \leq \sqrt{2}\kappa\rho \cdot \Theta(1),$$

$$\left\| \frac{\partial f_i^t(\boldsymbol{\theta})}{\partial \mathbf{W}^{(3)}} \right\|_2 = \kappa \left\| \mathbf{e}_{ij}(t)\mathbf{W}^{(4)}\mathbf{D}_i(t) \right\|_2 \leq \sqrt{2}\kappa\rho,$$

$$\left\| \frac{\partial f_i^t(\boldsymbol{\theta})}{\partial \mathbf{W}^{(4)}} \right\|_2 = \left\| \mathbf{s}_i(t) \right\|_2 = \Theta(1).$$

By the chain rule, we know that

$$\left\| \frac{\partial \text{loss}_i(\boldsymbol{\theta})}{\partial \mathbf{W}^{(k)}} \right\|_2 = \left\| \frac{\partial \text{loss}_i(\boldsymbol{\theta})}{\partial f_i^t(\boldsymbol{\theta})} \frac{\partial f_i^t(\boldsymbol{\theta})}{\partial \mathbf{W}^{(k)}} \right\|_2$$

$$\leq |\psi'(y_i f_i^t(\boldsymbol{\theta}))| \left\| \frac{\partial f_i^t(\boldsymbol{\theta})}{\partial \mathbf{W}^{(k)}} \right\|_2$$

$$\leq \left\| \frac{\partial f_i^t(\boldsymbol{\theta})}{\partial \mathbf{W}^{(k)}} \right\|_2,$$

where the last inequality holds because $|\psi'(y_i f_i^t(\boldsymbol{\theta}))| \leq 1$.

$\square$

In the following, we show that the cumulative loss can be upper bounded under small changes on the weight parameters.

**Lemma 17.** *For any $\epsilon, \delta, R > 0$, there exists*

$$m^\star = \mathcal{O}\left(\frac{4\rho^4 R^4}{\epsilon^4}\right)\log(1/\delta),$$

*such that if $m \geq m^\star$, then with probability at least $1 - \delta$ over the randomness of $\boldsymbol{\theta}_0$, for any $\boldsymbol{\theta}_\star \in \mathcal{B}(\boldsymbol{\theta}_0, Rm^{-1/2})$, with $\eta = \frac{\epsilon}{4\rho^2 m}$ and $N = \frac{8\rho^2 R^2}{\epsilon^2}$, the cumulative loss can be upper bounded by*

$$\frac{1}{N}\sum_{i=1}^N loss_i(\boldsymbol{\theta}_{i-1}) \leq \frac{1}{N}\sum_{i=1}^N loss_i(\boldsymbol{\theta}^\star) + \mathcal{O}(\epsilon).$$

*Proof of Lemma 17.* Let us define $\boldsymbol{\theta}^\star$ as the optimal solution that could minimize the cumulative loss over $N$ epochs, where at each epoch only a single data point is used as defined in Algorithm 1

$$\boldsymbol{\theta}^\star = \arg\min_{\boldsymbol{\theta}\in\mathcal{B}(\boldsymbol{\theta}_0,\omega)} \sum_{i=1}^N loss_i(\boldsymbol{\theta}).$$

Without loss of generality, let us assume the epoch loss $loss_i(\boldsymbol{\theta})$ is computed on the $i$-th node. Then, in the following, we try to show $\boldsymbol{\theta}_0, \boldsymbol{\theta}_1, \ldots, \boldsymbol{\theta}_{N-1} \in \mathcal{B}(\boldsymbol{\theta}_0, \omega)$, where $\omega = \epsilon/\rho$.

First of all, it is clear that $\boldsymbol{\theta}_0 \in \mathcal{B}(\boldsymbol{\theta}_0, \omega)$. Then, to show $\boldsymbol{\theta}_n \in \mathcal{B}(\boldsymbol{\theta}_0, \omega)$ for any $n \in \{1, \ldots, N-1\}$ and $k \in \{1, \ldots, 4\}$, we use our previous conclusion on the upper bound of gradient in Lemma 16 and have

$$\begin{aligned}
\|\mathbf{W}_N^{(k)} - \mathbf{W}_0^{(k)}\|_2 &\leq \sum_{i=1}^N \|\mathbf{W}_i^{(k)} - \mathbf{W}_{i-1}^{(k)}\|_2 \\
&= \sum_{i=1}^N \eta\left\|\frac{\partial loss_{i-1}(\boldsymbol{\theta}_{i-1})}{\partial \mathbf{W}_{i-1}^{(k)}}\right\|_2 \\
&\leq \begin{cases} \Theta\left(\eta N\sqrt{2}\kappa\rho\right) & \text{if } k=1,2 \\ \eta N\sqrt{2}\kappa\rho & \text{if } k=3, \quad \leq \Theta(\eta N\rho). \\ \eta N & \text{if } k=4, \end{cases}
\end{aligned}$$

By plugging in the choice of $\eta = \frac{\epsilon}{4\rho^2 m}$ and $N = \frac{8\rho^2 R^2}{\epsilon^2}$, we have

$$\|\mathbf{W}_N^{(k)} - \mathbf{W}_0^{(k)}\|_2 \leq \frac{2R^2}{m\epsilon} \cdot \Theta(\rho).$$

Changing norm from $\ell_2$-norm to Frobenius-norm, we have

$$\|\mathbf{W}_N^{(\ell)} - \mathbf{W}_0^{(\ell)}\|_F \leq \frac{2R^2}{\sqrt{m}\epsilon} \cdot \Theta(\rho). \tag{4}$$

By plugging in the selection of hidden dimension $m$ and the selection of $\kappa = 1/\sqrt{3}$, we have

$$\|\mathbf{W}_N^{(\ell)} - \mathbf{W}_0^{(\ell)}\|_F \leq \epsilon/\rho,$$

which means $\boldsymbol{\theta}_0, \ldots, \boldsymbol{\theta}_{N-1} \in \mathcal{B}(\boldsymbol{\theta}_0, \omega)$ for $\omega = \epsilon/\rho$.

Then, our next step is to bound $\text{loss}_i(\boldsymbol{\theta}_i) - \text{loss}_i(\boldsymbol{\theta}_\star)$. By Lemma 15, we know that

$$\text{loss}_{i+1}(\boldsymbol{\theta}_i) - \text{loss}_{i+1}(\boldsymbol{\theta}_\star) \leq \langle \nabla_\theta \text{loss}_{i+1}(\boldsymbol{\theta}_i), \boldsymbol{\theta}_i - \boldsymbol{\theta}_\star \rangle + \epsilon_{\text{lin}}$$

$$= \sum_{k=1}^{4} \left\langle \frac{\partial \text{loss}_{i+1}(\boldsymbol{\theta}_i)}{\partial \mathbf{W}^{(k)}}, \mathbf{W}_i^{(k)} - \mathbf{W}_\star^{(k)}) \right\rangle + \epsilon_{\text{lin}}$$

$$= \frac{1}{\eta} \sum_{k=1}^{4} \left\langle \mathbf{W}_i^{(k)} - \mathbf{W}_{i+1}^{(k)}, \mathbf{W}_i^{(k)} - \mathbf{W}_\star^{(k)}) \right\rangle + \epsilon_{\text{lin}}$$

$$\leq \frac{1}{2\eta} \sum_{k=1}^{4} \left( \|\mathbf{W}_i^{(k)} - \mathbf{W}_{i+1}^{(k)}\|_{\text{F}}^2 + \|\mathbf{W}_i^{(k)} - \mathbf{W}_\star^{(k)}\|_{\text{F}}^2 - \|\mathbf{W}_{i+1}^{(k)} - \mathbf{W}_\star^{(k)}\|_{\text{F}}^2 \right) + \epsilon_{\text{lin}}.$$

Then, our next step is to upper bound each term on the right hand size of inequality.

(1) According to the proof of Lemma 14, we have $\epsilon_{\text{lin}} \leq \mathcal{O}(\epsilon)$ by selecting $\omega = \mathcal{O}(\epsilon/\rho)$.

(2) Recall that $\|\mathbf{W}_i^{(k)} - \mathbf{W}_{i+1}^{(k)}\|_{\text{F}}^2$ could be upper bounded by

$$\|\mathbf{W}_i^{(\ell)} - \mathbf{W}_{i+1}^{(\ell)}\|_{\text{F}}^2 \leq \eta^2 \left\| \frac{\partial \text{loss}_i(\boldsymbol{\theta}_i)}{\partial \mathbf{W}^{(\ell)}} \right\|_{\text{F}}^2$$

$$\leq \eta^2 m \left\| \frac{\partial \text{loss}_i(\boldsymbol{\theta}_i)}{\partial \mathbf{W}^{(\ell)}} \right\|_2^2$$

$$\leq m\eta^2 \times \Theta\left(\rho^2\right).$$

(3) By finite sum $\|\mathbf{W}_i^{(\ell)} - \mathbf{W}_\star^{(\ell)}\|_{\text{F}}^2 - \|\mathbf{W}_{i+1}^{(\ell)} - \mathbf{W}_\star^{(\ell)}\|_{\text{F}}^2$ for $i = 1, \ldots, N$, we have

$$\frac{1}{N} \sum_{i=1}^{N} (\|\mathbf{W}_i^{(\ell)} - \mathbf{W}_\star^{(\ell)}\|_{\text{F}}^2 - \|\mathbf{W}_{i+1}^{(\ell)} - \mathbf{W}_\star^{(\ell)}\|_{\text{F}}^2) = \frac{1}{N}\|\mathbf{W}_0^{(\ell)} - \mathbf{W}_\star^{(\ell)}\|_{\text{F}}^2 \underbrace{- \frac{1}{N}\|\mathbf{W}_{N+1}^{(\ell)} - \mathbf{W}_\star^{(\ell)}\|_{\text{F}}^2}_{\leq 0}$$

$$\leq \frac{R^2}{mN},$$

where the inequality is due to $\boldsymbol{\theta}_\star \in \mathcal{B}(\boldsymbol{\theta}_0, Rm^{-1/2})$.

Finally, by combining the results above, we have

$$\frac{1}{N} \sum_{i=1}^{N} \text{loss}_i(\boldsymbol{\theta}_{i-1}) - \frac{1}{N} \sum_{i=1}^{N} \text{loss}_i(\boldsymbol{\theta}_\star) \leq \frac{4R^2}{2m\eta N} + 4\Theta\left(\frac{m\eta}{2} \times \rho^2\right) + \mathcal{O}(\epsilon).$$

By selecting $\eta = \frac{\epsilon}{4\rho^2 m}$ and $N = \frac{8\rho^2 R^2}{\epsilon^2}$, we have

$$\frac{4R^2}{2m\eta N} = \frac{4R^2}{2m} \times \frac{4\rho^2 m}{\epsilon} \times \frac{\epsilon^2}{8\rho^2 R^2} = \epsilon$$

$$4\Theta\left(\frac{m\eta}{2} \times \rho^2\right) = \Theta\left(2m \cdot \frac{\epsilon}{4\rho^2 m} \cdot \rho^2\right)$$

and therefore

$$\frac{1}{N} \sum_{i=1}^{N} \text{loss}_i(\boldsymbol{\theta}_{i-1}) - \frac{1}{N} \sum_{i=1}^{N} \text{loss}_i(\boldsymbol{\theta}_\star) \leq \mathcal{O}(\epsilon)$$

$\square$

By plugging in the selection of $\epsilon = \frac{4\rho R}{\sqrt{2N}}$, we have the following corollary. Please note that the last term is $\mathcal{O}(4R\rho/\sqrt{N})$ instead of $\mathcal{O}(4R\rho/\sqrt{2N})$ because big-O was used in GNN's and RNN's result to hide the constant $\sqrt{2}$ in denominator.

**Corollary 3.** *For any* $\delta, R > 0$, *there exists* $m^\star = \mathcal{O}(N^2/L^2)\log(1/\delta)$, *such that if* $m \geq m^\star$, *then with probability at least* $1 - \delta$ *over the randomness of* $\boldsymbol{\theta}_0$, *for any* $\boldsymbol{\theta}_\star \in \mathcal{B}(\boldsymbol{\theta}_0, Rm^{-1/2})$, *with* $\eta = \frac{R}{m\sqrt{2N}\rho}$, *the cumulative loss can be upper bounded by*

$$\frac{1}{N}\sum_{i=1}^{N} loss_i(\boldsymbol{\theta}_{i-1}) \leq \frac{1}{N}\sum_{i=1}^{N} loss_i(\boldsymbol{\theta}^\star) + \mathcal{O}\left(\frac{4\rho R}{\sqrt{N}}\right).$$

In the following, we present the expected 0-1 error bound, which consists of two terms: (1) the expected 0-1 error with the neural tangent random feature function and (2) the standard large-deviation error term.

**Lemma 18.** *For any* $\delta \in (0, 1/e]$ *and* $R > 0$, *there exists* $m^\star = \mathcal{O}(N^2/L^2)\log(1/\delta)$, *such that if* $m \geq m^\star$, *then with probability at least* $1 - \delta$ *over the randomness of* $\boldsymbol{\theta}_0$ *with step size* $\eta = \frac{R}{m\sqrt{2N}\rho}$

$$\mathbb{E}_{\widetilde{\boldsymbol{\theta}}}[\ell_N^{0-1}(\widetilde{\boldsymbol{\theta}})|\mathcal{D}_1^{N-1}] \leq \frac{4}{N}\inf_{f \in \mathcal{F}(\boldsymbol{\theta}_0, R)}\sum_{i=1}^{N}\psi(y_i f_i) + \mathcal{O}\left(\sqrt{\frac{\log(1/\delta)}{N}}\right) + \mathcal{O}\left(\frac{4R\rho}{\sqrt{N}}\right) + \Delta,$$

*where* $\mathcal{F}(\boldsymbol{\theta}_0, R)$ *is the neural tangent random feature function class,* $\widetilde{\boldsymbol{\theta}}$ *is uniformly selected from* $\{\boldsymbol{\theta}_0, \ldots, \boldsymbol{\theta}_{N-1}\}$, $\mathcal{D}_1^{N-1}$ *is the sequence of data points sampled before the $N$-th iteration, and the expectation is computed on the uniform selection of weight parameters* $\widetilde{\boldsymbol{\theta}}$ *and the condition sampling of $N$-th iteration data examples.*

*Proof of Lemma 18.* The proof follows the proof of Lemma 6.

$\square$

## E.2 PROOF OF THEOREM 4

In the following, we show that the expected error is bounded by $\sqrt{\mathbf{y}^\top(\mathbf{J}\mathbf{J}^\top)^{-1}\mathbf{y}}$ and is proportional to $4\rho/\sqrt{N}$.

**Theorem 4** (Memory-based TGNN). *For any* $\delta \in (0, 1/e]$ *and* $R > 0$, *there exists* $m^\star = \mathcal{O}(N^2/L^2)\log(1/\delta)$ *such that if* $m \geq m^\star$, *then with probability at least* $1 - \delta$ *over the randomness of* $\boldsymbol{\theta}_0$ *with step size* $\eta = \frac{R}{m\sqrt{2N}\rho}$

$$\mathbb{E}_{\widetilde{\boldsymbol{\theta}}}[\ell_N^{0-1}(\widetilde{\boldsymbol{\theta}})|\mathcal{D}_1^{N-1}] \leq \mathcal{O}\left(\frac{4R\rho}{\sqrt{N}}\right) + \mathcal{O}\left(\sqrt{\frac{\log(1/\delta)}{N}}\right) + \Delta$$

*where* $R = \mathcal{O}(\sqrt{\mathbf{y}^\top(\mathbf{J}\mathbf{J}^\top)^{-1}\mathbf{y}})$, $\widetilde{\boldsymbol{\theta}}$ *is uniformly selected from* $\{\boldsymbol{\theta}_0, \ldots, \boldsymbol{\theta}_{N-1}\}$, $\mathcal{D}_1^{N-1}$ *is the sequence of data points sampled before the $N$-th iteration, and the expectation is computed on the uniform selection of weight parameters* $\widetilde{\boldsymbol{\theta}}$ *and the condition sampling of $N$-th iteration data examples.*

The proof of Theorem 4 follows the proof of Theorem 2.

## F    USEFUL LEMMAS

**Lemma 19.** *Let us define* $\mathbf{W}^{(\ell)} \in \mathbb{R}^{m \times m}$, $\forall \ell \in \{2, \ldots, L\}$, *where each element of* $\mathbf{W}^{(\ell)}$ *is initialized by* $[\mathbf{W}^{(\ell)}]_{ij} \sim \mathcal{N}(0, 1/m)$, $\forall i, j \in [m]$. *Then, with probability at least* $1 - 2 \exp(-m/2)$, *we have* $\|\mathbf{W}^{(\ell)}\|_2 \leq 3$.

*Proof.* According to Theorem 4.4.5 of Vershynin (2018), we know that given an $m \times n$ matrix $\mathbf{A}$ whose entries are independent standard normal variables, then for every $t \geq 0$, with probability at least $1 - 2 \exp(-t^2/2)$, we have $\lambda_{\max}(\mathbf{A}) \leq \sqrt{m} + \sqrt{n} + t$. We conclude the proof by choosing $t = \sqrt{m}$ and $n = m$. □

**Lemma 20.** *The* $\ell_2$*-norm of weight parameters in the last layer* $\mathbf{W}^{(L)}$ *is bounded by* $\sqrt{2}$, *i.e., we have* $\|\mathbf{W}^{(L)}\|_2 \leq \sqrt{2}$.

*Proof.* Please refer to the proof of Lemma 8 in Zhu et al. (2022). □

**Lemma 21.** *For any* $\ell \in \{0, 1, \ldots, L - 1\}$ *and* $\mathbf{x}_i \sim P_X$, *then for ReLU, LeakyReLU, Sigmoid, and Tanh, we have* $\|\mathbf{h}_i^{(\ell)}\|_2^2 = \Theta(1)$ *with probability at least* $1 - \ell \exp(-\Omega(m))$ *over* $\{\mathbf{W}^{(1)}, \ldots, \mathbf{W}^{(\ell)}\}$.

*Proof.* The proof follows the proof the Lemma 2 in Zhu et al. (2022) by expending their results from feed-forward network to graph neural network, which requires taking the node dependency into consideration.

We prove $\|\mathbf{h}_i^{(\ell)}\|_2^2 = \Theta(1)$ by showing the following two inequalities hold simultaneously

$$
\begin{aligned}
\mathbf{E}_{\mathbf{W}^{(\ell)}} \|\mathbf{h}_i^{(\ell)}\|_2^2 &\geq \min_{i \in \mathcal{V}} \mathbf{E}_{\mathbf{W}^{(\ell)}} \|\mathbf{h}_i^{(\ell)}\|_2^2 \geq \Omega(1) \\
\mathbf{E}_{\mathbf{W}^{(\ell)}} \|\mathbf{h}_i^{(\ell)}\|_2^2 &\leq \max_{i \in \mathcal{V}} \mathbf{E}_{\mathbf{W}^{(\ell)}} \|\mathbf{h}_i^{(\ell)}\|_2^2 \leq \mathcal{O}(1)
\end{aligned}
\tag{5}
$$

which implies $\mathbf{E}_{\mathbf{W}^{(\ell)}} \|\mathbf{h}_i^{(\ell)}\|_2^2 = \Theta(1)$.

Then by using Bernstein's inequality to the sum of $m$ *i.i.d.* random variables in $\|\mathbf{h}_i^{(\ell)}\|_2^2 = \sum_{i=1}^m (h_{ik}^{(\ell)})^2$, we have

$$
\frac{1}{2} \mathbf{E}_{\mathbf{W}^{(\ell)}} \left[ \|\mathbf{h}_i^{(\ell)}\|_2^2 \right] \leq \|\mathbf{h}_i^{(\ell)}\|_2^2 \leq \frac{3}{2} \mathbf{E}_{\mathbf{W}^{(\ell)}} \left[ \|\mathbf{h}_i^{(\ell)}\|_2^2 \right],
$$

which implies $\|\mathbf{h}_i^{(\ell)}\|_2^2 = \Theta(1)$.

According to Assumption 1, we know that $\|\mathbf{h}_i\|_2^2 = 1$ for all $i \in \mathcal{V}$, therefore the results hold for $\ell = 0$.

Then, we assume the result holds for $\ell - 1$, i.e.,

$$
\min_{i \in \mathcal{V}} \mathbf{E}_{\mathbf{W}^{(\ell)}} \|\mathbf{h}_i^{(\ell-1)}\|_2^2 \geq \Omega(1), \ \max_{i \in \mathcal{V}} \mathbf{E}_{\mathbf{W}^{(\ell)}} \|\mathbf{h}_i^{(\ell-1)}\|_2^2 \leq \mathcal{O}(1)
$$

with probability at least $1 - (\ell - 1) \exp(-\Omega(m))$.

Our goal is to show the result also holds for $\ell$ by conditioning on the event of $\mathbf{W}^{(1)}, \ldots, \mathbf{W}^{(\ell-1)}$ and studying the bound over $\mathbf{W}^{(\ell)}$.

For the ease of presentation, let us denote $\mathbf{w}_k$ as the $k$-th row of $\mathbf{W}^{(\ell)}$ where $\mathbf{w}_k \sim \mathcal{N}(0, \mathbf{I}_m/m)$ and $h_{ik}^{(\ell)}$ represents the $j$-th element of $\mathbf{h}_i^{(\ell)}$. By the forward propagation rule of GNN, we have

$$
\begin{aligned}
\mathbf{E}_{\mathbf{W}^{(\ell)}}\left[\|\mathbf{h}_i^{(\ell)}\|_2^2\right] &= \sum_{k=1}^{m} \mathbf{E}_{\mathbf{w}_k}\left[(h_{ik}^{(\ell)})^2\right] \\
&= \sum_{k=1}^{m} \mathbf{E}_{\mathbf{w}_k}\left[\left(\sigma_\ell(\langle \mathbf{w}_k, \widetilde{\mathbf{z}}_i^{(\ell-1)}\rangle) + \alpha_{\ell-1}[\mathbf{h}_i^{(\ell-1)}]_j\right)^2\right] \\
&= \sum_{k=1}^{m} \mathbf{E}_{\mathbf{w}_k}\left[\sigma_\ell^2(\langle \mathbf{w}_k, \widetilde{\mathbf{z}}_i^{(\ell-1)}\rangle)\right] + \alpha_{\ell-1}^2 \|\mathbf{h}_i^{(\ell-1)}\|_2^2 + \sum_{k=1}^{m} 2\alpha_{\ell-1}\mathbf{E}_{\mathbf{w}_k}\left[\sigma_\ell(\langle \mathbf{w}_k, \widetilde{\mathbf{z}}_i^{(\ell-1)}\rangle) \cdot h_{ij}^{(\ell-1)}\right] \\
&= m \cdot \mathbb{E}_{w\sim\mathcal{N}(0,\frac{1}{m}\|\widetilde{\mathbf{z}}_i^{(\ell-1)}\|_2^2)}[\sigma_\ell^2(w)] + \alpha_{\ell-1}^2\|\mathbf{h}_i^{(\ell-1)}\|_2^2 + 2\alpha_{\ell-1}\mathbb{E}_{w\sim\mathcal{N}(0,\frac{1}{m}\|\widetilde{\mathbf{z}}_i^{(\ell-1)}\|_2^2)}[\sigma_\ell(w)]\left(\sum_{k=1}^{m} h_{ik}^{(\ell-1)}\right).
\end{aligned}
$$

By Lemma 22, we know that if $\sigma_{\ell-1}(\cdot)$ is ReLU, LeakyReLU, Sigmoid, or Tanh, we have

$$
\begin{aligned}
m \cdot \mathbb{E}_{w\sim\mathcal{N}(0,\frac{1}{m}\|\widetilde{\mathbf{z}}_i^{(\ell-1)}\|_2^2)}[\sigma_\ell^2(w)] &= m \times \Theta\left(\frac{1}{m}\|\widetilde{\mathbf{z}}_i^{(\ell-1)}\|_2^2\right) \\
&= \Theta\left(\|\widetilde{\mathbf{z}}_i^{(\ell-1)}\|_2^2\right),
\end{aligned}
$$

which implies

$$
\begin{aligned}
\max_{i\in\mathcal{V}} m \cdot \mathbb{E}_{w\sim\mathcal{N}(0,\frac{1}{m}\|\widetilde{\mathbf{z}}_i^{(\ell-1)}\|_2^2)}[\sigma_\ell^2(w)] &\leq \max_{i\in\mathcal{V}} \mathcal{O}(\tau \cdot \|\mathbf{h}_i^{(\ell-1)}\|_2^2), \\
\min_{i\in\mathcal{V}} m \cdot \mathbb{E}_{w\sim\mathcal{N}(0,\frac{1}{m}\|\widetilde{\mathbf{z}}_i^{(\ell-1)}\|_2^2)}[\sigma_\ell^2(w)] &\geq \min_{i\in\mathcal{V}} \Omega(\tau' \cdot \|\mathbf{h}_i^{(\ell-1)}\|_2^2).
\end{aligned}
$$

If $\sigma_{\ell-1}(\cdot)$ is ReLU or LeakyReLU,

$$
\begin{aligned}
0 \leq \mathbb{E}_{w\sim\mathcal{N}(0,\frac{1}{m}\|\widetilde{\mathbf{z}}_i^{(\ell-1)}\|_2^2)}[\sigma_\ell(w)] &\leq \mathbb{E}_{w\sim\mathcal{N}(0,\frac{1}{m}\|\widetilde{\mathbf{z}}_i^{(\ell-1)}\|_2^2)}[\text{ReLU}(w)] \\
&= \frac{2}{5\sqrt{m}}\|\widetilde{\mathbf{z}}_i^{(\ell-1)}\|_2.
\end{aligned}
$$

Besides, according to the relationship between $\ell_1$ norm and $\ell_2$ norm, we have

$$
-\sqrt{m}\|\mathbf{h}_i^{(\ell-1)}\|_2 \leq \left(\sum_{k=1}^{m} h_{ik}^{(\ell-1)}\right) \leq \sqrt{m}\|\mathbf{h}_i^{(\ell-1)}\|_2.
$$

Therefore, we know that

$$
\begin{aligned}
-\frac{2}{5}\|\mathbf{h}_i^{(\ell-1)}\|_2\|\widetilde{\mathbf{z}}_i^{(\ell-1)}\|_2 &\leq \mathbb{E}_{w\sim\mathcal{N}(0,\frac{1}{m}\|\widetilde{\mathbf{z}}_i^{(\ell-1)}\|_2^2)}[\sigma_\ell(w)]\left(\sum_{k=1}^{m} h_{ik}^{(\ell-1)}\right) \\
&\leq \frac{2}{5}\|\mathbf{h}_i^{(\ell-1)}\|_2\|\widetilde{\mathbf{z}}_i^{(\ell-1)}\|_2,
\end{aligned}
$$

which implies

$$
\begin{aligned}
\max_{i\in\mathcal{V}} \mathbb{E}_{w\sim\mathcal{N}(0,\frac{1}{m}\|\widetilde{\mathbf{z}}_i^{(\ell-1)}\|_2^2)}[\sigma_\ell(w)]\left(\sum_{k=1}^{m} h_{ik}^{(\ell-1)}\right) &\leq \frac{2\tau}{5}\max_{i\in\mathcal{V}}\|\mathbf{h}_i^{(\ell-1)}\|_2^2, \\
\min_{i\in\mathcal{V}} \mathbb{E}_{w\sim\mathcal{N}(0,\frac{1}{m}\|\widetilde{\mathbf{z}}_i^{(\ell-1)}\|_2^2)}[\sigma_\ell(w)]\left(\sum_{k=1}^{m} h_{ik}^{(\ell-1)}\right) &\geq -\frac{2\tau}{5}\min_{i\in\mathcal{V}}\|\mathbf{h}_i^{(\ell-1)}\|_2^2.
\end{aligned}
$$

By plugging the results inside, we have

$$
\begin{aligned}
\max_{i\in\mathcal{V}} \mathbf{E}_{\mathbf{W}^{(\ell)}}\left[\|\mathbf{h}_i^{(\ell)}\|_2^2\right] &\leq \mathcal{O}\left(\tau \cdot \max_{j\in\mathcal{V}}\|\mathbf{h}_j^{(\ell-1)}\|_2^2\right) + \left(1 + \frac{4\tau}{5}\alpha_{\ell-1}\right)\max_{i\in\mathcal{V}}\|\mathbf{h}_i^{(\ell-1)}\|_2^2 \leq \mathcal{O}(1), \\
\min_{i\in\mathcal{V}} \mathbf{E}_{\mathbf{W}^{(\ell)}}\left[\|\mathbf{h}_i^{(\ell)}\|_2^2\right] &\geq \Omega\left(\tau' \cdot \min_{j\in\mathcal{V}}\|\mathbf{h}_j^{(\ell-1)}\|_2^2\right) + \left(1 - \frac{4\tau}{5}\alpha_{\ell-1}\right)\min_{i\in\mathcal{V}}\|\mathbf{h}_i^{(\ell-1)}\|_2^2 \geq \Omega(1),
\end{aligned}
$$

which implies $\mathbf{E}_{\mathbf{W}^{(\ell)}}\left[\|\mathbf{h}_i^{(\ell)}\|_2^2\right] = \Theta(1)$ if $\sigma_{\ell-1}(\cdot)$ is ReLU or LeakyReLU.

If $\sigma_{\ell-1}$ is Sigmoid or Tanh, we have

$$\mathbb{E}_{w \sim \mathcal{N}(0, \frac{1}{m}\|\widetilde{\mathbf{z}}_i^{(\ell-1)}\|_2^2)}[\sigma_\ell(w)] = 0.$$

By plugging the results inside, we have

$$\max_{i \in \mathcal{V}} \mathbf{E}_{\mathbf{W}^{(\ell)}}\left[\|\mathbf{h}_i^{(\ell)}\|_2^2\right] \leq \mathcal{O}\left(\tau \cdot \max_{j \in \mathcal{V}} \|\mathbf{h}_j^{(\ell-1)}\|_2^2\right) + \max_{i \in \mathcal{V}} \|\mathbf{h}_i^{(\ell-1)}\|_2^2 \leq \mathcal{O}(1),$$

$$\min_{i \in \mathcal{V}} \mathbf{E}_{\mathbf{W}^{(\ell)}}\left[\|\mathbf{h}_i^{(\ell)}\|_2^2\right] \geq \mathcal{O}\left(\tau' \cdot \min_{j \in \mathcal{V}} \|\mathbf{h}_j^{(\ell-1)}\|_2^2\right) + \min_{i \in \mathcal{V}} \|\mathbf{h}_i^{(\ell-1)}\|_2^2 \geq \Omega(1),$$

which implies $\mathbf{E}_{\mathbf{W}^{(\ell)}}\left[\|\mathbf{h}_i^{(\ell)}\|_2^2\right] = \Theta(1)$ if $\sigma_{\ell-1}(\cdot)$ is Sigmoid or Tanh. $\qquad\square$

---

**Lemma 22.** *Let us denote* $\mathbf{I}_m$ *as a identity matrix of size* $m \times m$.

$\mathbf{A}^{(1)} = \mathbf{G}^{(1)} = \mathbf{X}\mathbf{X}^\top,$

$\mathbf{A}^{(2)} = \mathbf{G}^{(2)} = 2\mathbb{E}_{\mathbf{w} \sim \mathcal{N}(\mathbf{0}, \mathbf{I}_d)}[\sigma_1(\sqrt{\widetilde{\mathbf{A}}^{(1)}}\mathbf{w})\sigma_1(\sqrt{\widetilde{\mathbf{A}}^{(1)}}\mathbf{w})^\top], \quad \widetilde{\mathbf{A}}^{(1)} = \mathbf{P}\sqrt{\mathbf{A}^{(1)}}(\mathbf{P}\sqrt{\mathbf{A}^{(1)}})^\top,$

$\mathbf{G}^{(\ell)} = 2\mathbb{E}_{\mathbf{w} \sim \mathcal{N}(\mathbf{0}, \mathbf{I}_m)}[\sigma_{\ell-1}(\sqrt{\widetilde{\mathbf{A}}^{(\ell-1)}}\mathbf{w})\sigma_{\ell-1}(\sqrt{\widetilde{\mathbf{A}}^{(\ell-1)}}\mathbf{w})^\top], \quad \widetilde{\mathbf{A}}^{(\ell-1)} = \mathbf{P}\sqrt{\mathbf{A}^{(\ell-1)}}(\mathbf{P}\sqrt{\mathbf{A}^{(\ell-1)}})^\top,$

$\mathbf{A}^{(\ell)} = \mathbf{G}^{(\ell)} + \alpha_{\ell-1} \cdot \mathbf{A}^{(\ell-1)}.$

*We know that* $G_{ii}^{(\ell)} = \Theta(\widetilde{A}_{ii}^{(\ell-1)})$, *where* $G_{ii}^{(\ell)}$ *is the* $i$-*th row and* $i$-*th column of* $\mathbf{G}^{(\ell)}$ *and* $\widetilde{A}_{ii}^{(\ell-1)}$ *is the* $i$-*th row and* $i$-*th column of* $\widetilde{\mathbf{A}}^{(\ell-1)}$.

---

*Proof.* **ReLU.** When $\sigma_{\ell-1}$ is ReLU, we have

$$\begin{aligned}
G_{ii}^{(\ell)} &= \mathbb{E}_{w \sim \mathcal{N}(0, \widetilde{A}_{ii}^{(\ell-1)})}[\sigma_{\ell-1}(w)^2] \\
&= \int_{-\infty}^{\infty} \frac{2}{\sqrt{2\pi \widetilde{A}_{ii}^{(\ell-1)}}} \exp\left(-\frac{x^2}{2\widetilde{A}_{ii}^{(\ell-1)}}\right) \max(0, x)^2 dx \\
&= \int_0^{\infty} \frac{2}{\sqrt{2\pi \widetilde{A}_{ii}^{(\ell-1)}}} \exp\left(-\frac{x^2}{2\widetilde{A}_{ii}^{(\ell-1)}}\right) x^2 dx \\
&= \widetilde{A}_{ii}^{(\ell-1)}.
\end{aligned}$$

**LeakyReLU.** When $\sigma_{\ell-1}$ is LeakyReLU, we have

$$\begin{aligned}
G_{ii}^{(\ell)} &= \mathbb{E}_{w \sim \mathcal{N}(0, \widetilde{A}_{ii}^{(\ell-1)})}[\sigma_{\ell-1}(w)^2] \\
&= \int_{-\infty}^{\infty} \frac{2}{\sqrt{2\pi \widetilde{A}_{ii}^{(\ell-1)}}} \exp\left(-\frac{x^2}{2\widetilde{A}_{ii}^{(\ell-1)}}\right) \max(\eta x, x)^2 dx \\
&= \int_0^{\infty} \frac{2}{\sqrt{2\pi \widetilde{A}_{ii}^{(\ell-1)}}} \exp\left(-\frac{x^2}{2\widetilde{A}_{ii}^{(\ell-1)}}\right) x^2 dx + \int_{-\infty}^0 \frac{2}{\sqrt{2\pi \widetilde{A}_{ii}^{(\ell-1)}}} \exp\left(-\frac{x^2}{2\widetilde{A}_{ii}^{(\ell-1)}}\right) \eta^2 x^2 dx \\
&= (1 + \eta^2)\widetilde{A}_{ii}^{(\ell-1)}.
\end{aligned}$$

**Sigmoid.** When $\sigma_{\ell-1}$ is Sigmoid, we can upper bound $G_{ii}^{(\ell)}$ by

$$
\begin{aligned}
G_{ii}^{(\ell)} &= \mathbb{E}_{w \sim \mathcal{N}(0, \widetilde{A}_{ii}^{(\ell-1)})}[\sigma_{\ell-1}(w)^2] \\
&= \int_{-\infty}^{\infty} \frac{2}{\sqrt{2\pi \widetilde{A}_{ii}^{(\ell-1)}}} \exp\left(-\frac{x^2}{2\widetilde{A}_{ii}^{(\ell-1)}}\right) \text{Sigmoid}(x)^2 dx \\
&\leq \int_{-\infty}^{\infty} \frac{2}{\sqrt{2\pi \widetilde{A}_{ii}^{(\ell-1)}}} \exp\left(-\frac{x^2}{2\widetilde{A}_{ii}^{(\ell-1)}}\right) \left(\frac{1}{4} - \exp(-x^2/4)\right) dx \\
&= \frac{1}{2} - \frac{1}{2\sqrt{1 + \widetilde{A}_{ii}^{(\ell-1)}/2}} \\
&\leq \widetilde{A}_{ii}^{(\ell-1)}/8
\end{aligned}
$$

and lower bound $G_{ii}^{(\ell)}$ by

$$
\begin{aligned}
G_{ii}^{(\ell)} &= \mathbb{E}_{w \sim \mathcal{N}(0, \widetilde{A}_{ii}^{(\ell-1)})}[\sigma_{\ell-1}(w)^2] \\
&= \int_{-\infty}^{\infty} \frac{2}{\sqrt{2\pi \widetilde{A}_{ii}^{(\ell-1)}}} \exp\left(-\frac{x^2}{2\widetilde{A}_{ii}^{(\ell-1)}}\right) \text{Sigmoid}(x)^2 dx \\
&\geq \int_{-\infty}^{\infty} \frac{2}{\sqrt{2\pi \widetilde{A}_{ii}^{(\ell-1)}}} \exp\left(-\frac{x^2}{2\widetilde{A}_{ii}^{(\ell-1)}}\right) \left(\frac{1}{4} - \exp(-x^2/8)\right) dx \\
&= \frac{1}{2} - \frac{1}{2\sqrt{1 + \widetilde{A}_{ii}^{(\ell-1)}/4}} \\
&\geq \frac{1 - (1 + G_{\max}/4)^{-1/2}}{2G_{\max}} \widetilde{A}_{ii}^{(\ell-1)}.
\end{aligned}
$$

**Tanh.** When $\sigma_{\ell-1}$ is Tanh, we can upper bound $G_{ii}^{(\ell)}$ by

$$
\begin{aligned}
G_{ii}^{(\ell)} &= \mathbb{E}_{w \sim \mathcal{N}(0, \widetilde{A}_{ii}^{(\ell-1)})}[\sigma_{\ell-1}(w)^2] \\
&= \int_{-\infty}^{\infty} \frac{2}{\sqrt{2\pi \widetilde{A}_{ii}^{(\ell-1)}}} \exp\left(-\frac{x^2}{2\widetilde{A}_{ii}^{(\ell-1)}}\right) \text{Tanh}(x)^2 dx \\
&\leq \int_{-\infty}^{\infty} \frac{2}{\sqrt{2\pi \widetilde{A}_{ii}^{(\ell-1)}}} \exp\left(-\frac{x^2}{2\widetilde{A}_{ii}^{(\ell-1)}}\right) \left(1 - \exp(-x^2)\right) dx \\
&= 2 - \frac{2}{\sqrt{1 + 2\widetilde{A}_{ii}^{(\ell-1)}}} \\
&\leq 2\widetilde{A}_{ii}^{(\ell-1)}
\end{aligned}
$$

and lower bound $G_{ii}^{(\ell)}$ by

$$
\begin{aligned}
G_{ii}^{(\ell)} &= \mathbb{E}_{w \sim \mathcal{N}(0, \widetilde{A}_{ii}^{(\ell-1)})}[\sigma_{\ell-1}(w)^2] \\
&= \int_{-\infty}^{\infty} \frac{2}{\sqrt{2\pi \widetilde{A}_{ii}^{(\ell-1)}}} \exp\left(-\frac{x^2}{2\widetilde{A}_{ii}^{(\ell-1)}}\right) \mathrm{Tanh}(x)^2 dx \\
&\geq \int_{-\infty}^{\infty} \frac{2}{\sqrt{2\pi \widetilde{A}_{ii}^{(\ell-1)}}} \exp\left(-\frac{x^2}{2\widetilde{A}_{ii}^{(\ell-1)}}\right) \left(1 - \exp(-x^2/2)\right) dx \\
&= 2 - \frac{2}{\sqrt{1 + \widetilde{A}_{ii}^{(\ell-1)}}} \\
&\geq \frac{2(1 - (1 + G_{\max})^{-1/2})}{G_{\max}} \widetilde{A}_{ii}^{(\ell-1)}.
\end{aligned}
$$

$\square$

**Proposition 1.** *Let $h_0, ..., h_{n-1}$ be the ensemble of hypothesis generated by an arbitrary online algorithm working with a loss satisfying $\ell \in [0, 1]$. Let us define $X_1^{t-1} = \{(x_1, x_1), ..., (x_{t-1}, y_{t-1})\}$ a sequence of samples from 1-st to $(t-1)$-th iteration. Then, for any $\delta \in (0, 1]$, we have*

$$
P\left(\frac{1}{n}\sum_{t=1}^{n}\mathbb{E}\left[\ell(h_{t-1}(x_t), y_t)|X_1^{t-1}\right] - \frac{1}{n}\sum_{t=1}^{n}\ell(h_{t-1}(x_t), y_t) \geq \sqrt{\frac{2}{n}\ln\frac{1}{\delta}}\right) \leq \delta,
$$

*where the expectation is computed on random sample $(x_t, y_t)$ condition on the previous sequence $X_1^{t-1} = \{(x_1, x_1), ..., (x_{t-1}, y_{t-1})\}$*

*Proof.* This proof extends the Proposition 1 of Cesa-Bianchi et al. (2004) from i.i.d. data to time-series data. Our definition on time-series data follows Kuznetsov & Mohri (2016).

For each $t = 1, \ldots, n$ set

$$
V_{t-1} = \mathbb{E}\left[\ell(h_{t-1}(x_t), y_t)|X_1^{t-1}\right] - \ell(h_{t-1}(x_t), y_t).
$$

We have $V_{t-1} \in [-1, 1]$ since loss $\ell \in [0, 1]$.

By taking expectation condition on the previous sequence $X_1^{t-1} = \{(x_1, x_1), ..., (x_{t-1}, y_{t-1})\}$, we have

$$
\mathbb{E}[V_{t-1}|X_1^{t-1}] = \mathbb{E}\left[\ell(h_{t-1}(x_t), y_t)|X_1^{t-1}\right] - \mathbb{E}\left[\ell(h_{t-1}(x_t), y_t)|X_1^{t-1}\right] = 0
$$

Finally, we conclude our proof by using Hoeffding-Azuma inequality to a sequence of dependent random variables $\{V_{t-1}\}_{t=1}^{n}$:

$$
P\left(\frac{1}{n}\sum_{t=1}^{n}V_{t-1} - \mathbb{E}\left[\frac{1}{n}\sum_{t=1}^{n}V_{t-1} \mid X_1^{t-1}\right] \geq \sqrt{\frac{2}{n}\ln\frac{1}{\delta}}\right) \leq \delta.
$$

$\square$

## G   GRADIENT COMPUTATION

Let us denote $i_\ell$ as the $i$-th node at the $\ell$-th layer. By the definition of the forward propagation rule, we have $i = i_{L-1} = i_L$. For notation simplicity, let us define

$$\tilde{\mathbf{z}}_{i_\ell}^{(\ell-1)} = \sum_{i_{\ell-1}\in\mathcal{N}(i_\ell)} P_{i_\ell,i_{\ell-1}}\mathbf{h}_{i_{\ell-1}}^{(\ell-1)}, \ \forall \ell \in [L-1]$$

and $\mathbf{D}_{i,\ell} = \mathrm{diag}(\sigma_\ell'(\mathbf{z}_i^{(\ell)})) \in \mathbb{R}^{d_\ell\times d_\ell}$ is a diagonal matrix.

When $\ell = L$, we know that the gradient with respect to $\mathbf{W}^{(L)}$ is computed as

$$\frac{\partial f_{i_L}(\boldsymbol{\theta})}{\partial \mathbf{W}^{(L)}} = \mathbf{h}_{i_L}^{(L-1)} \in \mathbb{R}^{1\times d_{L-1}}. \tag{6}$$

When $\ell = L-1$, by the chain rule, we know the gradient with respect to $\mathbf{W}^{(L-1)}$ is computed as

$$\frac{\partial f_{i_L}(\boldsymbol{\theta})}{\partial \mathbf{W}^{(L-1)}} = \frac{\partial f_{i_L}(\boldsymbol{\theta})}{\partial \mathbf{h}_{i_L}^{(L-1)}}\frac{\partial \mathbf{h}_{i_L}^{(L-1)}}{\partial \mathbf{W}^{(L-1)}}$$

$$= [\mathbf{W}^{(L)}\mathbf{D}_{i_{L-1}}^{(L-1)}]^\top \tilde{\mathbf{z}}_{i_{L-1}}^{(L-2)} \in \mathbb{R}^{d_{L-1}\times d_{L-2}}.$$

When $\ell = L-2$, by the chain rule, we know the gradient with respect to $\mathbf{W}^{(L-2)}$ is computed as

$$\frac{\partial f_{i_L}(\boldsymbol{\theta})}{\partial \mathbf{W}^{(L-2)}} = \frac{\partial f_{i_L}(\boldsymbol{\theta})}{\partial \mathbf{h}_{i_{L-1}}^{(L-1)}} \sum_{i_{L-2}\in\mathcal{N}(i_{L-1})} \frac{\partial \mathbf{h}_{i_{L-1}}^{(L-1)}}{\partial \mathbf{h}_{i_{L-2}}^{(L-2)}}\frac{\partial \mathbf{h}_{i_{L-2}}^{(L-2)}}{\partial \mathbf{W}^{(L-2)}}$$

$$\underset{(a)}{=} \sum_{i_{L-2}\in\mathcal{N}(i_{L-1})} P_{i_{L-1},i_{L-2}}\left[\mathbf{W}^{(L)}\left(\mathbf{D}_{i_{L-1}}^{(L-1)}\mathbf{W}^{(L-1)} + \alpha_{L-2}\mathbf{I}_m\right)\mathbf{D}_{i_{L-2}}^{(L-2)}\right]^\top \tilde{\mathbf{z}}_{i_{L-2}}^{(L-3)} \in \mathbb{R}^{d_{L-2}\times d_{L-3}},$$

where the equality (a) holds because $i = i_{L-1} = i_L$ by definition.

When $\ell = L-3$, by the chain rule, we know the gradient with respect to $\mathbf{W}^{(L-3)}$ is computed as

$$\frac{\partial f_{i_L}(\boldsymbol{\theta})}{\partial \mathbf{W}^{(L-3)}} = \frac{\partial f_{i_L}(\boldsymbol{\theta})}{\partial \mathbf{h}_{i_{L-1}}^{(L-1)}} \sum_{i_{L-2}\in\mathcal{N}(i_{L-1})} \frac{\partial \mathbf{h}_{i_{L-1}}^{(L-1)}}{\partial \mathbf{h}_{i_{L-2}}^{(L-2)}} \sum_{i_{L-2}\in\mathcal{N}(i_{L-1})} \frac{\partial \mathbf{h}_{i_{L-2}}^{(L-2)}}{\partial \mathbf{h}_{i_{L-3}}^{(L-3)}}\frac{\partial \mathbf{h}_{i_{L-3}}^{(L-3)}}{\partial \mathbf{W}^{(L-3)}}$$

$$= \sum_{i_{L-2}\in\mathcal{N}(i_{L-1})} P_{i_{L-1},i_{L-2}} \sum_{i_{L-3}\in\mathcal{N}(i_{L-2})} P_{i_{L-2},i_{L-3}}$$

$$\left[\mathbf{W}^{(L)}\left(\mathbf{D}_{i_{L-1}}^{(L-1)}\mathbf{W}^{(L-1)} + \alpha_{L-2}\mathbf{I}_m\right)\left(\mathbf{D}_{i_{L-2}}^{(L-2)}\mathbf{W}^{(L-2)} + \alpha_{L-3}\mathbf{I}_m\right)\mathbf{D}_{i_{L-3}}^{(L-3)}\right]^\top \tilde{\mathbf{z}}_{i_{L-3}}^{(L-4)} \in \mathbb{R}^{d_{L-3}\times d_{L-4}}.$$

For notation simplicity, let us define

$$\mathbf{G}^\ell(i_\ell, i_{\ell+1}, \ldots, i_{L-1})$$
$$= \left[\mathbf{W}^{(L)}\left(\mathbf{D}_{i_{L-1}}^{(L-1)}\mathbf{W}^{(L-1)} + \alpha_{L-2}\mathbf{I}_m\right)\ldots\left(\mathbf{D}_{i_{\ell+1}}^{(\ell+1)}\mathbf{W}^{(\ell+1)} + \alpha_\ell\mathbf{I}_m\right)\mathbf{D}_{i_\ell}^{(\ell)} \in \mathbb{R}^{1\times d_\ell}\right]^\top \tilde{\mathbf{z}}_{i_\ell}^{(\ell-1)} \in \mathbb{R}^{d_\ell\times d_{\ell-1}}.$$

By recursion, we know that for $\ell = 1, \ldots, L-2$, the gradient with respect to $\mathbf{W}^{(\ell)}$ is computed by

$$\frac{\partial f_i(\boldsymbol{\theta})}{\partial \mathbf{W}^{(\ell)}} = \sum_{i_{L-2}\in\mathcal{N}(i_{L-1})} \sum_{i_{L-3}\in\mathcal{N}(i_{L-2})} \cdots \sum_{i_\ell\in\mathcal{N}(i_{\ell+1})} P_{i_{L-1},i_{L-2}}P_{i_{L-2},i_{L-3}}\ldots P_{i_{\ell+1},i_\ell}\mathbf{G}^\ell(i_\ell, i_{\ell+1}, \ldots, i_{L-1}).$$

# H    DETAILS ON RELATED WORKS

## H.1    TEMPORAL GRAPH LEARNING

Figure 2 is an illustration of temporal graph, where each node has node feature $\mathbf{x}_i$, each node pair could have multiple temporal edges with different timestamps $t$ and edge features $\mathbf{e}_{ij}(t)$. We classify the existing temporal graph learning methods into *memory-based* (e.g., JODIE), *GNN-based* (e.g., TGAT, TGSRec), *memory&GNN-based* (e.g., TGN, APAN, and PINT), *RNN-based* (e.g., CAW), and *GNN&RNN-based* (e.g., DySAT) methods. We briefly introduce the most representative method for each category and defer more TGL methods to Appendix H.4.

**Memory-based method.** JODIE Kumar et al. (2019) maintains a memory block for each node and updates the memory block by an RNN upon the happening of each interaction. Let us denote $\mathbf{s}_i(t)$ as the memory of node $v_i$ at time $t$, denote $h_i^t$ as the timestamp that node $v_i$ latest interacts with other nodes before time $t$. When node $v_i$ interacts with other node at time $t$, JODIE updates the memory-block of node $v_i$ by

$$\mathbf{s}_i(t) = \text{RNN}\left(\mathbf{s}_i^+(h_i^t), (\mathbf{s}_j^+(h_j^t), \mathbf{e}_{ij}(t))\right),$$

where $\mathbf{s}_i^+(t) = \text{StopGrad}(\mathbf{s}_i(t))$ is applying stop gradient on $\mathbf{s}_i(t)$ and the stop gradient is required to reduce the computational cost. Finally, $\mathbf{s}_i(t)$ will be used for the downstream tasks. Maintaining the memory block allows each node to have access to the full historical interaction information of its temporal graph neighbors. However, the performance might be sub-optimal since RNN cannot update the memory blocks due to the stop gradient operation.

**GNN-based method.** TGAT Xu et al. (2020a) first constructs the temporal computation graph, where all the paths that from the root node to the leaf nodes in the computation graph respect the temporal order. Then, TGAT recursively computes the hidden representation of each node by

$$\mathbf{h}_i^{(\ell)}(t_i) = \text{AGG}^\ell\left(\left\{(\mathbf{h}_j^{(\ell-1)}(t_j), \mathbf{e}_{ij}(t_j)) \mid j \in \mathcal{N}(i, t_i)\right\}\right),$$

where $\mathbf{h}_i^{(0)} = \mathbf{x}_i$ is the node feature and TGAT uses self-attention for aggregation. Finally, the final representation $\mathbf{h}_i^{(L)}(t) = \text{MLP}(\mathbf{h}_i^{(L-1)}(t))$ will be used for downstream tasks. TGSRec Fan et al. (2021) proposes to advance self-attention by collaborative attention, such that self-attention can simultaneously capture collaborative signals from both users and items, as well as consider temporal dynamics inside the sequential pattern.

**Memory&GNN-based method.** TGN Rossi et al. (2020) is a combination of memory- and GNN-based methods. TGN first uses memory blocks to capture all temporal interactions (similarly to JODIE) then applies GNN on the latest representation of the memory blocks of each node to capture the spatial information (similarly to TGAT). Different from pure GNN-based methods, TGN uses $\mathbf{h}_i^{(0)}(t_i) = \mathbf{s}_i(t_i)$ instead. APAN Wang et al. (2021b) flips the order of GNN and memory blocks in TGN by first computing the node representations for both nodes that are involved in an interaction, then using these node representations to update the memory blocks with RNN modules. PINT Souza et al. (2022) uses an injective aggregation and pre-computed positional encoding to improve the expressive power of TGN.

**RNN-based method.** CAW Wang et al. (2021c) proposes to first construct a set of sequential temporal events as $\mathcal{S}_i(t) = \{\mathbf{v}_1, \ldots, \mathbf{v}_{L-1}\}$, where each temporal event is generated by initiating a number of temporal walks on the temporal graph starting at node $v_i$ at time $t$. Then, the frequency of the node that appears in the temporal walks is used as the representation of each event $\mathbf{v}_\ell$. Finally, RNN is used to aggregate all temporal events by

$$\mathbf{h}_\ell = \text{RNN}(\mathbf{h}_{\ell-1}, \mathbf{v}_\ell), \ \forall \ell \in [L-1], \mathbf{h}_0 = \mathbf{0}.$$

To this end, the final representation $\mathbf{h}_L = \text{MLP}(\mathbf{h}_{L-1})$ is used for downstream tasks.

**GNN&RNN-based method.** DySAT Sankar et al. (2020) first pre-processes the temporal graph into multiple snapshot graphs by splitting all timestamps into multiple time slots and merging all edges in each time slot. After that, DySAT applies GNN on each snapshot graph independently to extract the spatial features and applies RNN on the output of GNN to extract spatial features at different timestamps to capture the temporal dependencies of snapshot graphs.

## H.2 Expressive power of TGL algorithms

Souza et al. (2022); Gao & Ribeiro (2022) study the expressive power of temporal graph neural networks through the lens of Weisfeiler-Lehman isomorphism test (1-WL test):

- Souza et al. (2022) show that ① using injective aggregation in temporal graph neural networks is essentially important to achieve the same expressive power as applying 1-WL test on temporal graph and ② memory blocks augmented TGL algorithms (e.g., JODIE Kumar et al. (2019) and TGN Rossi et al. (2020)) are strictly more powerful than the TGL algorithms that solely relying on local message passing (e.g., TGAT Xu et al. (2020a)), unless the number of message passing steps is large enough.
- Gao & Ribeiro (2022) cast existing temporal graph methods into time-and-graph (i.e., GNN and RNN are intertwined to represent the temporal evolution of node attributes in the graph) and time-then-graph (i.e., first use RNN then use GNN). They show that time-then-graph representations have an expressivity advantage over time-and-graph representations.

On the other hand, Gao et al. (2023); Bouritsas et al. (2022); Li et al. (2020); Srinivasan & Ribeiro (2019); Wang et al. (2022); Abboud et al. (2020) also study the expressive power via isomorphism test on static graph, which is not the main focus of this paper.

## H.3 Generalization analysis on graph representation learning

In recent years, a large number of papers are working on the generalization of GCNs using different measurements. In particular, existing works aims at providing an upper bound on the difference between training error and evaluation error.

- *Uniform stability* considers how the perturbation of training examples would affect the output of an algorithm. Verma & Zhang (2019) analyze the uniform stability of the single-layer GCN model and derives a generalization bound that depends on the largest absolute eigenvalue of its graph convolution filter and the number of training iterations. Zhou & Wang (2021) generalize Verma & Zhang (2019) to 2-layer GCN. Cong et al. (2021) study the generalization ability of multiple GNN structures via transductive uniform stability. They show that the generalization gap of GCN depends on the node degree and it increases exponentially with the number of layers and the number of training iterations.
- *Rademacher complexity* considers how the perturbation of weight parameters would affect the output of an algorithm. Garg et al. (2020) consider a special case of GCN structure where all layers are sharing the same weight matrix. They show that the generalization error grows proportional to the maximum node degree, the number of layers, and the hidden feature dimension. Oono & Suzuki (2020) study the transductive Rademacher complexity of multi-scale GNN (e.g., jumping knowledge network Xu et al. (2018)) trained with a gradient boosting algorithm. Du et al. (2019) study the Rademacher complexity under an over-parameterized regime using the Neural Tangent Kernel (NTK) of an infinite-wide single-layer GNN.
- *PAC-Bayesian* considers how the perturbation of weight parameters would affect the output of an algorithm. Liao et al. (2020) show that the maximum node degree and spectral norm of the weights govern the generalization bounds. In practice, the PAC-Bayesian bound in Liao et al. (2020) has a tighter dependency on the maximum node degree and the maximum hidden dimension than the Rademacher complexity bound in Garg et al. (2020).
- Xu et al. (2020c) derive a *PAC-learning* sample complexity bound that decreases with better alignment.
- Maskey et al. (2022) study *uniform convergence* generalization on the random graph.
- Kuznetsov & Mohri (2015; 2016) study the generalization of non-stationary time series, but they did not take the neural architecture design into consideration.

Most of the aforementioned generalization measurements are data-label independent and only dependent on architecture of a model (e.g., number of layers and hidden dimension), which cannot fully explain why one method is better than another.

In this paper, we study the generalization error of multiple temporal graph learning algorithms under the over-parameterized regime, but without the infinite-wide assumption. Our analysis establishes a strong connection between the upper bound of evaluation error and feature-label alignment (FLA). FLA has been previously appeared in the generalization analysis of over-parameterized neural networks in Arora et al. (2019); Du et al. (2019); Cao & Gu (2019), which could reflect how well the representation of different algorithms aligned with its ground truth labels. We extend the generalization analysis to multi-layer GNN, multi-steps RNN, and memory-based temporal graph learning methods, as well as propose to use the FLA as a proxy of expressive power.

### H.4 More temporal graph learning algorithms

In the following, we review more existing temporal graph learning algorithms:

- *MeTA* Wang et al. (2021d) proposes data augmentation to overcome the over-fitting issue in temporal graph learning. More specifically, they generate a few graphs with different data augmentation magnitudes and perform the message passing between these graphs to provide adaptively augmented inputs for every predictions.

- *TCL* Wang et al. (2021a) proposes to first use a transformer to separately extract the temporal neighborhood representations associated with the two interaction nodes, then utilizes a co-attentional transformer to model inter-dependencies at a semantic level. To boost model performance, contrastive learning is used to maximize the mutual information between the predictive representations of two future interaction nodes.

- *TNS* Wang et al. (2021e) proposes a temporal-aware neighbor sampling strategy that can provide an adaptive receptive neighborhood for each node at any time.

- *LSTSR* Chi et al. (2022) proposes Long Short-Term Preference Modeling for Continuous-Time Sequential Recommendation to capture the evolution of short-term preference under dynamic graph.

- *DyRep* Trivedi et al. (2019) uses RNNs to propagate messages in interactions to update node representations.

- *DynAERNN* Goyal et al. (2018) uses a fully connected layer to first encode the network representation, then pass the encoded features to the RNN, and use the fully connected network to decode the future network structure.

- *VRGNN* Hajiramezanali et al. (2019) generalizes variational graph auto-encoder to temporal graphs, which makes priors dependent on historical dynamics and captures these dynamics using RNN.

- *EvolveGCN* Pareja et al. (2020) uses RNN to estimate GCN parameters for future snapshots.

- *DDGCL* Tian et al. (2021) proposes a debiased GAN-type contrastive loss as the learning objective to correct the sampling bias that occurred in the negative sample construction process of temporal graph learning.

## I  Further discussions and clarifications on important details

### I.1 Discussion on considering feature-label alignment as a proxy of expressiveness

In the following, we provide a high-level intuition on why we treat feature-label alignment as a proxy of expressiveness. Let us consider loss function as logistic loss and our objective function as

$$\mathcal{L}(\boldsymbol{\theta}) = \sum_{i=1}^{N} \psi(y_i f_i(\boldsymbol{\theta})), \ \psi(x) = \log(1 + \exp(-x)).$$

Since $\psi(x)$ is a monotonically decreasing function, minimizing $\mathcal{L}(\boldsymbol{\theta})$ can be achieved by finding $\boldsymbol{\theta}$ that maximizing $\mathcal{L}'(\boldsymbol{\theta}) = \sum_{i=1}^{N} y_i f_i(\boldsymbol{\theta})$.

By Taylor expanding the network function with respect to the weights around its initialization $\boldsymbol{\theta}_0$, we have

$$f_i(\boldsymbol{\theta}_0 + \Delta\boldsymbol{\theta}) \approx f_i(\boldsymbol{\theta}_0) + \langle \nabla_\theta f_i(\boldsymbol{\theta}_0), \Delta\boldsymbol{\theta} \rangle,$$

which implies

$$\mathcal{L}'(\boldsymbol{\theta}_0 + \Delta\boldsymbol{\theta}) = \sum_{i=1}^{N} y_i f_i(\boldsymbol{\theta}_0 + \Delta\boldsymbol{\theta}) \approx \sum_{i=1}^{N} y_i f_i(\boldsymbol{\theta}_0) + \sum_{i=1}^{N} y_i \langle \nabla_\theta f_i(\boldsymbol{\theta}_0), \Delta\boldsymbol{\theta} \rangle.$$

The above equation tells us that we can minimize $\mathcal{L}'(\boldsymbol{\theta}_0 + \Delta\boldsymbol{\theta})$ by looking for the perturbation $\Delta\boldsymbol{\theta}$ on the weight parameters that maximize the second term $\sum_{i=1}^{N} y_i \langle \nabla_\theta f_i(\boldsymbol{\theta}_0), \Delta\boldsymbol{\theta} \rangle$. In other words, we are looking for $\Delta\boldsymbol{\theta} \in \mathbb{R}^{|\boldsymbol{\theta}|}$ such that

$$\mathbf{J}\Delta\boldsymbol{\theta} = C\mathbf{y}, \text{ where } \mathbf{J} = [\text{vec}(\nabla_\theta f_i(\boldsymbol{\theta}_0))]_{i=1}^{N} \in \mathbb{R}^{N \times |\boldsymbol{\theta}|} \text{ and } C > 0. \tag{7}$$

The larger the constant $C$, the smaller the logistic loss. Here we just think of it as a constant that is the same for all methods.

Finally, let us define the singular value decomposition of $\mathbf{J}$ as $\mathbf{J} = \mathbf{P}\boldsymbol{\Lambda}\mathbf{Q}^\top$ where $\mathbf{P} \in \mathbb{R}^{N \times N}, \mathbf{Q} \in \mathbb{R}^{|\boldsymbol{\theta}| \times |\boldsymbol{\theta}|}$ are orthonormal matrices that each column vectors are singular vectors, $\boldsymbol{\Lambda} \in \mathbb{R}^{N \times |\boldsymbol{\theta}|}$ is the singular value matrix. In the over-parameterized regime, we assume $|\boldsymbol{\theta}| \gg N$ and the smallest singular value of $(\mathbf{J}\mathbf{J}^\top)^{-1}$ is always positive Nguyen (2021).

Then, the connection between $\|\Delta\boldsymbol{\theta}\|_2^2$ and feature-label alignment is derived by

$$\begin{aligned}
\|\Delta\boldsymbol{\theta}\|_2^2 &= C^2 \|(\mathbf{Q}\boldsymbol{\Lambda}^{-1}\mathbf{P}^\top)\mathbf{y}\|_2^2 \\
&= C^2 \mathbf{y}^\top (\mathbf{Q}\boldsymbol{\Lambda}^{-1}\mathbf{P}^\top)^\top (\mathbf{Q}\boldsymbol{\Lambda}^{-1}\mathbf{P}^\top)\mathbf{y} \\
&= C^2 \mathbf{y}^\top (\mathbf{P}\boldsymbol{\Lambda}^{-1}\mathbf{Q}^\top \mathbf{Q}\boldsymbol{\Lambda}^{-1}\mathbf{P}^\top)\mathbf{y} \\
&= C^2 \mathbf{y}^\top (\mathbf{J}\mathbf{J}^\top)^{-1}\mathbf{y},
\end{aligned}$$

which means $\|\Delta\boldsymbol{\theta}\|_2 = \mathcal{O}(\sqrt{\mathbf{y}^\top (\mathbf{J}\mathbf{J}^\top)^{-1}\mathbf{y}})$. One can also refer to the proof the Theorem 2 for more details.

It is natural to think a model is expressive if only small perturbations on the random initialized weight parameters are required to achieve low error.

Besides, feature-label alignment is closely related to both the convergence and the generalization ability of neural networks Du et al. (2018); Arora et al. (2019). For example:

- **Connection to convergence analysis.** Du et al. (2018) show that when training over-parameterized neural networks with square loss $\ell(\boldsymbol{\theta}) = \sum_{i=1}^{N}(y_i - f_i(\boldsymbol{\theta}))^2$, the square loss at the $k$-th iteration $\ell(\boldsymbol{\theta}_k)$ can be upper bounded by

$$\ell(\boldsymbol{\theta}_k) \leq \left(1 - \frac{\eta\lambda_{\min}(\mathbf{K}^\infty)}{4}\right)^k \ell(\boldsymbol{\theta}_0),$$

  where $\eta$ is the learning rate, $\lambda_{\min}(\mathbf{K}^\infty)$ is the smallest eigenvalue of $\mathbf{K}^\infty$, and $\mathbf{K}^\infty$ is the neural tangent kernel of an infinite-wide two-layer ReLU network, i.e., we have $\mathbf{K}^\infty = \mathbf{J}\mathbf{J}^\top$ as $|\boldsymbol{\theta}| \to \infty$. From this equation, we know that the larger the $\lambda_{\min}(\mathbf{K}^\infty)$, the faster the convergence speed. Interestingly, $\lambda_{\min}(\mathbf{K}^\infty)$ is in fact closely related to the feature-alignment term because

$$\mathbf{y}^\top(\mathbf{K}^{-1})\mathbf{y} \leq \frac{\|\mathbf{y}\|_2^2}{\lambda_{\min}(\mathbf{K})} \Leftarrow \frac{1}{\lambda_{\min}(\mathbf{K})} = \lambda_{\max}(\mathbf{K}^{-1}) = \max_{\mathbf{y} \in \mathbb{R}^N} \frac{\mathbf{y}^\top(\mathbf{K}^{-1})\mathbf{y}}{\|\mathbf{y}\|_2^2}.$$

  That is to say, the smaller the feature-label alignment score, the larger the $\lambda_{\min}(\mathbf{K}^\infty)$, and potentially the faster the convergence.

- **Connection to generalization.** Besides, Arora et al. (2019) show that for any 1-Lipschitz loss function the generalization error of the two-layer infinite-wide ReLU network found by gradient descent is bounded by the feature-label alignment term

$$\sqrt{\frac{\mathbf{y}^\top(\mathbf{K}^\infty)^{-1}\mathbf{y}}{N}},$$

where $\mathbf{K}^\infty$ is the neural tangent kernel of infinite-wide two-layer ReLU network, similar to the convergence analysis we discussed previously in Du et al. (2018). Our results in Theorem 1 are similar to their results. However, our results hold for multi-layer neural network with different activation functions (e.g., ReLU, LeakyReLU, Sigmoid, and Tanh) without the infinite-wide assumption.

## I.2 COMPARISON BETWEEN THE CLASSICAL STATISTICAL LEARNING THEORY TO THE GENERALIZATION ANALYSIS USED IN THIS PAPER

Classical statistical learning theories (e.g., Rademacher complexity, uniform stability, and PAC-Bayesian that are reviewed in Appendix H.3) are closely related to the Lipschitz continuity and smoothness constants of a neural network. As a result, the generalization error usually increases proportionally to the number of layers and hidden dimension size. Classical statistical learning theories ① cannot capture the details of the network, ② ignore the relationship between multiple layers, and ③ could have a vague bound when studying deep neural networks with a large number of parameters. For example, the Rademacher complexity analysis in Garg et al. (2020) shows that the generalization error can be upper bounded by

$$\mathcal{O}\left(\frac{mD \max\{L, \sqrt{dL}\}}{\sqrt{N}}\right),$$

where $m$ is the hidden dimension, $D = \max_i |\mathcal{N}(v_i)|$ is the maximum number of neighbors, $L$ is the number of layers, and $N$ is the number of training data. The bound becomes vague when the hidden dimension size $m$ is large. This is because it is well known that deep neural networks work well in practice and have large $m$, while the theory is suggesting the opposite. To solve such an issue, over-parameterized regime generalization analysis (e.g., NTK-based or mean-field-based analysis) has been proposed to show that deep learning models have good generalization in theory, e.g., Arora et al. (2019); Du et al. (2019); Cao & Gu (2019). Our analysis is under the over-parameterized regime and requires the hidden dimension size $m$ large enough in theory. However, in practice, our experiment results show that the generalization bound is still meaningful even though $m$ is not that big (e.g., we are using $m = 100$ for all methods).

## I.3 DISCUSSION ON THE VARIATION OF FEATURE-LABEL ALIGNMENT (FLA) SCORE IN FIGURE 3

This is because the FLA is computed by $\mathbf{y}^\top (\mathbf{J}\mathbf{J}^\top)^{-1}\mathbf{y}$, where $\mathbf{J} = [\text{vec}(\nabla_\theta f_i(\boldsymbol{\theta}_0))]_{i=1}^N$ and $\boldsymbol{\theta}_0$ is the weight parameters at initialization. At each run, the initialization might be slightly different due to different random seeds, therefore FLA is also different at each run. The generalization error in Figure 1 changes at each run for the same reason since the generalization error is dependent on the FLA score. Besides, we would like to notice that FLA mathematically converges to a constant value as the hidden dimension size goes to infinity. In other words, given an over-parameterized model, the FLA score that computed with different random seed is almost identical.

## I.4 DETAILS ON THE COMPUTATION OF GENERALIZATION ERRORS IN FIGURE 1

The computation of the generalization error follows Theorem 1 which is dominated by the first term $\mathcal{O}(DRC/\sqrt{N})$. Here the feature-label alignment related term $R$ is empirically computed and the number of training data $N$ is identical to all methods. Therefore, in the following, we will explicitly provide discussion on how $DC$ is computed. Recall from Theorem 1 that ① $L - 1$ equals to the number of layers/steps in GNN/RNN, ② $C_{\text{GNN}} = ((1 + 3\rho)\tau)^{L-1}$, $D_{\text{GNN}} = L$ for $L$-layer GNN, ③ $C_{\text{RNN}} = (1 + 3\rho/\sqrt{2})^{L-1}$, $D_{\text{RNN}} = L$ for $L$-step RNN, and ④ $C_{\text{memory}} = \rho$ and $D_{\text{memory}} = 4$ for memory-based method. Therefore, we adopt the following settings

- Since TGAT Xu et al. (2020a) is using 2-layer self-attention for aggregation and using ReLU activation, we set $\tau = 1, \rho = 1, D_{\text{GNN}} = L_{\text{GNN}} = 3$.
- Since TGN and APAN are using 1-layer self-attention for aggregation and using ReLU activation, we set $\tau = 1, \rho = 1, D_{\text{GNN}} = L_{\text{GNN}} = 2, D_{\text{memory}} = 4$. We first compute the generalization error for GNN and memory-based method respectively, then multiply the generalization error of GNN and memory-based method together.

- Since JODIE Kumar et al. (2019) is memory-based method and using Tanh activation, we set $\rho = 1$, $D_{\text{memory}} = 4$.

- Since DySAT Sankar et al. (2020) is using 2-layer self-attention and 3-step RNN with both ReLU and Tanh as activation function, we set $\tau = 1$, $\rho = 1$, $D_{\text{GNN}} = L_{\text{GNN}} = 3$, $D_{\text{RNN}} = L_{\text{RNN}} = 4$. We first compute the generalization error for GNN and RNN respectively, then multiply the generalization error of GNN and RNN together.

- Since SToNe is using 1-layer aggregation and ReLU activation, we set $\tau = 1$, $\rho = 1$, $D_{\text{GNN}} = L_{\text{GNN}} = 2$.

Moreover, to alleviate the computation burden and make sure the over-parameterization assumption holds, we compute feature-label alignment on the last $5,000$ data points in the training set instead of all the training data. In practice, it takes less than 5 seconds to compute the feature-label alignment for each method on our GPU.

Furthermore, we do not calculate the generalization error of pure RNN-based methods (e.g., CAW Wang et al. (2021c)) due to the following reasons:

- Computing the generalization error requires computing the feature-label alignment (FLA) score, which is non-trivial for CAW Wang et al. (2021c) because their implementation is complicated (e.g., performing temporal walks and computing augmented node features) and adapting their method to our framework is challenging.

- We have already compared the generalization error to that of DySAT Sankar et al. (2020), which combines both RNN and memory-based methods.

Moreover, we did not consider layer normalization in theoretical analysis because it is data dependent and changes during training. We simply assume data is zero-mean with standard deviation as one.

Finally, the generalization error of GraphMixer Cong et al. (2023) is not calculated due to the non-trivial nature of deriving it for the over-parameterized MLP-mixer Tolstikhin et al. (2021). This challenge persists because the gradient dynamics of attention mechanisms, such as the token mixer in MLP-mixer and self-attention in GAT Veličković et al. (2017), are not well understood and requires assumptions on infinite input data Hron et al. (2020). Consequently, the lack of comprehensive understanding in this area hinders the establishment of a comparable generalization error bound for GraphMixer, similar to the one found in Theorem 1 for other methods.

## I.5 DISCUSSION ON USING GENERALIZATION ERROR AS THE SOLE MEASUREMENT OF TGL METHODS' EFFECTIVENESS

We want to make it clear that using the generalization error as the sole measure to determine the effectiveness of different TGL methods is not advisable. This is because there can be discrepancies between the theoretical analysis bounds and the real-world experiment results, due to factors such as

- *Assumptions not always holding in practice.* For instance, the theory suggests using SGD with a mini-batch size of 1, but in actuality, using a larger mini-batch size such as 600, leads to a more stable gradient, faster convergence, and improved overall performance.

- *Differences in neural architecture used in analysis and experiments.* For example, existing methods are using layer-normalization and self-attention, however, these were not taken into consideration as the theoretical analysis of these functions is still under-explored open problems.

Instead, we believe that the theoretical analysis in this paper can provide valuable insights for designing practical TGL algorithms in future studies.

## I.6 DISCUSSION ON THE RELATIONSHIP BETWEEN EXPRESSIVENESS AND GENERALIZATION ABILITY

It is worth noting that we cannot assume that a high level of expressive power automatically leads to good generalization ability, or vice versa. This can be seen in Theorem 1, where the expressive

power is measured by feature-label alignment (FLA) and generalization ability is determined by both FLA and the number of layers/steps. An algorithm with high expressive power and small FLA could still have poor generalization ability if it has a large number of layers/steps. The same principle applies to other expressive power analyzing methods, e.g., expressiveness analysis via 1-WL test in Souza et al. (2022). According to the 1-WL test, the maximum expressive power can be achieved through the use of injective aggregation functions, such as using hash functions or one-hot node identity vectors. However, these methods could harm generalization ability because hash functions are untrainable and the use of one-hot node identity vectors can lead to overfitting.

### I.7 DISCUSSION ON USING DIRECTION VS UNDIRECTED GRAPH IN EXPERIMENTS

It is worth noting that "directed" and "undirected" graph are just different data structures to represent the same raw data, switching from data structures to another will not introduce new information on data. Our method performs better on undirected graph structure because "applying our sampling strategy to undirected graph" could tell us how frequent two nodes interact, which is important in our algorithm design.

Moreover, we note that changing the "directed" to "undirected" graph for baseline methods has little impact on their performance. To demonstrate this, we have conducted an additional ablation study by comparing the baselines' average precision with different input data structure. The results reported on the left side of the arrow represent their original directed graph input setting, while the results reported on the right side of the arrow represent the use of an undirected graph. As shown in the table, utilizing undirected graph does not affect the baselines' performance because the information on "how frequent two nodes interact" could be learned from the data through their algorithm design.

| Average precision | Reddit | Wiki | MOOC | LastFM |
|---|---|---|---|---|
| JODIE | 99.75→99.44 | 98.94→98.90 | 98.99→98.38 | 79.41→79.91 |
| TGAT | 99.56→99.71 | 98.69→98.35 | 99.28→98.41 | 75.16→74.69 |
| TGN | 98.83→99.74 | 99.61→99.58 | 99.63→99.43 | 91.04→91.06 |
| DySAT | 98.55→98.53 | 96.64→96.62 | 98.76→98.70 | 76.28→76.23 |

### I.8 COMPARISON TO EXISTING WORK GRAPHMIXER CONG ET AL. (2023)

Our paper has a similar observation with Cong et al. (2023) that a simpler neural architecture could achieve state-of-the-art performance by carefully selecting the input data structure and utilizing inductive bias for neural network design. Comparing to Cong et al. (2023), our paper has the following further contributions:

- Our paper provides the theoretical support for their empirical observations, i.e., input data selection is important and simpler model performs well;

- Our paper designs the first unified theoretical analysis framework that can capture both the impact of "neural architecture" and "input data selection" to analyze different TGL methods;

- Our paper proposes an algorithm has a simpler neural architecture but similar or even better performance.

### I.9 DISCUSSION ON THE THEORETICAL ANALYZED CAW TO ITS IN PRACTICE IMPLEMENTATION

In this paper, we first classify existing TGL algorithms into GNN-based, RNN-based, and memory-based methods, then explicitly analysis the generalization ability of each category in an unified theoretical analysis framework using a "node-level task. CAW belongs to the RNN-based category because CAW samples "temporal events" using random walks and aggregate information via RNN. However, there are small discrepancies between the algorithm formation of RNN-based TGL method and the CAW algorithm itself. This is because CAW is originally designed for "edge-level task", where edge features are computed through "random walk paths" sampled from two nodes on both sides of the edge. To apply it to "node-level" theoretical analysis, it is natural to imagine that the target node features are computed on the "random walk path" sampled at that target node.

Since we are unable to incorporate all the implementation details into our mathematical analysis, we may need to make some adjustments/simplification for theoretical analysis as long as they could still capture the main spirits of the original algorithm, i.e., sample temporal events via random walks and aggregate information via RNN. Moreover, please note that the changes we made to our analysis are considered small in terms of generalization analysis, as transitioning from "node-level" to "edge-level" analysis will not impact our conclusion regarding generalization bound. For example to extend the results of the CAW to its original "edge-level task" setting, we simply need to treat each temporal event $v_\ell$ as the combination of the $\ell$-th link feature from two distinct random paths. In this paper, we chose the "node-level analysis" for the sake of clarity in presentation.

## I.10 DISCUSSION ON THE "STRONG CORRATION" BETWEEN "GENERALIZATION BOUND" AND "AVERAGE PRECISION" IN FIGURE 1

It is worth noting that the generalization bound is a "worst-case upper bound" that describes the theoretical limit, and there is often a gap between this upper bound and the actual generalization performance. Therefore, we are not expecting such a strong correlation exists. Although without "strong negative correlation", as shown in Figure 1, we can still clearly observe there exists a negative correlation between the generalization error bound and the average precision score. Therefore, we believe that the theoretical analysis in this paper can provide valuable insights for designing practical TGL algorithms in future studies. On the other hand, tightening the theoretical upper bound is actively studied in the theoretical machine learning field. To the best of our knowledge, our paper is the first to analyze the generalization ability of a number of TGL models under a unified theoretical framework. Compared to existing generalization analyses in statistical learning theory, our generalization bound is data-label dependent and directly captures the difference between training and evaluation error.

## I.11 DISCUSSION ON THE AGGREGATION WEIGHT

In SToNe, the aggregation weight $\alpha \in \mathbb{R}^K$ is a trainable vector, whereas in TGN with sum-aggregation, the aggregation weights are fixed and are equal to 1. By using a learnable aggregation weight $\alpha$, SToNe is able to adaptively learn the importance of temporal neighbors. In contrast, as shown in the table below, replacing it with a fixed vector, the performance slightly decreases.

|  | Reddit | Wiki | MOOC | LastFM | GDELT | UCI |
|---|---|---|---|---|---|---|
| SToNe (trainable $\alpha$) | $99.89 \pm 0.00$ | $99.85 \pm 0.05$ | $99.88 \pm 0.04$ | $95.74 \pm 0.13$ | $99.11 \pm 0.03$ | $94.60 \pm 0.31$ |
| SToNe (fixed $\alpha$) | $99.82 \pm 0.00$ | $99.79 \pm 0.05$ | $99.55 \pm 0.05$ | $95.56 \pm 0.13$ | $99.02 \pm 0.04$ | $83.35 \pm 0.32$ |

## I.12 DISCUSSION ON THE EXPRESSIVE POWER OF SToNe

**Expressive power with depth.** According to the theoritical results on WL-based expressive power analysis, deeper GNN may have more expressive power than a shallow GNN due to its larger receptive field. However, it is worth noting that we may also improve the expressiveness by using different input data selection schema and neural architectures, and achieve a good model performance without a deep GNN structure. For example, as shown in Figure 5, using more hops will hurt performance but also lead to higher computation complexity in SToNe. Similar observation has been made in a recent concurrent work Cong et al. (2023). Previous works Xu et al. (2020b; 2021b) also emphasize the importance of input data and neural architecture on model performance.

**Expressive power with permutation-invariance.** Moreover, we would like to note that SToNe assigning different elements in $\mathbb{H}_i(t)$ with same timestamps to different $\alpha_k$ could break permutation-invariance, and potentially results in a lower expressive power from the WL-based expressive power perspective. However, as demonstrated in our paper, replacing it with permutation-invariance functions (e.g., self-attention) could slightly degraded the performance. Therefore, an interesting future direction could be designing a new aggregation strategy that both enjoys the expressiveness brought by permutation-invariance and the simplicity of our model design.

### I.13 Discussion on the generalization bound of SToNe

In this paper, we only derive the generalization bounds for GNN-based, RNN-based, and memory-based TGL methods, but did not explicitly derive the generalization bound for SToNe. However, it is worth noting that SToNe (without layer normalization) could be think of as 1-hop GNN-based method using 1-layer sum-aggregation and ReLU activation, therefore we set activation function Lipschitz constant $\rho = 1$, and $D_{\text{GNN}} = L_{\text{GNN}} = 2$.

Comparing to the original generalization bound for GNN-based method, we note that $\tau$ is a notation originally used for GNN's average-aggregation in Assumption 3. Here we reuse this notation for SToNe because a similar assumption for the sum aggregation could be made for SToNe by setting $\tau$ as a large number that satisfy the condition $\tau \geq \sum_{i=1}^{K} \alpha_i$. In practice, the $\tau$ is not arbitrary large because each $\alpha_i, \forall i \in \{1, \ldots, K\}$ is sampled from uniform distribution $\mathcal{U}(-\sqrt{3/K}, \sqrt{3/K})$ according to the PyTorch Doc, and the model is trained with $\ell_2$-regularization on weight parameters. For the completeness, we empirically compute the $\sum_{i=1}^{K} \alpha_i$ value before and after training:

|                 | Reddit          | Wiki            | MOOC            | LastFM          |
|-----------------|-----------------|-----------------|-----------------|-----------------|
| Before training | $0.21 \pm 0.67$ | $0.39 \pm 0.52$ | $-0.30 \pm 0.52$ | $0.05 \pm 0.52$ |
| After training  | $0.13 \pm 0.27$ | $0.10 \pm 0.24$ | $-0.02 \pm 0.41$ | $-0.10 \pm 0.50$ |

Another approach that fulfills the aforementioned assumption is to consider $\alpha_1 = \ldots = \alpha_K = 1/K$ as non-trainable parameters after initialization. By adopting this assumption, SToNe can be considered as a 1-hop GNN-based TGL algorithm with row-normalized neighbor aggregation, adhering to Assumption 3. To provide a comprehensive analysis, we also present the performance of SToNe when utilizing non-trainable $\boldsymbol{\alpha}$. Based on the outcomes, we observe that assuming non-trainable aggregation weights yields comparable results across most datasets, thereby indicating the mildness of this assumption.

|                             | Reddit           | Wiki             | MOOC             | LastFM           | GDELT            | UCI              |
|-----------------------------|------------------|------------------|------------------|------------------|------------------|------------------|
| SToNe (trainable $\boldsymbol{\alpha}$) | $99.89 \pm 0.00$ | $99.85 \pm 0.05$ | $99.88 \pm 0.04$ | $95.74 \pm 0.13$ | $99.11 \pm 0.03$ | $94.60 \pm 0.31$ |
| SToNe (fixed $\boldsymbol{\alpha}$)     | $99.82 \pm 0.00$ | $99.79 \pm 0.05$ | $99.55 \pm 0.05$ | $95.56 \pm 0.13$ | $99.02 \pm 0.04$ | $83.35 \pm 0.32$ |

Moreover, we would like to highlight that $\tau$ is only used to empirically evaluate our generalization bound in Figure 1, therefore it is fair to set $\tau = 1$ when comparing with the empirical generalization bounds of other methods in Figure 1, based on the empirical computed numbers in table. Notice that whether we select $\tau = 1$ or not does not affect our results in Theorem 1 since we treat $\tau$ as a random variable in the upper bound.

