# OpenReview forum: "On the Generalization of Temporal Graph Learning with Theoretical Insights"
_ICLR.cc/2024/Conference — Submitted to ICLR 2024_

### Official Review · Reviewer_rLms · 2023-10-30

**Soundness:** 3 good
**Presentation:** 3 good
**Contribution:** 3 good
**Rating:** 8
**Confidence:** 4

**Summary:**

Authors study the generalization ability of different Temporal Graph Learning (TGL) algorithms including GNN-based, RNN-based, and memory-based methods, under the finite-wide over-parameterized regime. To this end, they provide a unified framework for these algorithms and bound their generalization error measured by means of the {0,1}-loss. Their bound, explicited in Theorem 1, emphasizes mainly the dependencies of the generalization error to the number of layers/steps and the feature-label alignment (FLA) score. These motivate the use of shallow and non-recursive networks operating on properly shaped input data instead of depending on memory components. De facto, authors introduce a novel model coined SToNe that fits to these principles. Then they show that SToNE achieves highly competitive performances on link prediction tasks in transductive and inductive settings, with considerably less parameters and better running times.

**Strengths:**

-	The main theoretical result is clear and depends on reasonable assumptions. Plus it covers a lot of models proposed in the TGL literature. I did not find mistakes in the proofs.
-	The subsequent analysis and choices w.r.t architectures, hyperparameters and FLA are compelling.
-	SToNe shows clearly strong empirical performances reported both in the main paper and supplementary material. The latter also contains various ablation and sensitivity analysis of interest.

**Weaknesses:**

There are some typing erros in the main paper, and many ones in the supplementary material which is quite needed so please correct them.

-	1. The presentations of the methods in Section 3, could be clearly improved. Some parts are rather unclear without checking at the corresponding literature: i) no context is provided on Algorithm 1 to position it in the corresponding optimization literature; ii) the time-encoding vector is never clearly defined. ; iii) the concept of temporal events for RNN is not clear.
-	2. Theorem 1: The dynamics w.r.t m* are not discussed, hence do not allow to access whether the chosen $m >= m^*$ in the experiments is relevant.
-	3. Some aspects regarding FLA are not clear: i) Rankings in Figure 3 do not seem to be consistent across methods, authors should explain this further. ; ii) The invertibility of $JJ^\top$ is not clearly discussed. Authors refer to (Nguyen, 2021) for this matter where results in this paper seems to depend on the assumption that at least one layer has to more parameters than there are samples in the dataset. It is not clear that used architectures satisfy this assumption.
-	4. A clear illustration of SToNe would be useful.

**Questions:**

I invite the authors to discuss the weaknesses I have mentioned above and to provide additional results/analyses for refutation. Follows some questions/remarks to clarify some points:

-	Q1. Could you further discuss the need for the LayerNormalization in SToNe ? e.g instead of no normalization or BatchNormalization ?
-	Remark: I suggest you complete the list of additional experiments reported in the Supp mentioned in the "More experimental results" paragraph.

---

> ### Author Response · Authors · 2023-11-19
>
> **Q1. The dynamics w.r.t. $m^\star$ in Theorem.**
>
> Thank you for your question. Increasing neural network hidden dimension $m$ will not affect the upper bound in Theorem 1, but it will only affect the closeness of the gradient descent solution to the random initialization, i.e., affect $\\| \widetilde{\boldsymbol{\theta}} - \boldsymbol{\theta}_0 \\|\_\mathrm{F}$. More specifically, the larger the $m$, the smaller the $\omega$ would be in Lemmas 1 to 6, and the closer $\widetilde{\boldsymbol{\theta}}$ to $\boldsymbol{\theta}_0$. This phenomenon is also know as "lazy learning" where weights only move very slightly during training in over-parameterized regime. Please refer to Section 2 of [1] for a more detailed discussion. We will clarify this in the revised version.
>
> [1] Bietti, A., and Mairal, J., On the inductive bias of neural tangent kernels. NeurIPS 2019.
>
>
> **Q2. More discussion on the ranking of different methods in Fig 3.**
>
> Thank you for your suggestion. The feature label alignment (FLA) is a component of the generalization bound (i.e., the variable $R$ in Theorem 1), which tells us how data selection affect generalization.
> FLA is defined as $\mathbf{y}^\top (\mathbf{J} \mathbf{J}^\top)^{-1} \mathbf{y}$, which only requires the label $\mathbf{y}$ and gradient $\mathbf{J}$ on training data to compute. This makes it a more effective tool for analyzing the performance of different temporal graph learning methods.
>
> In Figure 3, we compare the empirically computed FLA of different methods, and we have the following discussion based on our experience on temporal graph learning:
>
> (1) JODIE has a relatively larger FLA score than most of the other TGL methods. This is potentially due to “stop gradient” operation that prevents gradients from flowing through the memory blocks and could potentially hurt the expressive power.
>
> (2) APAN and TGN alleviate the expressive power degradation issue by applying a single layer GNN on top of the memory blocks, resulting in a smaller FLA than the pure memory-based method.
>
> (3)TGAT has a relatively smaller FLA score than other methods,which is expected since GNN has achieved outstanding performance on static graphs.
>
> (4) DySAT is originally designed for snapshot graphs, so its FLA score might be highly dependent on the selection of time-span size when converting a temporal graph to snapshot graphs. A non-optimal choice of time-span might cause information loss, which partially explains why DySAT’s FLA is large.
>
>
> **Q3. The invertibility of $\mathbf{J} \mathbf{J}^\top$ need more discussion**
>
> Thank you for your suggestion.
> The matrix $\mathbf{J} \mathbf{J}^\top$ is always invertible, as the condition on the hidden dimension $m$ in Theorem 1 already satisfies the condition in Nguyen et al. (2021). This is because $\mathbf{J}\in\mathbb{R}^{N \times d}$ is a fat matrix with more parameters ($d$) than training data ($N$), and all eigenvalues of $\mathbf{J} \mathbf{J}^\top$ are positive, being the square of the singular values of $\mathbf{J}$ as long as $\mathbf{J}$ is not low rank (e.g., no duplicate data). Therefore, $\mathbf{J} \mathbf{J}^\top$ is invertible.
>
>
> **Q4. Discussion the need for Layer Normalization?**
>
> Thank you for your question. We found that using layer normalization (LN) can help improve the stability and consistency of our model's performance when repeating experiments multiple times. This is because LN normalizes the activations within a neural network layer, making the model output less sensitive to changes in input data.
> We also experimented with batch normalization, which gave similar results. However, since batch normalization computes scaling weights from training data, it may introduce potential temporal information leakage. To avoid this issue, we prefer to use LN instead of batch normalization. We will provide more details on this in our final version.

---

### Official Review · Reviewer_w7M7 · 2023-10-31

**Soundness:** 2 fair
**Presentation:** 3 good
**Contribution:** 2 fair
**Rating:** 3
**Confidence:** 4

**Summary:**

This paper investigates the connection between the generalization error of temporal graph learning (TGL) algorithms and the feature-label alignment (FLA) score. The authors find that FLA can be used to estimate the expressive power and explain the performance of different methods. Based on their theoretical analysis, a new TGL method Simplified-Temporal-Graph-Network (SToNe) is proposed. The proposed method seems to have a small generalization error, better overall performance and lower model complexity.

**Strengths:**

The theoretical analysis and proof in this article are very detailed. The appended section also provides ample details. The discovered connection between FLA and the generalization error of temporal graph learning is quite intriguing. In addition, through the experiments, the proposed method demonstrated  good performance, e.g., small generalization error, better overall performance and lower model complexity.

**Weaknesses:**

*The motivation to define a new indicator for the generalization error is unclear: In the paper, the authors indicate that the generalization error bound decreases with respect to the number of training data, but increases with respect to the number of layers/steps in the neural-network-based methods and the feature-label alignment (FLA). Thus, FLA can be used as a proxy for the expressive power measurement. Although there is a connection between FLA and generalization error, using FLA can be unnecessary unless unique advantages of FLA exist, e.g., it is easier to measure.

*The uniqueness of the temporal graph learning algorithms, compared to static graph learning algorithms, is not reflected: Based on the definition 1, FLA can be used for all gradient-based algorithms. It is not specific to temporal graph learning methods.

*The proposed method is essentially a GNN method augmented with input data selection, hence have limited novelty as a new algorithm: Based on Section 4.1, the proposed method comprises three parts: input data preparation, encoding features via GNN and link prediction via MLP, which is similar to existing methods except for the feature engineering component.

*The performance resulting from the proposed method in the experiment is not  impressive empirically: some existing methods, e.g., TGN and GraphMixer, have a similar performance. Their average precision on the first five datasets are very close. Although the proposed method has better performance on the last dataset UCI, it is unclear whether this last dataset has some special data distribution or setting.

**Questions:**

1. What’s the advantage of using FLA? In experiments, the authors still use the generalization gap, i.e., the absolute difference between the training and validation average precision scores, to compare different algorithms.
2. The FLA score of different algorithms should be shown as part of experimental results.
3. Why do most existing methods perform well on all datasets except for UCI dataset? Does UCI dataset contain a special distribution or other specific settings?
4. Can FLA be used for the generalization error of static graph learning algorithms? If so, what’s the uniqueness of the temporal graph learning algorithms in the aspect of generalization error?

---

> ### Author Response · Authors · 2023-11-19
>
> Thank you for your comments.
> Before addressing your concerns, we would like to highlight that feature-label alignment (FLA) is proposed to solve limitations of existing generalization analysis tools. These limitations prevent us from better understanding the performance of different temporal graph learning methods.
> More specifically, existing generalization theoretical methods are data-independent and only dependent on the number of layers or hidden dimensions. However, these approaches cannot fully explain the performance difference between different algorithms. Since the selection of input data is very important in temporal graph learning, we propose FLA  to make generalization bound data-dependent.
> It is worth noting that FLA is defined as $\mathbf{y}^\top (\mathbf{J} \mathbf{J}^\top)^{-1} \mathbf{y}$, which is data structure agnostic. It only requires the label $\mathbf{y}$ and gradient $\mathbf{J}$ on training data to compute. This makes it a more effective tool for analyzing the performance of different temporal graph learning methods.
>
>  **Q1. Although there is a connection between FLA and generalization error, using FLA can be unnecessary unless unique advantages of FLA exist.**
>
> Thank you for your comment. Please note that FLA is necessary if we want to understand the selection of input data and labels to the model performance. Existing generalization analysis cannot achieve this. For more information, please refer to our Section 2 paragraph “generalization and expressive power” and Appendix I.2.
>
>
> **Q2. The uniqueness of the temporal graph learning algorithms, compared to static graph learning algorithms, is not reflected.**
>
> We conducted a theoretical analysis of GNN-based, RNN-based, and Memory-based temporal graph learning methods. These methods are distinct from static graph learning algorithms because they take into account temporal information during training.
>
>
> **Q3. The proposed method is essentially a GNN method augmented with input data selection, hence having limited novelty as a new algorithm.**
>
> Our proposal is novel because (1) it is inspired by key insights of our theoretical analysis of temporal graph methods, (2) it has a small generalization error, (3) it performs well empirically with low model complexity. We have summarized the differences compared to existing methods in Section 4.2.
>
>
> **Q4. The performance resulting from the proposed method in the experiment is not impressive empirically.**
>
> Please note that our method not only closely matches the performance of state-of-the-art temporal graph learning methods, but also offers a better theoretical understanding. Additionally, it has significantly lower model complexity, as shown in Table 2.
>
>
> **Q5. What’s the advantage of using FLA? In experiments, the authors still use the generalization gap.**
>
> Please note that FLA is a component of the generalization bound (i.e., the variable $R$ in Theorem 1), which tells us how data selection affect generalization.
> Other terms in Theorem 1 provide insights into the effect of model architecture on generalization ability.
> Therefore, to gain a more comprehensive understanding, we must also take into account the generalization gap.
>
>
> **Q6. The FLA score of different algorithms should be shown as part of experimental results**
>
>
> Please refer to Figure 3. As shown in Figure 3, we know
>
> (1) JODIE has a relatively larger FLA score than most of the other TGL methods. This is potentially due to “stop gradient” operation that prevents gradients from flowing through the memory blocks and could potentially hurt the expressive power.
>
> (2) APAN and TGN alleviate the expressive power degradation issue by applying a single layer GNN on top of the memory blocks, resulting in a smaller FLA than the pure memory-based method.
>
> (3)TGAT has a relatively smaller FLA score than other methods,which is expected since GNN has achieved outstanding performance on static graphs.
>
> (4) DySAT is originally designed for snapshot graphs, so its FLA score might be highly dependent on the selection of time-span size when converting a temporal graph to snapshot graphs. A non-optimal choice of time-span might cause information loss, which partially explains why DySAT’s FLA is large.

---

### Official Review · Reviewer_g85Z · 2023-10-31

**Soundness:** 2 fair
**Presentation:** 2 fair
**Contribution:** 3 good
**Rating:** 6
**Confidence:** 3

**Summary:**

This paper considered the generalization bound for three methods of the temporal graph learning. It is the first attempt to reveal the relevance of the feature-label alignment in the performance of algorithms in this task theoretically. It also designed a more efficient algorithm for this task based on their theoretical results.

**Strengths:**

In general, it is quite interesting to study the interaction of FLA and the model performance. This paper introduced this idea to the temporal graph learning task and designed a novel model based on their theoretical findings. I believe both the originality and quality of the idea is great. The main body of this paper is also well organized.

**Weaknesses:**

The main weakness from my perspective is the vagueness of the proof for the main theorems. Please refer to the QUESTIONS for more details.

**Questions:**

1. I think the discussion of the existing generalized bounds for GNN and so on should be elaborated on more in the literature review part and the comparison with the novel bound derived in this paper should be provided.
2. The definition of FLA depends on the gradient of $f$ with respect to $\theta$. However, the ReLU networks are non-smooth, which may make FLA undefinable.
3. In the statement of Theorem 1, when $\theta_0$ is randomly initialized, $R$ should be a random variable. It is vague to compare two random variables in the result in theorem 1.
4. Based on your findings, SToNe outperforms other methods due to its smaller FLA and thus smaller generalization bound. I am curious about the comparison of FLA of the different methods in those real datasets. The current experiments to illustrate the effect of FLA are all on SToNe itself.

The following are some questions in the proofs.

5. The proof of lemma 5 relies on the upper bound of $\|\partial loss_i(\theta)/\partial W^{(l)}\|$ for $\theta$ from the iteration of SGD. This upper bound is obtained in lemma 4, which is built on lemma 8 from Zhu et al. (2022). However, in that paper, they assumed that $W$ is a guassian random matrix, which is different in this paper (note that in this case, $W$ is obtained from SGD). Hence, i don't think that lemma can be applied here and there are some flaws in upper bounding $\|\partial loss_i(\theta)/\partial W^{(l)}\|$.
6. In the proof of Theorem 2, the part of constructing $f'\in \mathcal{F}(\theta_0,R)$ such that $\psi$ can be upper bounded by $1/\sqrt{N}$ is confusing (e.g., why $\langle \nabla_\theta f_i(\theta_0),\theta \rangle =y_i(B+B')$). I can't get the point how $R$ can make a difference in the result.

I would appreciate it if you can clear my concerns.

---

> ### Author Response · Authors · 2023-11-19
>
> **Q1. more discussion of existing generalized bounds and its comparison**
>
> Thank you for the suggestion. Most of the existing generalization bounds (e.g., Rademacher complexity, PAC-Bayesian, and Uniform Stability bounds) are data-independent and rely solely on factors like the number of layers or hidden dimensions. In contrast, our feature-label alignment generalization bound, grounded in deep learning theory, is data-dependent and operates under different assumptions. For instance, the aforementioned bounds require assumptions on weight matrix norms during training, whereas our deep learning theory bounds do not necessitate such assumptions. Consequently, direct comparisons between these results may not be practical.
>
>
> **Q2. ReLU need gradient to compute. the ReLU networks are non-smooth, which may make FLA undefinable.**
>
> Thank you for raising this excellent point. Please note that the feature-label alignment (FLA) value is empirically computed by taking the gradient of the neural network with respect to each data point. As you correctly pointed out, ReLU is a non-smooth activation function (because the gradient at 0 is not defined mathematically). However, in practical implementations using PyTorch or TensorFlow, the gradient is always set to 0 when the input data is 0. Therefore, this characteristic will not impact our definition of FLA, and we did not use this smoothness elsewhere with FLA. We will make this clarification in the revised version.
>
>
> **Q3. Since the initial weight parameters $\theta_0$ are randomly initialized, the feature-label alignment constant $R$ should be random variable. Making it vague to compare in Theorem 1.**
>
> Indeed, the feature-label alignment value depends on the initial weight parameters. Therefore, when comparing across different methods, we conduct our experiments multiple times (e.g., Figure 1), and then report their mean and standard deviation (Figure 3) for a meaningful comparison. On the other hand, it is worth noting that this feature-label alignment constant will converge to a fixed value regardless of its initialization as the neural network dimension approaches infinity, as indicated by concentration inequality.
>
> **Q4. Comparison of FLA of the different methods.**
>
> Please refer to Figure 3. As shown in Figure 3, we know
>
> (1) JODIE has a relatively larger FLA score than most of the other TGL methods. This is potentially due to “stop gradient” operation that prevents gradients from flowing through the memory blocks and could potentially hurt the expressive power.
>
> (2) APAN and TGN alleviate the expressive power degradation issue by applying a single layer GNN on top of the memory blocks, resulting in a smaller FLA than the pure memory-based method.
>
> (3)TGAT has a relatively smaller FLA score than other methods,which is expected since GNN has achieved outstanding performance on static graphs.
>
> (4) DySAT is originally designed for snapshot graphs, so its FLA score might be highly dependent on the selection of time-span size when converting a temporal graph to snapshot graphs. A non-optimal choice of time-span might cause information loss, which partially explains why DySAT’s FLA is large.
>
> **Q5. The upper bound of gradient in Zhu et al. (2022)'s Lemma 8 assumed $\mathbf{W}$ is a guassian random matrix, however, in this paper $\mathbf{W}$ is from SGD iterations.**
>
> Please note that Lemma 8 of Zhu et al. (2022) only requires that the weight parameters be sufficiently close to the initialized weight parameters. In fact, their Lemma 8 is also applied to the output of SGD parameters in their Lemma 11.
>
> Q6. How can feature-label alignment $R$ make a difference in the proof of Theorem 2.
>
> The feature-label alignment  $R$ controls the range we select  $f^\prime \in \mathcal{F}(\boldsymbol{\theta}\_0, R)$ such that $\frac{1}{N} \sum\_{i=1}^N \psi(y\_i f\_i^\prime) = O(1/\sqrt{N})$.
> We need $\frac{1}{N} \sum\_{i=1}^N \psi(y\_i f\_i^\prime) = O(1/\sqrt{N})$ because the second and third term in Lemma 6 is $O(1/\sqrt{N})$.
>
> To achieve this goal, according to the definition of $\mathcal{F}(\boldsymbol{\theta}\_0, R)=\\{ f(\boldsymbol{\theta}\_0) + \langle \nabla f(\boldsymbol{\theta}\_0) , \boldsymbol{\theta}\rangle | \boldsymbol{\theta} \in \mathcal{B}(0, R/\sqrt{m}) \\}$, we have to select $R$ and $\boldsymbol{\theta} \in \mathcal{B}(0, R/\sqrt{m})$ such that
> $$
>     \langle \nabla f\_i(\boldsymbol{\theta}\_0) , \boldsymbol{\theta}\rangle = y\_i \Big( \underbrace{- \log(\exp(N^{-1/2})-1)}\_{B} + \underbrace{\max_i |f_i(\boldsymbol{\theta}_0)|}\_{B^\prime} \Big)
> $$
> As detailed in the proof, one solution of $R$ and $\boldsymbol{\theta}$ is
> \begin{equation*}
>     \text{vec}(\boldsymbol{\theta}) = \mathbf{Q} \Lambda^{-1} \mathbf{P}^\top \hat{\mathbf{y}} \text{ and }
>     R = \mathcal{O}\left(\sqrt{\mathbf{y}^\top (\mathbf{J\mathbf{J}^\top})^{-1} \mathbf{y}}\right),
> \end{equation*}
> where this $R$ is our feature-label alignment.

---

> > ### Comment · Reviewer_g85Z · 2023-11-20
> >
> > Thank you for your detailed response. I still have some concerns about the use of FLA and the proof.
> > 1. For the experiment part, as indicated by Table 1, SToNe is always better than TGAT. However, Figure 3 shows that FLA of SToNe is always larger than TGAT. This observation disputes the argument of using TGA to explain the performance of deep learning models.
> > 2. For the use of lemma 8 in Zhu et al. (2022), if you go deeper into the proof in the original paper, you will find that their proof relies on the fact that $m |W_L|_2^2$ is a random Variables obey chi-square distribution. This comes from the assumption that $W$ follows the normal distribution. There are also many other places where they use the assumption that $W$ follows the normal distribution, e.g. lemma 6 and lemma 7, which have been extensively used in the proof of lemma 8 there.

---

> ### Author Response · Authors · 2023-11-22
>
> Q1.
>
> Thank you for bringing up this point. We note that the generalization bound is not only influenced by FLA (represented by $R$), but also by the neural architecture, particularly the number of layers $D$ and quantity $C$, as indicated by the first term $O(\frac{DRC}{\sqrt{N}})$ in Theorem 1. In Figure 3, SToNe has a slightly higher value for FLA (i.e., larger $R$). However, because TGAT uses the 2-layer architecture (same as its official implementation) while SToNe uses a 1-layer architecture, it results in a larger value for $C$ and $D$ for TGAT (we note that $C$ grows exponentially with the number of layers, and $D$ grows linearly). Taking all of these factors into account, we anticipate that TGAT's overall generalization error will be greater. We only show the comparison on FLA in Figure 3, which does not take neural architecture into account. We appreciate the comment and will make sure to clarify this in the revised version.
>
>
>
> Q2.
>
> We apologize for any confusion and aim to provide additional clarity as outlined below (primarily due to incorrect notation and missing details in the proofs of [1], but the final statements in [1] remain unchanged).
>
> As outlined in our Appendix C.1, the gist of the proof is:
>
> - (i) We show that all the iterates of SGD $\\{\boldsymbol{\theta}_i|i=1,\ldots, N\\}$  and optimal solution $\boldsymbol{\theta}^*$ lie in a ball of radius $\omega$ around the initial weights $\boldsymbol{\theta}_0$ (which is a random matrix and its characteristics can be quantified with high probability, e.g., $\\|\mathbf{D}\\|_2 \leq \rho, \\|\boldsymbol{f}\_{i, \ell-1}\\| \leq \Theta(1), \\|\mathbf{W}_0^{(\ell)}\\|_2 \leq 3, \forall \ell \in [L-1], \\|\boldsymbol{W}_0^{(L)}\\|_2 \leq \sqrt{2}$,
>
> - (ii) We utilize the the bounds in (i) to upper bound the desired quantities about SGD iterates (e.g., for any $\boldsymbol{\theta}_i$ we can show $\\| \mathbf{W}_i^{(\ell)} \\|_2 = \\| \mathbf{W}_i^{(\ell)} - \mathbf{W}_0^{(\ell)} + \mathbf{W}_0^{(\ell)} \\|_2 \leq \\|\mathbf{W}_i^{(\ell)} - \mathbf{W}_0^{(\ell)} \\|\_{\textrm{F}}+ \\|\mathbf{W}_0^{(\ell)} \\|_2 \leq (\omega + 3)$ or similarly $\\|\mathbf{W}_i^{(L)}\\| \leq (\omega + \sqrt{2})$, similarly for other quantities,
>
> - (iii) By proper choice of number of neurons etc we can pick $\omega$  to achieve the desired accuracy. Therefore, by guaranteeing the $\omega$-closeness of SGD iterates to  initialization, we only require the randomness of initial weights to bound the necessary quantities related to SGD iterates.
>
> The Lemma 8 of [1] states that for any $\mathbf{W}, \mathbf{W}^\prime$ close to randomly initialized weight parameters $\mathbf{W}^{(0)}$ with a distance of at most $\omega=1/(3\text{Lip}+1)^{L-1}$,  with high probability, neural network function is almost linear in terms of its weights, i.e., $|f(x_i, \mathbf{W}^\prime) - f(x_i, \mathbf{W}) - \langle \nabla f(x_i, \mathbf{W}), \mathbf{W}^\prime - \mathbf{W} \rangle | = \mathcal{O}(1) $.
> However, in the proof  where the authors utilize the chi-square distribution to bound $\\|\mathbf{W}_L\\|_2$,  it should be about $\\|\mathbf{W}^{(0)}_L\\|_2$ instead of $\\|\mathbf{W}_L\\|_2$, which caused the confusion you pointed out.
> Please note that in the final inequality of Eq. 41 in [1], we need to bound $\\|\mathbf{W}_L\\|_2$, $\\|\mathbf{W}'_L\\|_2$, $\\|\mathbf{D}\_{i,r}\\|_2$,  $\\|\mathbf{W}'\_{\ell} - \mathbf{W}\_{\ell}\\|_2$, and $\\|\boldsymbol{f}\_{i, \ell-1}\\|_2$ etc,
> where $\mathbf{W}$ and $\mathbf{W}'$ are arbitrary matrices (SGD parameters) but close enough to the random initialization weights $\mathbf{W}^{(0)}$. By having a bound on $\\|\mathbf{W}^{(0)}_L\\|_2$, $\\|\mathbf{W}^{(0)}_r\\|_2, r \in [L-1]$, $\\|\mathbf{D}\\|_2$ (which can bounded at initial random parameters with high probability),  under the assumption that $\mathbf{W}, \mathbf{W}' \in \mathcal{B}(\mathbf{W}^{(0)}, \omega)$, we can bound these quantities about $\mathbf{W}, \mathbf{W}' $ as pointed in step (ii) above, which leads to the final inequality at the end of the proof of Lemma 8 in [1]. A similar argument holds for bounding $\\| \nabla\_{\mathbf{W}\_\ell} L_i(\mathbf{W}^{(i)}) \\|_2$ as in the proof of Lemma 10 in [1] (here authors of [1] used the correct notation $\\|\mathbf{W}^{(0)}_r\\|_2 \leq 3$ and $\\|\mathbf{W}^{(0)}_L\\|_2 \leq \sqrt{2}$ etc to bound $\\|\mathbf{W}^{}_L\\|_2$ and other quantities about arbitrary $\mathbf{W} \in \mathcal{B}(\mathbf{W}^{(0)}, \omega)$ to get the final upper bound). We note that an alternative approach is to directly bound these quantities using the machinery developed in [2] (as in lazy training or NTK), but we found the bounding through $\mathbf{W}^{(0)}$ more convenient.  We'd gladly incorporate comprehensive proof details for these lemmas, ensuring correct notation for thoroughness.
>
> [1] Generalization Properties of NAS under Activation and Skip Connection Search
>
> [2] A Convergence Theory for Deep Learning via Over-Parameterization

---

### Official Review · Reviewer_2VYF · 2023-11-01

**Soundness:** 2 fair
**Presentation:** 2 fair
**Contribution:** 2 fair
**Rating:** 5
**Confidence:** 2

**Summary:**

The submission addresses the increasingly researched direction of learning temporal graph structures, which is significantly more challenging compared to static graph learning. While multiple methods have been proposed, there is not yet a good understanding of when or why certain methods outperform others, in particular in terms of how well they predict temporal interactions between nodes in the future. This lack of understanding is something this manuscript aims to address by establishing an upper bound for the expected error in predictions in various proposed temporal graph learning frameworks. In particular this upper bound depends (in an increasing way) on the number of layers in the GNN and RNN based methods, the Lipschitz constant of the activation function in the memory based method, and the feature-label alignment score (FLA) in all considered methods.
This is taken as guiding principle to propose a new method with few layers and small FLA, the latter of which is achieved by a selection of some of the most recent interactions as input data. The proposed new method is shown to behave well (i.e. comparatively or better) in terms of generalizability as well as FLA and running time compared to methods from previous literature.

**Strengths:**

I believe the considered problem of temporal graph learning is relevant and fitting for ICLR. Also the more specific goal of explaining and justifying certain design choices of frameworks to learn temporal graphs is well-motivated. The empirical results suggest that the features identified as important for low generalization error, namely a small number of layers and small FLA, indeed are worth constraining when designing algorithms for temporal graph learning.

**Weaknesses:**

I do not see the formal and strong theoretic claim that the submission frames its theoretic contribution as strong. Specifically, since the given result only established an upper bound, I believe strictly speaking it can merely indicate desirable aspects of methods for temporal graph learning, and not really explain the difference in performance. This is because we have no lower bound relating the generalization error to e.g. the number of layers. This means a small number of layers gives us a good bound on the expected error with high probability, but a larger number of layers does not necessarily lead to  a large expected error.

Minor comments:
- abstract: *a* better overall performance
- page 1, last line: *has* demonstrated that simple
- if my major concern is correct, if would be better to say in the caption of Fig 1: Relationship between the *bound on the* generalization error
- page 4: I think it would be good to introduce the notation used in the definition of loss^{0-1}_i
- Algorithm 1: condition*ed*
- Assumption 1: *have* \ell_2-norm
- Thm. 1: R is fixed twice in the theorem statement
- page 8: could be *thought* of
- page 8: "compatible" is probably the wrong word here. I would suggest competitive or comparable

**Questions:**

Can you discuss my concerns about the framing of the theoretical contributions?

Can you give a formal argument of why memory blocks increase the FLA? Is this always true? (In particular my question relates to the second bullet on the top of page 7.)

---

> ### Author Response · Authors · 2023-11-19
>
> **Q1. No lower bound relating the generalization error to e.g. the number of layers. The framing of the theoretical contributions: the results only have an upper bound, but lower bound is also expected.**
>
>
> Thank you for your suggestion. We fully agree that studying the lower bound is also an interesting and important future direction.
> However, it's worth noting that lower bound analysis is specific to each case (e.g., a specific number of layers, architecture), which may not be feasible if we want to examine the effects of neural architecture selection and data selection on generalization ability.
> To the best of our knowledge, there are no previous works on the generalization lower bound of deep neural networks (previous studies only applies to 2-layer MLP [1] or decision tree [2]), let alone temporal graph learning methods.
> Our paper primarily focuses on the upper bound of generalization, which captures the worst-case generalization ability of various methods.
>
> [1] Lower Bounds on the Generalization Error of Nonlinear Learning Models
>
> [2] Margin-Based Generalization Lower Bounds for Boosted Classifiers
>
> **Q2: Give a formal argument of why memory blocks increase the FLA.**
>
> Feature-label alignment (FLA) increases in memory-based methods potentially because they (1) utilize the stop-gradient operation and (2) aggregate an excessive amount of temporal neighbor information.
>
> As a reminder, FLA is defined as $\mathbf{y}^\top (\mathbf{J} \mathbf{J}^\top)^{-1} \mathbf{y}$, where $\mathbf{J}$ represents a stack of gradients computed for each data point using the randomly initialized weight parameters. Since memory-based methods aggregate information across all time steps, they involve that random weight parameters significantly more times than GNN-based or RNN-based methods. Consequently, their FLA is expected to have a larger lower bound because $\mathbf{y}^\top (\mathbf{J} \mathbf{J}^\top)^{-1} \mathbf{y} \geq 1/\lambda_{\max}(\mathbf{J} \mathbf{J}^\top)$  and $\lambda_{\max}(\mathbf{J} \mathbf{J}^\top)$ is smaller.
> To illustrate why involving more random initialized weight parameters could result in a smaller $\lambda_{\max}(\mathbf{J} \mathbf{J}^\top)$, we use a toy example with the stop-gradient operator:
>
> $$
> \hat{y}\_i = \text{stopgrad}(\mathbf{s}(\mathbf{x}\_i))^\top \mathbf{v},  \mathbf{s}(\mathbf{x}\_i) = \sigma(\ldots \sigma( \sigma(\mathbf{x}\_i \mathbf{W}) \mathbf{W})\ldots \mathbf{W} )
> $$
>
> By definition, the feature-label alignment is computed as:
> $$
>     \mathbf{y}^\top(\mathbf{J} \mathbf{J}^\top)^{-1} \mathbf{y} = \mathbf{y}^\top\left( \sum\_{i=1}^N \mathbf{s}(\mathbf{x}\_i)\mathbf{s}(\mathbf{x}\_i)^\top \right)^{-1} \mathbf{y} \geq \| \mathbf{y} \|\_2^2 / \lambda\_{\max}\left(\sum\_{i=1}^N \mathbf{s}(\mathbf{x}\_i)\mathbf{s}(\mathbf{x}\_i)^\top\right)
> $$
> In general, the more random initialized weight parameters $\mathbf{W}$ are involved, the more randomness there is in $\mathbf{s}(\mathbf{x}\_i)$, leading to a smaller largest eigenvalue of $\sum\_{i=1}^N \mathbf{s}(\mathbf{x}\_i) \mathbf{s}(\mathbf{x}\_i)^\top$.

---

> > ### Comment · Reviewer_2VYF · 2023-11-22
> >
> > Thank you for your response.
> > My second question is completely answered. As for my first question/concern, I am still not completely convinced about the framing of the results and would suggest discussing or rephrasing it. I am not saying that a lower bound is totally necessary for the presented results to be interesting. I would just dispute the fact that giving upper bounds alone e.g. "show that the FLA
> > could potentially explain why the memory-based methods do not outperform GNN-/RNN-based methods, even though their generalization error is less affected as the number of layers/steps increases." To justify claims like this (which occur in this direct quote and in variants in the introduction) I would say a lower bound is necessary. I know this might be a bit pedantic but I did get a feeling that in this aspect the contribution is less strong than what it is being sold as and think this could be something that is fixable with only minor rewriting.

---

> > > ### Author Response · Authors · 2023-11-22
> > >
> > > We are happy that you found our response to the second question convincing.
> > >
> > > With regards to implications of upper bound, we totally agree with you that the statement of our claim should be rewritten as we are not aware of any lower bound  to avoid conveying the wrong message. We will gladly incorporate your comment and make the implications of obtained upper bound  crystal clear in the subsequent version.

---

### Meta-Review · Area_Chair_YjKD · 2023-12-15

**Metareview:**

This paper studies the generalization error of various neural network-based methods for temporal graph learning and proposes an approach called SToNe that enjoys nice properties. However, as Reviewer g85Z pointed out, there are some concerns on the theoretical contributions of the paper. Specifically, the non-smoothness of the ReLU function should be treated with rigor, as the practical implementation does not necessarily imply the soundness of the proposed procedure. Furthermore, the issue concerning the use of the random matrix result in Zhu et al. (2022) has not been completely resolved, and the authors' response will need another round of scrutiny. Based on the above, I regrettably have to reject the paper.

**Justification For Why Not Higher Score:**

The correctness of the technical results is not completely resolved.

**Justification For Why Not Lower Score:**

N/A

---

### Decision · Program_Chairs · 2024-01-16

Reject